# Do LLM Agents Have Regret?
# A Case Study in Online Learning and Games

## Abstract

Despite Large language models' (LLMs) emerging successes, the performance of LLM agents in decision-making has not been fully investigated through quantitative metrics, especially in the multi-agent setting when they interact with each other, a typical scenario in real-world LLM-agent applications. To better understand the limits of LLM agents in these interactive environments, we propose to study their interactions in benchmark decision-making settings in online learning and game theory, through the performance metric of *regret*. We first empirically study the no-regret behaviors of LLMs in canonical (non-stationary) online learning problems, as well as the emergence of equilibria when LLM agents interact through playing repeated games. We then provide some theoretical insights into the no-regret behaviors of LLM agents, under certain assumptions on the supervised pre-training and the rationality model of human decision-makers who generate the data. Notably, we also identify (simple) cases where advanced LLMs such as GPT-4 fail to be no-regret. To promote the no-regret behaviors, we propose a novel *unsupervised* training loss of *regret-loss*, which, in contrast to the supervised pre-training loss, does not require the labels of (optimal) actions. Finally, we establish the *statistical* guarantee of generalization bound for regret-loss minimization, and more importantly, the *optimization* guarantee that minimizing such a loss may *automatically* lead to known no-regret learning algorithms. Our further experiments demonstrate the effectiveness of our regret-loss, especially in addressing the above "regrettable" cases.

## 1. Introduction

Large language model (LLM) agent interacts with the (physical) world in a *dynamic/sequential* way: it uses LLMs as an oracle for reasoning, then acts in the environment based on the reasoning and the feedback it perceives over time. LLM agent has achieved impressive successes in social science (Park et al., 2022; 2023) applications. Besides being *dynamic*, another increasingly captivating feature of LLM-based decision-making is the involvement of *strategic* interactions, oftentimes among multiple LLM agents (Fu et al., 2023; Du et al., 2023; Aher et al., 2023; Park et al., 2023). Moreover, LLMs have also exhibited remarkable potential in solving various games (Bakhtin et al., 2022; Mukobi et al., 2023). These exciting empirical successes call for a rigorous examination and understanding through a theoretical lens of decision-making.

*Regret* has been a core metric in (online) decision-making. It measures how "sorry" the decision-maker is, in retrospect, not to have followed the best prediction in hindsight (Shalev-Shwartz, 2012). It provides not only a sensible way to *evaluate* the sophistication level of online decision-makers, but also a quantitative way to measure their *robustness* against arbitrary (and possibly adversarial) environments. More importantly, it inherently offers a connection to modeling and analyzing *strategic behaviors*: the long-run interaction of no-regret learners leads to certain *equilibrium* when they repeatedly play games (Cesa-Bianchi & Lugosi, 2006). In fact, *no-regret* learning has served as a natural model for predicting and explaining human behaviors in strategic decision-making, with experimental evidence (Erev & Roth, 1998; Nekipelov et al., 2015; Balseiro & Gur, 2019). It has thus been posited as an important model of "rational behavior" in playing games (Blum et al., 2008; Roughgarden, 2015; Roughgarden et al., 2017). Thus, it is natural to ask:

*Can we examine and better understand the online and strategic decision-making behaviors of LLMs through the lens of* regret*?*

Acknowledging that LLM(-agents) are extremely complicated to analyze, to gain some insights into the question, we focus on benchmark decision-making settings: online learning with convex (linear) loss functions, and playing repeated games. We defer a detailed literature review to Appendix B, and summarize our contributions as follows.

**Contributions.** First, we carefully examine the performance of several representative pre-trained LLMs in several

---

[1]Anonymous Institution, Anonymous City, Anonymous Region, Anonymous Country. Correspondence to: Anonymous Author <anon.email@domain.com>.

Preliminary work. Under review by the International Conference on Machine Learning (ICML). Do not distribute.

online decision-making settings, in terms of *regret*. We observe that oftentimes, LLM agents exhibit no-regret behaviors in these (non-stationary) online learning settings, where the loss functions change over time either arbitrarily (and even adversarially), or in playing both representative and randomly generated repeated games. For the latter, equilibria will emerge as the long-term behavior of the multi-LLM interactions. Second, we provide some theoretical insights into the observed no-regret behaviors, based on some hypothetical model of the human decision-makers who generate the data, and certain assumptions on the *supervised pre-training* procedure, a common practice in training large models for decision-making: we make a connection of pre-trained LLMs to the known no-regret algorithm of *follow-the-perturbed-leader* (FTPL) under such assumptions. Third, we also identify (simple) cases where advanced LLMs such as GPT-4 fail to be no-regret. We thus propose a novel *unsupervised* training loss, *regret-loss*, which, in contrast to the supervised pre-training loss, does not require the *labels* of (optimal) actions. We then establish both statistical and optimization guarantees for regret-loss minimization, which, in particular, shows that minimizing such a loss automatically leads to known no-regret learning algorithms. Our further experiments demonstrate the effectiveness of regret-loss, also in addressing the above "regrettable" cases.

## 2. Preliminaries

We defer the notation to Appendix D.

### 2.1. Online Learning & Games

**Online learning.** We consider the online learning setting where an agent interacts with the environment for $T$ rounds, by iteratively making decisions based on the feedback she receives. Specifically, at each time step $t$, the agent chooses her decision policy $\pi_t \in \Pi$ for some bounded domain $\Pi$, and after her commitment to $\pi_t$, a bounded loss function $f_t : \Pi \to [-B, B]$ for some constant $B > 0$ is chosen by the environment, potentially in an adversarial fashion. The agent thus incurs a loss of $f_t(\pi_t)$, and will update her decision to $\pi_{t+1}$ using the feedback. We focus on the most basic setting where the agent chooses actions from a finite set $\mathcal{A}$ every round, which is also referred to as the *Experts Problem* (Littlestone & Warmuth, 1994; Hazan, 2016), without loss of much generality (c.f. Appendix D.4 for a discussion). In this case, $\Pi$ becomes the simplex over $\mathcal{A}$, i.e., $\Pi = \Delta(\mathcal{A})$, and $f_t(\pi_t) = \langle \ell_t, \pi_t \rangle$ for some loss *vector* $\ell_t \in \mathbb{R}^d$ that may change over time, where $d := |\mathcal{A}|$. Hereafter, we will by default refer to this setting that does *not* make any assumptions on the loss sequence $(\ell_t)_{t \in [T]}$ simply as *online learning*. Moreover, if the loss functions change over time (usually with certain bounded variation), we will refer to it as *non-stationary online learning* for short.

**Repeated games.** Consider a normal-form game $\mathcal{G} = \langle N, \{\mathcal{A}_n\}_{n \in [N]}, \{r_n\}_{n \in [N]} \rangle$, where $N$ is the number of players, $\mathcal{A}_n$ and $r_n : \mathcal{A}_1 \times \cdots \times \mathcal{A}_N \to [-B, B]$ are the action set and the payoff function of player $n$, respectively.

The $N$ players repeatedly play the game for $T$ rounds, each player $n$ maintains a strategy $\pi_{n,t} \in \Delta(\mathcal{A}_n)$ at time $t$, and takes action $a_{n,t} \sim \pi_{n,t}(\cdot)$. The $a_t = (a_{1,t}, \cdots, a_{N,t})$ determines the payoff of each player at time $t$, $\{r_n(a_t)\}_{n \in [N]}$.

### 2.2. Performance Metric: Regret

We now introduce *regret*, the core performance metric used in online learning and games. For a given algorithm $\mathscr{A}$, let $\pi_{\mathscr{A},t}$ denote the decision policy of the agent at time $t$ generated by $\mathscr{A}$. Then, the regret, which is the difference between the accumulated (expected) loss incurred by implementing $\mathscr{A}$ and that incurred by the best-in-hindsight fixed decision, can be defined as

$$\text{Regret}_{\mathscr{A}}\left((f_t)_{t \in [T]}\right) := \sum_{t=1}^{T} f_t(\pi_{\mathscr{A},t}) - \inf_{\pi \in \Pi} \sum_{t=1}^{T} f_t(\pi).$$

In the Experts Problem, the definition is instantiated as $\text{Regret}_{\mathscr{A}}((\ell_t)_{t \in [T]}) := \sum_{t=1}^{T} \langle \ell_t, \pi_{\mathscr{A},t} \rangle - \inf_{\pi \in \Pi} \sum_{t=1}^{T} \langle \ell_t, \pi \rangle$. An algorithm $\mathscr{A}$ is referred to as being *no-regret*, if $\text{Regret}_{\mathscr{A}}((f_t)_{t \in [T]}) \sim o(T)$, i.e., the regret grows sublinearly in $T$. Widely-known no-regret algorithms include follow-the-regularized-leader (FTRL) (Shalev-Shwartz & Singer, 2007), follow-the-perturbed-leader (Kalai & Vempala, 2005) (See Appendix D.3 for more details). In non-stationary online learning, the metric of *dynamic regret* (Zinkevich, 2003) is used, where the *comparator* changes over time.

## 3. Do Pre-Trained LLMs Have Regret? Experimental Validation

In this section, we explore the no-regret behaviors of representative LLMs (i.e., GPT-4 Turbo, GPT-4, and GPT-3.5 Turbo, Mixtral-8x7b-instruct, and Llama-3-70B-instruct), in the context of online learning and games. All experiments with LLMs are conducted using the public OpenAI (Openai, 2023) or LLM Engine (LLM Engine, 2023) Python API. We provided intuition as to why pre-trained LLM might be expected to be no-regret in Appendix E.1.

**Interaction protocol.** To enable the sequential interaction with LLMs, we first describe the setup and objective of our experimental study. At each round, we incorporate the entire history of loss vectors of past interactions into our prompts, as concatenated texts, and ask the LLM agent to determine a policy that guides the decision-making for the next round. Note that since we hope to *evaluate* the sophistication level of pre-trained LLMs through online learning or games, we only provide simple prompts that she should utilize the history information, without providing explicit rules of *how* to make use of the history information, nor asking her to *minimize regret* (in any sense). We defer detailed description to Appendix E.9, and an illustration of the protocol for playing repeated games is given in Figure E.1.

### 3.1. Framework for No-Regret Behavior Validation

Before delving into the results, we note that to the best of our knowledge, we are not aware of any principled framework for validating no-regret behaviors with finite-time ex-

perimental data. Therefore, we propose two frameworks, trend-checking/regression-based framework, to rigorously validate the no-regret behavior of algorithms over a *finite T*, which might be of independent interest. More details are deferred to Appendix E.3.

### 3.2. Results: Online Learning

We now present the experimental results of pre-trained LLMs in online learning in: 1) arbitrarily changing environments, 2) non-stationary environments, and 3) bandit-feedback environments. Results for 2) and 3) are deferred to Appendix E.5 and E.6.

**Online learning in arbitrarily changing environment.** We first consider the setting with arbitrarily changing environments, which are instantiated as follows: 1) *Randomly-generated loss sequences*. At every timestep, we generate a random loss vector $\ell_t \sim \text{Unif}(\times_{i=1}^{d}[\min\{x_i, y_i\}, \max\{x_i, y_i\}])$ for $\{x_i, y_i \sim \text{Unif}(0, 10)\}_{i \in [d]}$ or $\ell_t \sim \mathcal{N}(\boldsymbol{\mu}_d, I)$ with clipping to $[0, 10]$ to ensure boundedness of the loss, where $\boldsymbol{\mu}_d \sim \text{Unif}([0, 10]^d)$, such that the loss vectors of different timesteps can be *arbitrarily different*. 2) *Loss sequences with certain trends*. Although many real-world environments may change, they often change following certain patterns. Therefore, we consider two representative trends, the *linear* and *periodic* (sinusoid) trend. We sample $a, b \sim \text{Unif}([0, 10]^d)$ and let $\ell_t = (b - a)\frac{t}{T} + a$ for the linear trend and $\ell_t = 5(1 + \sin(at + b))$ for the periodic trend. In the experiments, we choose $d = 2$. The average regret (over multiple randomly generated instances) performance is presented in Figure E.2, where we compare GPT-4 with well-known no-regret algorithms, FTRL with entropy regularization and FTPL with Gaussian perturbations (with tuned parameters). These pre-trained LLMs are indeed no-regret and can have lower regret values than these baselines.

**Behavioral pattern of LLMs.** To understand how LLMs make decisions at each time step, we provided example outputs of LLMs *explaining* how they generate their policies in Appendix E.12. We find LLMs tend to use the history of the reward vectors by looking at their *sum/average*, and tend to introduce *randomization* in decision-making. These are known to be key to achieving no-regret behaviors in online learning and games (Cesa-Bianchi & Lugosi, 2006).

### 3.3. Results: Multi-Player Repeated Games

We now consider the setting when multiple LLMs make online strategic decisions in a *shared* environment repeatedly. Specifically, at each round, the loss vectors each agent receives are determined by both her payoff matrix and the strategies of all other agents. Note that the payoff matrix is not directly revealed to the LLM agent, but she has to make decisions in a completely online fashion based on the payoff vector marginalized by the opponents' strategies (See Figure E.1 for a prompt example). This is a typical scenario in learning in (repeated) games (Fudenberg & Levine, 1998).

**Randomly generated games.** To validate the no-regret behavior of LLMs, we also test on 50 randomly generated

three-player general-sum games, and 50 randomly generated four-player general-sum games, where each entry of the payoff matrix is sampled randomly from $\text{Unif}([0, 10])$. These are larger and more challenging settings than the structured and representative ones above.

We summarize the experimental results in Figure E.4, which are similar to the above in the online setting: for all types of games, pre-trained LLMs achieve sublinear regret, which is often lower than that obtained by FTRL/FTPL for most games. We provide six instances of three-player general-sum games and six instances of four-player general-sum games in Figure E.5 and Figure E.6, respectively. Occasionally, GPT-4 even provides a negative regret value.

### 3.4. Pre-Trained LLM Agents May Still Have Regret

It seems tempting to conclude that pre-trained LLMs are indeed no-regret in both online learning and playing repeated games. However, is this capability *universal*? We show that the no-regret property might break for LLM agents if the loss vectors are generated in a more adversarial way. Details are deferred to Appendix E.8.

## 4. Why Do Pre-Trained LLMs (Not) Have Regret? A Hypothetical Model and Some Theoretical Insights

We now provide some plausible explanations about the no-regret behavior of pre-trained LLMs, which are *hypothetical* by nature, since to the best of our knowledge, the details of pre-training these popular LLMs, regarding data distribution, training algorithm, etc., have not been revealed. We instead make the explanations based on some common assumptions in the literature for modeling human behaviors, and the recent literature on understanding LLMs/Transformers. We defer the definition of quantal response against multiple losses to Appendix F.2.1, which has been investigated in the learning-in-games and behavioral economics literature.

Pre-training of LLMs is predominantly based on *next-token prediction*. When applying LLMs to sequential decision-making, the model receives the context of the decision-making task as $(x_1, x_2, \cdots, x_N)$ and then generates $(x_{N+1}, \cdots, x_M)$ encoding the *action* for some $N, M \in \mathbb{N}^+$ and $N < M$, where each $x_i \in \mathcal{V}$ represents one *natural language token* for $i \in [M]$, and $\mathcal{V}$ is the finite token set. Meanwhile, large models are often (pre-)trained under several *fixed/stationary* environments (Laskin et al., 2023; Lin et al., 2024; Lee et al., 2023; Reed et al., 2022), which may limit their ability to handle *arbitrary/non-stationary/adversarial* loss sequences in our online learning setup. Thus, it is natural to ask: *Is it possible to have no-regret behaviors emerging as a consequence of this (optimal) action prediction, under only a fixed pre-training distribution of environments?*

Here we analyze a standard pre-training objective on a token sequence distribution $x_{1:N_{t+1}} \sim P_t^{text}$ for given $t \in [T]$, which is the expected log-likelihood maximization for next-

token prediction over $\Theta$, the parameter space of the LLM:

$$\max_{\theta \in \Theta} \quad \mathbb{E}_{x_{1:N_{t+1}} \sim P_t^{text}} \sum_{j=1}^{N_{t+1}} \log \mathrm{LLM}_\theta \left( x_j \,|\, x_{1:j-1} \right), \quad (4.1)$$

where we define $\mathrm{LLM}_\theta \left( x_1 \,|\, x_{1:0} \right) = \mathrm{LLM}_\theta \left( x_1 \right)$.

For the pre-training distribution, we model it as follows: there exists a latent variable $z$, representing the loss for the underlying *static* decision-making problem. We defer a detailed explanation for $z$ ad assumptions for pre-training distribution in Appendix F.3.

**Theorem 4.1** (Informal: Emergence of no-regret behavior)**.** *Suppose Assumption 1 holds with both the prior distribution on $z$ and the likelihood on $\{\ell_i \,|\, z\}_{i \in [t]}$ being Gaussian, and $x_{N_t+1:N_{t+1}}$ encodes the optimal action for $z$. Then, as long as the function class of $\mathrm{LLM}_\theta$ is expressive enough, with $\theta^\star$ being a maximizer of Equation (4.1), the behavior of $\mathrm{LLM}_{\theta^\star}$ follows quantal response, and also achieve no (dynamic) regret for (non-stationary) online learning with full-information/bandit feedback for arbitrary loss vectors.*

The formal statement and proof are deferred to Appendix F.7. The significance of our results lies in that even when pre-training is conducted solely with loss vectors from *stationary* distributions, it still enables the *emergence of no-regret behavior* in online learning against *potentially adversarial losses*. Key in the proof is an interesting connection of pre-trained LLM models to FTPL. Finally, we point out its implications for playing games in Appendix F.7.1. We also defer the experiment to compare theoretical results and LLMs' behavior in Appendix F.10.

Finally, we acknowledge that for existing pre-trained LLMs like GPT-4, the canonical assumptions above, though may be further relaxed (c.f. Remark F.3), may not hold in general. More importantly, the *supervision labels* may be sometimes imperfect or unavailable during the dataset collection. These caveats motivate the study in our next section.

# 5. Provably Promoting No-Regret Behavior by an Unsupervised Loss

In light of the observations in Section 3, we ask the question:

*Is there a way to enhance the no-regret property of LLM agents, **without** (optimal) action labels?*

We propose to train LLMs with a new *unsupervised learning* loss that naturally provides no-regret behaviors.

## 5.1. A New Unsupervised Training Loss: *Regret-Loss*

Intuitively, our new training loss is designed to enforce the trained LLM to minimize the regret under an arbitrary sequence of loss vectors. We define the training loss as

$$\mathcal{L}(\theta) := \max_{\ell_1, \ldots, \ell_T} \quad \mathrm{Regret}_{\mathrm{LLM}_\theta} \left( (\ell_t)_{t \in [T]} \right) \quad (5.1)$$

where $\|\ell_t\|_\infty \leq B$ for $t \in [T]$. As discussed in (Kirschner et al., 2023), directly minimizing the max regret can be computationally challenging, except for superficially simple problems. Therefore, we provide a general class of surrogate losses to approximate Equation (5.1) ($\mathcal{L}(\theta, k, N)$):

$$\mathbb{E} \left[ \frac{\sum_{j \in [N]} h(\mathrm{Regret}_{\mathrm{LLM}_\theta}((\ell_t^{(j)})_{t \in [T]})) f(\mathrm{Regret}_{\mathrm{LLM}_\theta}((\ell_t^{(j)})_{t \in [T]}), k)}{\sum_{j \in [N]} f(\mathrm{Regret}_{\mathrm{LLM}_\theta}((\ell_t^{(j)})_{t \in [T]}), k)} \right],$$
$$(5.2)$$

where $k \in \mathbb{N}^+$, $N \in \mathbb{N}^+$, and regularity conditions for $f$ and $h$ (Appendix G.1). Examples of such an $f$ include $f(x, k) = x^k$ and $\exp(kx)$. In Appendix G.3, we prove that under certain regularity conditions of $f$ and $h$, we have

$$\lim_{N, k \to \infty} \mathcal{L}(\theta, k, N) = h \left( \max_{\ell_1, \ldots, \ell_T} \mathrm{Regret}_{\mathrm{LLM}_\theta} ((\ell_t)_{t \in [T]}) \right).$$

We will hereafter refer to Equation (5.2) as the *regret-loss*.

## 5.2. Generalization and Regret Guarantees of Regret-Loss Minimization

We first establish a *statistical* guarantee under general parameterizations of $\mathrm{LLM}_\theta$ that is Lipschitz with respect to $\theta$, including the Transformer-based models as used in GPT-4 and most existing LLMs (see Proposition 2).

**Theorem 5.1.** (Regret, Informal)**.** *Under regular conditions on $f, h$, with high probably, we have*

$$h \left( \lim_{N \to \infty} \lim_{k \to \infty} \max_{\|\ell_t\|_\infty \leq B} \mathrm{Regret}_{\mathrm{LLM}_{\widehat{\theta}_{k,N,N_T}}} \left( (\ell_t)_{t \in [T]} \right) \right)$$

$$\leq h \left( \inf_{\theta \in \Theta} \max_{\|\ell_t\|_\infty \leq B} \mathrm{Regret}_{\mathrm{LLM}_\theta} \left( (\ell_t)_{t \in [T]} \right) \right) + \widetilde{\mathcal{O}} \left( \sqrt{\frac{d_\theta}{N_T}} \right).$$

We defer the proof of the theorem to Appendix G.5. Therefore, if additionally, the LLM parameterization (i.e., Transformers) can realize a no-regret algorithm (as to be shown next), then Theorem 5.1 means that with a large enough $N_T$, the learned $\mathrm{LLM}_{\widehat{\theta}_{k,N,N_T}}$ becomes a *no-regret* learner, i.e., $\mathrm{Regret}_{\mathrm{LLM}_{\widehat{\theta}_{k,N,N_T}}} \left( (\ell_t)_{t \in [T]} \right) = o(T)$. Finally, it is folklore that when multiple such LLMs interact, a coarse correlated equilibrium will emerge in the long term.

## 5.3. Minimizing Regret-Loss Can Automatically Produce Online Learning Algorithms

Despite the generality of the previous results, one cannot use an *infinitely large* $N$ and $k$ in practice. Hence, we now provide results when $N$ is finite, for the specific parameterization of the LLMs using Transformers. We focus on single-layer (linear) self-attention models, as in most recent theoretical studies of Transformers (Ahn et al., 2023; Zhang et al., 2023a; Mahankali et al., 2023), and $N = 1$. Under this condition, we have the following informal theorem

**Theorem 5.2** (Informal, emergence of FTRL)**.** *The configuration of the single-layer linear self-attention model is equivalent to FTRL with $L_2$-regularizer.*

We defer a detailed explanation to Appendix G.6. Theorem 5.2 shows the capability of self-attention models: it can realize online learning algorithms, thanks to our regret-loss. In particular, this can be achieved automatically by optimizing the new loss, *without* hard-coding the parameters of the Transformer. Lastly, we also provide experimental results for minimizing our *regret-loss* in various environments in Appendix G.12.

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

# Supplementary Materials for
## "Do LLM Agents Have Regret? A Case Study in Online Learning and Games"

## Contents

## A. Societal Impact

Our work aimed at a better understanding of LLMs for decision-making through the lens of regret minimization, with rigorous mathematical analysis. It is a theory-oriented work. As such, we do not anticipate any direct positive or negative societal impact from this research.

## B. Related Work

**LLM(-agent) for decision-making.**    The impressive capability of LLMs for *reasoning* (Bubeck et al., 2023; Achiam et al., 2023; Wei et al., 2022b;a; Srivastava et al., 2023; Yao et al., 2023a) has inspired a growing line of research on *LLM for (interactive) decision-making*, i.e., an LLM-based autonomous agent interacts with the environment by taking actions repeatedly/sequentially, based on the feedback it perceives. Some promises have been shown from a *planning* perspective (Hao et al., 2023; Valmeekam et al., 2023; Huang et al., 2022; Shen et al., 2023). In particular, for embodied AI applications, e.g., robotics, LLMs have achieved impressive performance when used as the controller for decision-making (Ahn et al., 2022; Yao et al., 2023b; Shinn et al., 2023; Wang et al., 2023b; Driess et al., 2023; Significant Gravitas, 2023). However, the performance of decision-making has not been rigorously characterized via the regret metric in these works. Very recently, (Liu et al., 2023b) has proposed a principled architecture for LLM-agent, with provable regret guarantees in stationary and stochastic decision-making environments, under the Bayesian adaptive Markov decision processes framework. In contrast, our work focuses on online learning and game-theoretic settings, in potentially adversarial and non-stationary environments. Moreover, (first part of) our work focuses on *evaluating* the intelligence level of LLM per se in decision-making (in terms of the regret metric), while (Liu et al., 2023b) focused on *developing* a new architecture that uses LLM as an oracle for reasoning, together with memory and specific planning/acting subroutines, *to achieve* sublinear (Bayesian) regret, in stationary and stochastic environments.

**LLMs in multi-agent environments.**    The interaction of multiple LLM agents has garnered significant attention lately. For example, (Fu et al., 2023) showed that LLMs can autonomously improve each other in a negotiation game by playing and criticizing each other. Similarly, (Du et al., 2023; Liang et al., 2023; Xiong et al., 2023; Chan et al., 2024; Li et al., 2023c) showed that multi-LLM *debate* can improve the reasoning and evaluation capabilities of the LLMs. (Qian et al., 2023; Schick et al., 2023; Wu et al., 2023) demonstrated the potential of multi-LLM interactions and collaboration in software development, writing, and problem-solving, respectively. (Zhang et al., 2024) exhibited a similar potential in embodied cooperative environments. More formally, multi-LLM interactions have also been investigated under a *game-theoretic* framework, to characterize the *strategic* decision-making of LLM agents. (Bakhtin et al., 2022; Mukobi et al., 2023) and (Xu et al., 2023b;a) have demonstrated the promise of LLMs in playing Diplomacy and WereWolf games, respectively, which are both language-based games with a mixture of competitive and cooperative agents. Note that these works utilized LLM to solve a specific rather than a general game. Related to our work, (Brookins & DeBacker, 2023; Akata et al., 2023; Lorè & Heydari, 2023; Brookins & DeBacker, 2023; Fan et al., 2023) have also used (repeated) matrix games as a benchmark to evaluate the reasoning capability and rationality of LLM agents. In contrast to our work, these empirical studies have not formally investigated LLM agents using the metric of *regret*, nor through the lenses of *online learning* and *equilibrium-computation*, which are all fundamental in modeling and analyzing strategic multi-agent interactions. Moreover, our work also provides theoretical results to explain and further enhance the no-regret property of LLM agents.

**LLMs & Human/Social behavior.**    LLMs have also been used to *simulate* the behavior of human beings, for social science and economics studies (Engel et al., 2023). The extent of LLMs simulating human behavior has been claimed as a way to evaluate the level of its intelligence in a controlled environment (Aher et al., 2023; Tsai et al., 2023). For example, (Li et al., 2023b; Hong et al., 2024; Zhao et al., 2023) showed that by specifying different "roles" to LLM agents, certain collaborative/competitive behaviors can emerge. (Argyle et al., 2023) showed that LLMs can emulate response distributions from diverse human subgroups, illustrating their adaptability. (Horton, 2023) argued that an LLM, as a computational model of humans, can be used as *homo economicus* when given endowments, information, preferences, etc., to gain new economic insights by simulating its interaction with other LLMs. (Park et al., 2022; 2023) proposed scalable simulators that can generate realistic social behaviors emerging in populated and interactive social systems, and the emerging behaviors of LLM agents in society have also been consistently observed in (Chen et al., 2024; 2023). (Li et al., 2023d;a) studied the opinion/behavioral dynamics of LLM agents on social networks. These empirical results have inspired our work, which can be viewed as an initial attempt towards quantitatively understanding the *emerging behavior* of LLMs as computational human models, given the well-known justification of *equilibrium* being a long-run emerging behavior of *learning dynamics*

(Fudenberg & Levine, 1998) and strategic interactions (Young, 2004; Camerer, 2011).

**Transformers & In-context-learning.** LLMs nowadays are predominantly built upon the architecture of Transformers (Vaswani et al., 2017). Transformers have exhibited a remarkable capacity of *in-context-learning* (ICL), which can construct new predictors from sequences of labeled examples as input, without further parameter updates. This has enabled the *few-shot learning* capability of Transformers (Brown et al., 2020; Garg et al., 2022; Min et al., 2022). The empirical successes have inspired burgeoning theoretical studies on ICL. (Xie et al., 2022) used a Bayesian inference framework to explain how ICL works, which has also been adopted in (Wang et al., 2023a; Jiang, 2023). (Akyürek et al., 2023; Von Oswald et al., 2023; Dai et al., 2023; Giannou et al., 2023) showed (among other results) that ICL comes from the fact that Transformers can implement the gradient descent (GD) algorithm. (Bai et al., 2023) further established that Transformers can implement a broad class of machine learning algorithms in context. Moreover, (Ahn et al., 2023; Zhang et al., 2023a; Mahankali et al., 2023) proved that a *minimizer* of the certain training loss among single-layer Transformers is equivalent to a single step of GD for linear regression. (Li et al., 2023e) established generalization bounds of ICL from a multi-task learning perspective. (Zhang et al., 2023b) argued that ICL implicitly implements Bayesian model averaging, and can be approximated by the attention mechanism. They also established a result on some *regret* metric. However, the regret notion is not defined for (online) decision-making, and is fundamentally different from ours that is standard in online learning and games. Also, we provide extensive experiments to validate the no-regret behavior by our definition. More recently, the ICL property has also been generalized to decision-making settings. (Laskin et al., 2023; Lee et al., 2023; Lin et al., 2024) investigated the in-context reinforcement learning (RL) property of Transformers under supervised pre-training, for solving stochastic bandits and Markov decision processes. In contrast, our work focuses on online learning settings with an arbitrary and *potentially adversarial* nature, as well as *game-theoretic* settings. We also provide a new *unsupervised* loss to promote the no-regret behavior in our settings.

**Online learning and games.** Online learning has been extensively studied to model the decision-making of an agent who interacts with the environment sequentially, with a potentially arbitrary sequence of loss functions (Shalev-Shwartz, 2012; Hazan, 2016), and has a deep connection to game theory (Cesa-Bianchi & Lugosi, 2006). In particular, regret, the difference between the incurred accumulated loss and the best-in-hindsight accumulated loss, has been the core performance metric, and a good online learning algorithm should have regret at most sublinear in time $T$ (i.e., of order $o(T)$), which is referred to as being *no-regret*. Many well-known algorithms can achieve no-regret against *arbitrary* loss sequences, e.g., multiplicative weight updates (MWU)/Hedge (Freund & Schapire, 1997; Arora et al., 2012), EXP3 (Auer et al., 2002), and more generally follow-the-regularized-leader (FTRL) (Shalev-Shwartz & Singer, 2007) and follow-the-perturbed-leader (FTPL) (Kalai & Vempala, 2005). In the bandit literature (Lattimore & Szepesvári, 2020; Bubeck et al., 2012), such a setting without any statistical assumptions on the losses is also referred to as the *adversarial/non-stochastic* setting. Following the conventions in this literature, the online settings we focus on shall not be confused with the stationary and *stochastic*(-bandit)/(-reinforcement learning) settings that have been explored in several other recent works on *Transformers for decision-making* (Lee et al., 2023; Lin et al., 2024). Centering around the regret metric, our work has also explored the non-stationary bandit setting (Besbes et al., 2014), as well as the repeated game setting where the environment itself consists of strategic agents (Cesa-Bianchi & Lugosi, 2006).

## C. Deferred Background

## D. Notation

We use $\mathbb{N}$ and $\mathbb{N}^+$ to denote the sets of non-negative and positive integers, respectively. For a finite set $\mathcal{S}$, we use $\Delta(\mathcal{S})$ to denote the simplex over $\mathcal{S}$. For $d \in \mathbb{N}^+$, we define $[d] := \{1, 2, \ldots, d\}$. For two vectors $x, y \in \mathbb{R}^d$, we use $\langle x, y \rangle$ to denote the inner product of $x$ and $y$. We define $\mathbf{0}_d$ and $\mathbf{1}_d$ as a $d$-dimensional zero or one vector, and $O_{d \times d}$ and $I_{d \times d}$ as a $d \times d$-dimensional zero matrix and identity matrix, respectively. We omit $d$ when it is clear from the context. We define $e_i$ as a unit vector (with proper dimension) whose $i$-th coordinate equal to 1. For $p \in \mathbb{R}^d$, $R > 0$ and $C \subseteq \mathbb{R}^d$ is a convex set, define $B(p, R, \|\cdot\|) := \{x \in \mathbb{R}^d \mid \|x - p\| \le R\}$, $\texttt{Proj}_{C, \|\cdot\|}(p) = \operatorname{argmin}_{x \in C} \|x - p\|$ (which is well defined as $C$ is a convex set), and $\texttt{clip}_R(x) := [\texttt{Proj}_{B(0, R, \|\cdot\|_2), \|\cdot\|_2}(x_i)]_{i \in [d]}$. Define $\texttt{Softmax}(x) := \left( \frac{e^{x_i}}{\sum_{i \in [d]} e^{x_i}} \right)_{i \in [d]}$ and $\texttt{ReLU}(x) = \max(0, x)$ for $x \in \mathbb{R}^d$. For $A \in \mathbb{R}^{m \times n}$ with $A_i$ denoting its $i$-th column, we define $\|A\|_{\text{op}} := \max_{\|x\|_2 \le 1} \|Ax\|_2$, $\|A\|_{2,\infty} := \sup_{i \in [n]} \|A_i\|_2$, $\|A\|_F$ as the Frobenius norm, and $A_{-1} := A_n$ to denote the last column vector of $A$. We define $\mathbb{R}^+ := \{x \mid x \ge 0\}$. For a set $\Pi$, define $\text{diam}(\Pi, \|\cdot\|) := \sup_{\pi_1, \pi_2 \in \Pi} \|\pi_1 - \pi_2\|$. We define $\mathbb{1}(\mathcal{E}) := 1$ if $\mathcal{E}$ is true,

and $\mathbb{1}(\mathcal{E}) := 0$ otherwise. For a random variable sequence $(X_n)_{n\in\mathbb{N}}$ and random variables $X, Y$, we denote $F_X$ as the cumulative distribution function of a random variable $X$, $X_n \xrightarrow{p} X$ if $\forall \epsilon > 0, \lim_{n\to\infty} \mathbb{P}(|X_n - X| > \epsilon) = 0$, $X_n \xrightarrow{d} X$ if $\lim_{n\to\infty} F_{X_n}(x) = F_X(x)$ for all $x$ where $F_X(x)$ is continuous, $X \stackrel{d}{=} Y$ if $F_X(x) = F_Y(x)$ for all $x$, $X_n \xrightarrow{a.s.} X$ if $\mathbb{P}(\lim_{n\to\infty} X_n = X) = 1$, and $\mathrm{esssup}(X) := \inf\{M \in \mathbb{R} : \mathbb{P}(X > M) = 0\}$. For a random variable $X$, we use $\mathrm{supp}(X)$ to denote its support. For functions $f, g : \mathbb{R} \to \mathbb{R}$, we define $g(x) = \mathcal{O}(f(x))$ if there exist $x_0, M < \infty$ such that $|g(x)| \leq M|f(x)|$ for all $x > x_0$. We use $f'$ to denote the derivative of $f$. Let $F : \Omega \to \mathbb{R}$ be a continuously-differentiable, strictly convex function defined on a convex set $\Omega$. The Bregman divergence associated with $F$ for points $p, q$ is defined as $D_F(p, q) := F(p) - F(q) - \langle \nabla F(q), p - q \rangle$. For a sequence $(\ell_t)_{t\in[T]}$ for some $T \in \mathbb{N}^+$, we define $\ell_{a:b} := (\ell_a, \cdots, \ell_b)$ for $1 \leq a \leq b \leq T$. If $a > b$, we define $\ell_{a:b} = \emptyset$.

### D.1. Additional Definitions for Appendix

**(Linear) Self-attention.** One key component in Transformers (Vaswani et al., 2017), the backbone of modern language models, is the *(self-)attention* mechanism. For simplicity, we here focus on introducing the *single-layer* self-attention architecture. The mechanism takes a sequence of vectors $Z = [z_1, \ldots, z_t] \in \mathbb{R}^{d\times t}$ as input, and outputs some sequence of $[\widehat{z}_1, \ldots, \widehat{z}_t] \in \mathbb{R}^{d\times t}$. For each $i \in [t]$ where $i > 1$, the output is generated by $\widehat{z}_i = (Vz_{1:i-1})\sigma((Kz_{1:i-1})^\mathsf{T}(Qz_i))$, where $z_{1:i-1}$ denotes the 1 to $i-1$ columns of $Z$, $\sigma$ is either the `Softmax` or `ReLU` activation function, and for the initial output, $\widehat{z}_1 = \mathbf{0}_d$. Here, $V, Q, K \in \mathbb{R}^{d\times d}$ are referred to as the *Value*, *Query*, and *Key* matrices, respectively. Following the theoretical framework in (Von Oswald et al., 2023; Mahankali et al., 2023), we exclude the attention score for a token $z_i$ in relation to itself. For theoretical analysis, we also consider the *linear* self-attention model, where $\widehat{z}_i = (Vz_{1:i-1})((Kz_{1:i-1})^\mathsf{T}(Qz_i))$. We write this (linear) self-attention layer's output as $\mathtt{(L)\,SA}_{(V,Q,K)}(Z)$. We define an $M$-head self-attention layer with $\theta = \{(V_m, Q_m, K_m)\}_{m\in[M]}$ as $\mathtt{M\text{-}(L)\,SA}_\theta(Z) := \sum_{m=1}^{M} \mathtt{(L)\,SA}_{(V_m,Q_m,K_m)}(Z)$. We define $\|\cdot\|_{\mathtt{M\text{-}(L)\,SA}}$ as $\|\theta\|_{\mathtt{M\text{-}(L)\,SA}} := \max_{m\in[M]} \{\|Q_m\|_{\mathrm{op}}, \|K_m\|_{\mathrm{op}}\} + \sum_{m=1}^{M} \|V_m\|_{\mathrm{op}}$.

**Transformers.** For a multi-layer perceptron (MLP) layer, it takes $Z = [z_1, \ldots, z_t] \in \mathbb{R}^{d\times t}$ as input, with parameter $\theta = (W_1, W_2) \in \mathbb{R}^{d'\times d} \times \mathbb{R}^{d\times d'}$ such that for each $i \in [t]$, the output is $\widehat{z}_i := W_2\sigma(W_1 z_i)$ where $\sigma$ is either `Softmax` or `ReLU`. We write the output of an MLP layer with parameter $\theta$ as $\mathtt{MLP}_\theta(Z)$. Defining $\|\cdot\|_{\mathtt{MLP}}$ as $\|\theta\|_{\mathtt{MLP}} := \|W_1\|_{\mathrm{op}} + \|W_2\|_{\mathrm{op}}$ and $\mathtt{ResNet}(f, Z) := Z + f(Z)$, we can define an $L$-layer Transformer with parameter $\theta = (\theta^{(lm)}, \theta^{(la)})_{l\in[L]}$ as

$$\mathtt{TF}_\theta(Z) := Z^{(L)},$$

where the output $Z^{(L)}$ is defined iteratively from $Z^{(0)} = \mathtt{clip}_R(Z) := \min(-R, \max(R, Z))$ and

$$Z^{(l)} = \mathtt{clip}_R\left(\mathtt{ResNet}\left(\mathtt{MLP}_{\theta^{(la)}}, \mathtt{ResNet}\left(\mathtt{M\text{-}(L)\,SA}_{\theta^{(lm)}}, Z^{(l-1)}\right)\right)\right),$$

for some $R > 0$. We define a class of Transformers with certain parameters as $\Theta_{d,L,M,d',B_{\mathrm{TF}}} := \{\theta = (\theta^{(lm)}, \theta^{(la)})_{l\in[L],m\in[M]} : \|\theta\|_{\mathrm{TF}} \leq B_{\mathrm{TF}}\}$, where $M$ is the number of heads of self-attention,

$$\|\theta\|_{\mathrm{TF}} := \max_{l\in[L]}\left\{\|\theta^{(la)}\|_{\mathtt{M\text{-}(L)\,SA}} + \|\theta^{(lm)}\|_{\mathtt{MLP}}\right\}, \tag{D.1}$$

and $B_{\mathrm{TF}} > 0$ is some constant. When it is clear from the context, we may omit the subscripts and write it as $\Theta$ for simplicity. We assume $R$ to be sufficiently large such that `clip` does not take effect on any of our approximation results.

### D.2. In-Context Learning

In-context learning is an emergent behavior of LLMs (Brown et al., 2020), which means that these models can adapt and learn from a limited number of examples provided within their immediate input context. In in-context learning, the prompt is usually constituted by a length of $T$ in-context (independent) examples $(x_t, y_t)_{t\in[T]}$ and $(T+1)$-th input $x_{T+1}$, so the $\mathrm{LLM}((z_t)_{t\in[T]}, x_{T+1})$ provides the inference of $y_{T+1}$, where $z_t = (x_t, y_t)$.

### D.3. Online Learning Algorithms

**Follow-the-regularized-leader (FTRL).** The Follow-the-Regularized-Leader (FTRL) algorithm (Shalev-Shwartz, 2007) is an iterative method that updates policy based on the observed data and a regularization term. The idea is to choose the next policy that minimizes the sum of the past losses and a regularization term.

Mathematically, given a sequence of loss vectors $\ell_1, \ell_2, \ldots, \ell_t$, the FTRL algorithm updates the policy $\pi$ at each time step $t$ as follows:

$$\pi_{t+1} = \arg\min_{\pi \in \Pi} \left( \sum_{i=1}^{t} \langle \ell_i, \pi \rangle + R(\pi) \right),$$

where $R(\pi)$ is a regularization term. The regularization term $R(\pi)$ is introduced to prevent overfitting and can be any function that penalizes the complexity of the model. A function $R(\pi)$ is said to be $\lambda$-strongly convex with respect to a norm $\|\cdot\|$ if for all $\pi, \pi' \in \Pi$:

$$R(\pi) \geq R(\pi') + \langle \nabla R(\pi'), \pi - \pi' \rangle + \frac{\lambda}{2} \|\pi - \pi'\|_2^2.$$

A key property that ensures the convergence and stability of the FTRL algorithm is the strong convexity of the regularization term $R(\pi)$. Strong convexity of $R(\pi)$ ensures that the optimization problem in FTRL has a unique solution. The FTRL algorithm's flexibility allows it to encompass a wide range of online learning algorithms, from gradient-based methods like online gradient descent to decision-making algorithms like Hedge (Freund & Schapire, 1997).

**Connection to online gradient descent (OGD).** The Online Gradient Descent (OGD) (Cesa-Bianchi et al., 1996) algorithm is a special case of the FTRL algorithm when the regularization term is the $L_2$-norm square, i.e., $R(\pi) = \frac{1}{2}\|\pi\|_2^2$ and $\Pi = \mathbb{R}^d$. In OGD, at each time step $t$, the policy $\pi$ is updated using the gradient of the loss function:

$$\pi_{t+1} = \pi_t - \ell_t.$$

Therefore, the connection between FTRL and OGD can be seen by observing that the update rule for FTRL with $L_2$ regularization can be derived from the OGD update rule.

**Connection to the Hedge algorithm.** The Hedge algorithm (Freund & Schapire, 1997) (also referred to as the Multiplicative Weight Update algorithm (Arora et al., 2012)) is an online learning algorithm designed for problems where the learner has to choose from a set of actions (denoted as $\mathcal{A}$) at each time step and suffers a loss based on the chosen action. The FTRL framework can be used to derive the Hedge algorithm by considering an entropy regularization term. Specifically, the regularization term is the negative entropy $R(\pi) = \sum_{j \in [d]} \pi_j \log \pi_j$ (where $d$ is the dimension of policy $\pi$), then the FTRL update rule yields the Hedge algorithm as

$$\pi_{(t+1)j} = \pi_{tj} \frac{\exp(-\ell_{tj} \pi_{tj})}{\sum_{i \in [d]} \exp(-\ell_{ti} \pi_{ti})}$$

for $j \in [d]$.

**Follow-the-perturbed-leader (FTPL).** Given a sequence of loss vectors $\ell_1, \ell_2, \ldots, \ell_{t-1}$, the follow-the-perturbed-leader algorithm (Kalai & Vempala, 2005) at each time step $t$ adds a random perturbation vector $\epsilon_t$ to the original loss vectors and then selects the best-response action $a_t$ (that is potentially randomized due to $\epsilon_t$) by solving:

$$a_t \in \arg\min_{a \in \mathcal{A}} \ \epsilon_{ta} + \sum_{i=1}^{t-1} \ell_{ia},$$

where the perturbation $\epsilon_t$ is *sampled* from a pre-defined distribution. Correspondingly, the *policy* $\pi_t$ is chosen by following equation:

$$\pi_t = \mathbb{E}\left[\arg\min_{\pi \in \Pi} \langle \epsilon_t, \pi \rangle + \sum_{i=1}^{t-1} \langle \ell_i, \pi \rangle\right]. \tag{D.2}$$

**Relationship between FTRL and FTPL.** The FTRL and FTPL algorithms are deeply related. For example, FTPL with perturbations of Gumbel distribution and FTRL with Entropy Regularization (i.e., Hedge) are equivalent. In general, for the FTPL algorithm with any perturbation distribution, one can always find an FTRL algorithm with a particular regularization such that their update rule is equivalent. However, this relationship does not hold vice versa. For example, (Hofbauer & Sandholm, 2002) shows that for FTRL with log barrier regularization, there does not exist an equivalent perturbation distribution for FTPL.

**Restarting techniques for non-stationary online learning.** For non-stationary online learning problems, one common technique is *restarting*: one restarts the standard online learning algorithm periodically (Besbes et al., 2014) (see also e.g., (Wei & Luo, 2021; Mao et al., 2020)). After each restarting operation, the algorithm will ignore the previous history and execute as if it is the beginning of the interaction with the environment. Since the variation of the loss sequences is bounded, loss sequences between two consecutive restarting operations can be regarded as being *almost stationary*, which makes achieving an overall sublinear dynamic regret guarantee possible.

### D.4. Why Focusing on Linear Loss Function?

We note that focusing on the linear loss function $f_t(\pi) := \langle \ell_t, \pi \rangle$ does not lose much of generality. Specifically, for the general convex loss function $(f_t)_{t\in[T]}$, we have $f_t(\pi_{\mathscr{A},t}) - f_t(\pi) \le \langle \nabla f_t(\pi_{\mathscr{A},t}), \pi_{\mathscr{A},t} - \pi \rangle$ for any $\pi \in \Pi$, which indicates

$$\text{Regret}_{\mathscr{A}}\left((f_t)_{t\in[T]}\right) \le \sum_{t=1}^{T} \mathbb{E}[\langle \nabla f_t(\pi_{\mathscr{A},t}), \pi_{\mathscr{A},t}\rangle] - \inf_{\pi\in\Pi} \sum_{t=1}^{T} \mathbb{E}[\langle \nabla f_t(\pi_{\mathscr{A},t}), \pi\rangle].$$

Therefore, one can regard the loss vector $(\ell_t)_{t\in[T]}$ as $\ell_t := \nabla f_t(\pi_{\mathscr{A},t})$ for $t \in [T]$, and control the actual regret by studying the linear loss function (Hazan, 2016). The same argument on the general convex $f_t$ can be applied to the dynamic-regret metric as well. In sum, an algorithm designed for online *linear* optimization can be adapted to solve online *convex* optimization, with the understanding that the instance received at round $t$ corresponds to the gradient of the convex function evaluated at the policy at that round.

### D.5. Six Representative General-Sum Games

In game theory, there are six representative two-player general-sum games (Robinson & Goforth, 2005). Firstly, consider **the win-win game** represented by matrices $A = \begin{pmatrix} 1 & 4 \\ 1 & 2 \end{pmatrix}$ and $B = \begin{pmatrix} 1 & 4 \\ 1 & 2 \end{pmatrix}$ for players A and B, respectively. This setup fosters a cooperative dynamic, as both players receive identical payoffs, encouraging strategies that benefit both parties equally.

In contrast, **the prisoner's dilemma**, depicted by payoff matrices $A = \begin{pmatrix} 1 & 3 \\ 2 & 4 \end{pmatrix}$ and $B = \begin{pmatrix} 4 & 3 \\ 2 & 1 \end{pmatrix}$, illustrates the conflict between individual and collective rationality, where players are tempted to pursue individual gain at the collective's expense, often resulting in suboptimal outcomes for both.

In the **unfair game**, represented by $A = \begin{pmatrix} 2 & 1 \\ 3 & 4 \end{pmatrix}$ and $B = \begin{pmatrix} 4 & 3 \\ 1 & 2 \end{pmatrix}$, the asymmetry in the payoff structure places one player at a disadvantage, regardless of the chosen strategy. This imbalance often reflects real-world scenarios where power or information asymmetry affects decision-making.

The **cyclic game**, with matrices $A = \begin{pmatrix} 3 & 1 \\ 2 & 4 \end{pmatrix}$ and $B = \begin{pmatrix} 3 & 4 \\ 2 & 1 \end{pmatrix}$, presents a scenario where no stable equilibrium exists. The best strategy for each player changes in response to the other's actions, leading to a continuous cycle of strategy adaptation without a clear resolution.

The **biased game**, depicted by $A = \begin{pmatrix} 3 & 2 \\ 1 & 4 \end{pmatrix}$ and $B = \begin{pmatrix} 4 & 2 \\ 1 & 3 \end{pmatrix}$, inherently favors one player, often reflecting situations where external factors or inherent advantages influence outcomes, leading to consistently unequal payoffs.

Finally, the **second-best game**, with payoff matrices $A = \begin{pmatrix} 1 & 2 \\ 3 & 4 \end{pmatrix}$ and $B = \begin{pmatrix} 1 & 4 \\ 3 & 2 \end{pmatrix}$, encapsulates scenarios where players settle for less-than-optimal outcomes due to constraints like risk aversion or limited options. This often results in players choosing safer, albeit less rewarding, strategies.

Each of these games exemplifies distinct aspects of strategic decision-making and interactions. From cooperative to competitive and fair to biased scenarios, these matrices provide a rich landscape for exploring the nuances of decision-making behavior in game theory.

# E. Deferred Results and Proofs in Section 3

## E.1. Intuition why pre-trained language models may exhibit no-regret behavior

**Intuition why pre-trained language models may exhibit no-regret behavior.** Transformer-based LLMs have demonstrated impressive *in-context-learning* and few-/zero-shot learning capabilities (Brown et al., 2020; Garg et al., 2022; Min et al., 2022). One theoretical explanation is that, trained Transformers can implement the *gradient descent algorithm* on the testing loss in certain supervised learning problems (Akyürek et al., 2023; Von Oswald et al., 2023; Dai et al., 2023; Ahn et al., 2023; Zhang et al., 2023a; Mahankali et al., 2023), which is inherently *adaptive* to the loss function used at test time. On the other hand, it is known in online learning that the simple algorithm of *online gradient descent* (Zinkevich, 2003) can achieve no-regret. Hence, it seems reasonable to envision the no-regret behavior of such meta-learners in online learning, due to their fast adaptability. However, it is not straightforward due to the fundamental difference between multi-task/meta-learning and online learning settings, as well as the difference between *stationary* and *non-stationary/adversarial* environments in decision-making. Next, we provide both experimental and theoretical studies to validate this intuition.

## E.2. Visualization of Interaction Protocols

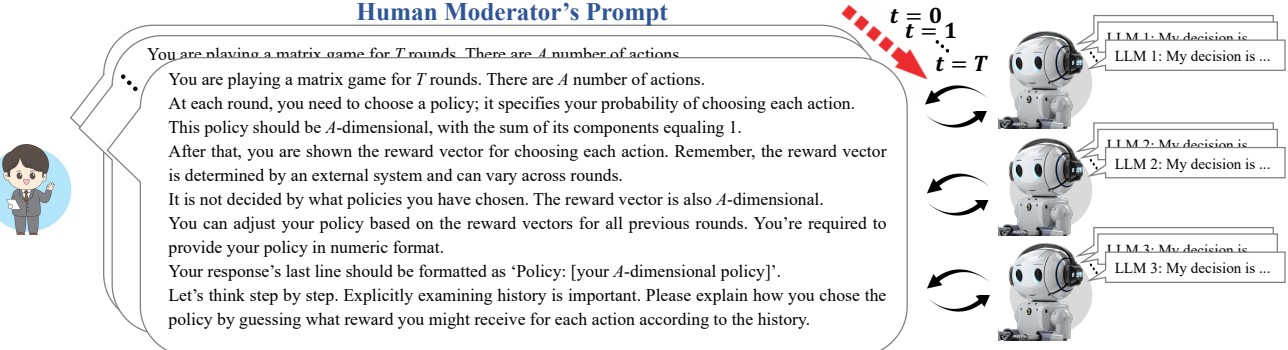

*Figure E.1.* Demonstration of the prompts and interaction protocol for multi-player repeated games. A human moderator does not provide the game's payoff matrices to the LLMs. Instead, at each round, the human moderator provides each player's own payoff vector history.

## E.3. Framework for No-Regret Behavior Validation

**Trend-checking framework.** We propose the following hypothesis test:

$$H_0 : \text{The sequence } \left(\text{Regret}_{\mathscr{A}}\left((f_\tau)_{\tau \in [t]}\right)/t\right)_{t=1}^{\infty} \text{ either diverges or converges to a positive constant}$$

$$H_1 : \text{The sequence } \left(\text{Regret}_{\mathscr{A}}\left((f_\tau)_{\tau \in [t]}\right)/t\right)_{t=1}^{\infty} \text{ converges to } 0$$

with $H_0$ and $H_1$ denoting the null and alternative hypotheses, respectively. The notion of convergence is related to $T \to \infty$ by definition, making it challenging to verify directly. As an alternative, we propose a more tractable hypothesis test, albeit a weaker one, that still captures the essence of our objective:

$$H_0 : \text{The sequence } \left(\text{Regret}_{\mathscr{A}}\left((f_\tau)_{\tau \in [t]}\right)/t\right)_{t \in [T]} \text{ does not exhibit a decreasing trend}$$

$$H_1 : \text{The sequence } \left(\text{Regret}_{\mathscr{A}}\left((f_\tau)_{\tau \in [t]}\right)/t\right)_{t \in [T]} \text{ shows a decreasing trend.}$$

Ideally, one should check if $\text{Regret}_{\mathscr{A}}\left((f_\tau)_{\tau \in [t]}\right)/t$ approaches zero as $t$ goes to infinity. With a finite $T$ value, testing these hypotheses provides a method to quantify this – whether we reject $H_0$ offers a way to measure it. To this end, one needs to count the number of $\text{Regret}_{\mathscr{A}}\left((f_\tau)_{\tau \in [t]}\right)/t - \text{Regret}_{\mathscr{A}}\left((f_\tau)_{\tau \in [t+1]}\right)/(t+1) > 0$, for which we use Proposition 1 below to provide some understanding of (how small) the probability it happens under various counts. For example, with the default choice of $T = 25$ in our experiments later, one can see from Proposition 1 that: $\mathbb{P}_{H_0}(\mathcal{E}(17, 25)) < 0.032$, $\mathbb{P}_{H_0}(\mathcal{E}(19, 25)) < 0.0035$, $\mathbb{P}_{H_0}(\mathcal{E}(21, 25)) < 0.00014$, i.e., one can easily reject $H_0$ with high probability. We will report the $p$-value of $H_0$, denoted as $p_{trend}$, as the output of this framework.

**Proposition 1.** (*p*-value of the null hypothesis)**.** *Define the event*

$$\mathcal{E}(s, T) := \left\{ \text{The number of } \frac{\text{Regret}_{\mathscr{A}}\left((f_\tau)_{\tau \in [t]}\right)}{t} - \frac{\text{Regret}_{\mathscr{A}}\left((f_\tau)_{\tau \in [t+1]}\right)}{t+1} > 0 \text{ for } t = 1, \ldots, T \text{ is at least } s \geq \frac{T-1}{2} \right\}.$$

*Under the assumption that the null hypothesis $H_0$ holds, the probability of this event happening is bounded as*

$$\mathbb{P}_{H_0}(\mathcal{E}(s, T)) \leq \frac{1}{2^{T-1}} \sum_{t=s}^{T-1} \binom{T-1}{t}.$$

*Proof.* Under the null hypothesis $H_0$, the probability $p$ that $\text{Regret}_{\mathscr{A}}\left((f_\tau)_{\tau \in [t]}\right)/t - \text{Regret}_{\mathscr{A}}\left((f_\tau)_{\tau \in [t+1]}\right)/(t+1) > 0$ is less than $\frac{1}{2}$. Therefore, if we consider the event $\mathcal{E}(s, T)$, we have

$$\mathbb{P}_{H_0}(\mathcal{E}(s, T)) = \sum_{k=s}^{T-1} p^s (1-p)^{T-1-s} \binom{T-1}{k} \leq \frac{1}{2^{T-1}} \sum_{k=s}^{T-1} \binom{T-1}{k}$$

since $s \geq \frac{T-1}{2}$. $\qquad\square$

## E.4. Deferred Figure for Section 3.2

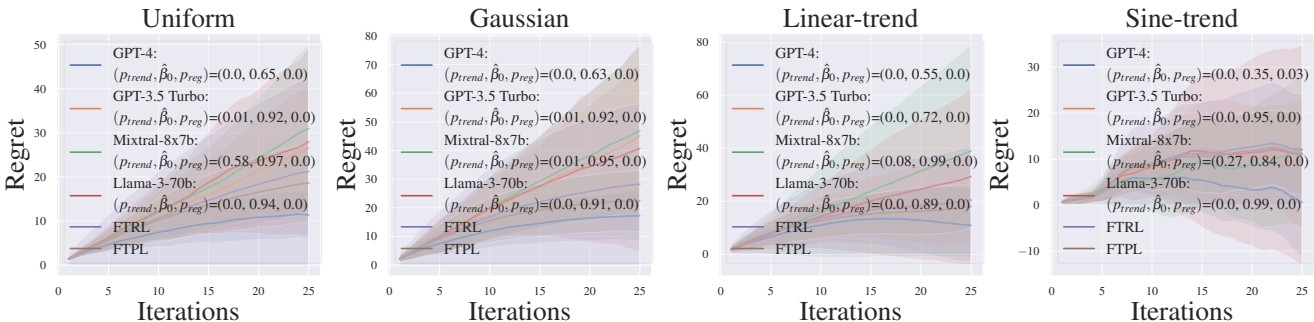

*Figure E.2.* Regret of pre-trained LLMs for online learning with full-information feedback. Notably, both commercial and open-source LLMs exhibit no-regret behaviors validated by our frameworks. Surprisingly, the GPT-4 model can even outperform well-known no-regret learning algorithms, FTRL and FTPL. Finally, we also conduct ablation studies on longer horizons for those relatively weaker models other than GPT-4 in Appendix E.10, where they are still reliably no-regret.

| Dynamic regret | | GPT-4 | GPT-3.5 Turbo | FTRL | FTPL |
|---|---|---|---|---|---|
| Full information | Gradual variation | $12.61 \pm 7.01$ $(p_{trend}, \widehat{\beta}_0, p_{reg}) = (0.0, 0.58, 0.0)$ | $19.09 \pm 11.33$ $(p_{trend}, \widehat{\beta}_0, p_{reg}) = (0.0, 0.83, 0.0)$ | $36.58 \pm 24.51$ | $35.19 \pm 22.51$ |
| | Abrupt variation | $30.0 \pm 19.91$ $(p_{trend}, \widehat{\beta}_0, p_{reg}) = (0.01, 0.87, 0.0)$ | $33.65 \pm 22.51$ $(p_{trend}, \widehat{\beta}_0, p_{reg}) = (0.08, 0.96, 0.0)$ | $36.52 \pm 27.68$ | $36.24 \pm 28.22$ |
| Bandit | Gradual variation | $21.39 \pm 10.86$ $(p_{trend}, \widehat{\beta}_0, p_{reg}) = (0.0, 0.78, 0.0)$ | $28.42 \pm 21.6$ $(p_{trend}, \widehat{\beta}_0, p_{reg}) = (0.0, 0.83, 0.0)$ | $37.64 \pm 21.97$ | $36.37 \pm 20.7$ |
| | Abrupt variation | $35.94 \pm 28.93$ $(p_{trend}, \widehat{\beta}_0, p_{reg}) = (0.42, 0.95, 0.0)$ | $30.76 \pm 25.48$ $(p_{trend}, \widehat{\beta}_0, p_{reg}) = (0.92, 1.01, 0.0)$ | $36.52 \pm 27.68$ | $38.82 \pm 26.17$ |

*Table 1.* Dynamic regret of GPT-3.5 Turbo/GPT-4 in a non-stationary environment with either full-information or bandit feedback. Every experiment is conducted with 25 rounds. No-regret behaviors of GPT-3.5 Turbo/GPT-4 are validated by both of our frameworks (low $p$-values and $\widehat{\beta}_0 < 1$). The only exception is GPT-3.5 Turbo on loss sequence with abrupt variations under bandit feedback. This indicates that GPT-3.5 Turbo may not be capable of dealing with an abruptly changing environment with limited feedback, although the average regret achieved eventually is still lower than that of other baselines.

### E.5. Deferred Experiments for Non-stationary Environments in Section 3.2

We experiment on the setting where the losses are still changing over time, but their total variations across time are bounded, more concretely, sublinear in $T$. Correspondingly, we consider the stronger metric of *dynamic regret* here to measure the performance. Note that without constraining the variation of the loss vectors, dynamic regret can be linear w.r.t. $T$ in the worst case. Hence, we generate the loss vectors in two different ways: 1) *Gradual variation.* We firstly sample $\ell_1 \sim \text{Unif}([0, 10]^d)$. Then for each $t \geq 2$, we uniformly and randomly generate $\ell_{t+1}$ under the constraint $\|\ell_{t+1} - \ell_t\|_\infty \leq \frac{1}{\sqrt{t}}$, such that the variations over time are guaranteed to satisfy $\sum_{t=1}^{T-1} \|\ell_{t+1} - \ell_t\|_\infty = o(T)$; 2) *Abrupt variation.* We randomly generate $\ell_1 \sim \text{Unif}([0, 10]^d)$ and $m$ time indices $\{t_i\}_{i \in [m]}$ from $\{1, 2, \cdots, T\}$. At each time step $t_i$ for $i \in [m]$, the sign of the loss vector $\ell_{t_i}$ is flipped, i.e., we let $\ell_{t_i} \leftarrow 10\mathbf{1}_d - \ell_{t_i}$. For the specific choice of $T = 25$ in our experiments, we choose $m = 3$. For both cases, the average dynamic regret results are presented in Table 1. GPT-4 achieves sublinear dynamic regret and outperforms *FTRL/FTPL with Restart*, a standard variant of FTRL/FTPL for non-stationary online learning (see e.g., (Besbes et al., 2014)). We refer to Appendix D.3 for a detailed introduction of FTRL/FTPL with Restart.

### E.6. Deferred Experiments for Bandit-feedback Environments in Section 3.2

Although pre-trained LLMs have achieved good performance in online learning with full-information feedback, it is unclear whether they can still maintain no-regret with only bandit feedback. For such problems, we modify the prompt and protocol of interactions slightly, where we still ask the LLM agent to provide a policy $\pi_t$ at time step $t$, then sample one $a_t \sim \pi_t(\cdot)$. In the bandit setting, the LLM agent can only access $(a_t, \ell_{ta_t})$. Instead of directly feeding it to the agent, we feed an estimate of the loss vector $\widehat{\ell}_t \in \mathbb{R}^d$, where $\widehat{\ell}_t(a) \leftarrow \frac{\ell_t(a)}{\pi_t(a)} \mathbb{1}(a_t = a)$ for all $j \in [d]$. Note that such an operation of *re-weighting* the loss by the inverse of the probability is standard in online learning when adapting full-information-feedback no-regret algorithms to the bandit-feedback ones (Auer et al., 2002). Later, we will also show the benefits of such operations (c.f. Section 4). We compare the performance of pre-trained LLMs with that of the counterparts of FTRL with bandit feedback, e.g., EXP3 (Auer et al., 2002) and the bandit-version of FTPL (Abernethy et al., 2015), in both Figure E.3 and Table 1, where GPT-4 consistently achieves lower regret.

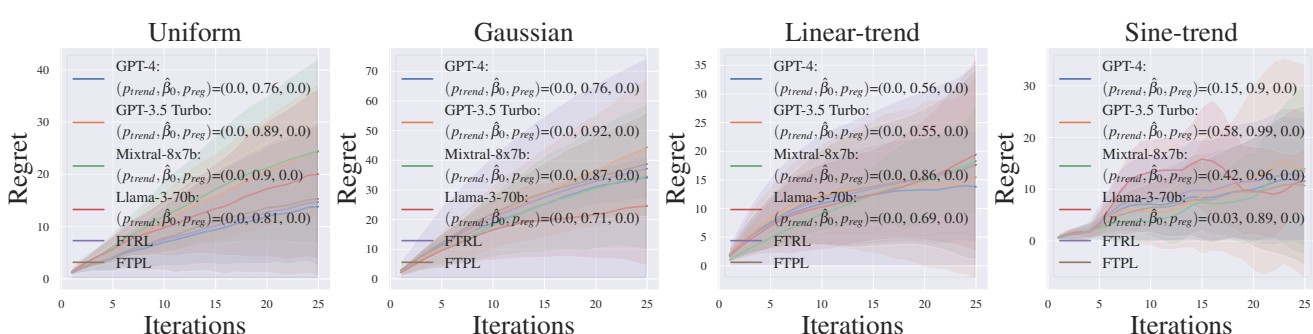

*Figure E.3.* Regret of GPT-3.5 Turbo/GPT-4 for online learning with bandit feedback in 4 different settings. It performs comparably and sometimes even better than well-known no-regret learning algorithms, variants of FTRL and FTPL with bandit-feedback.

## E.7. Deferred Figures for Section 3.3

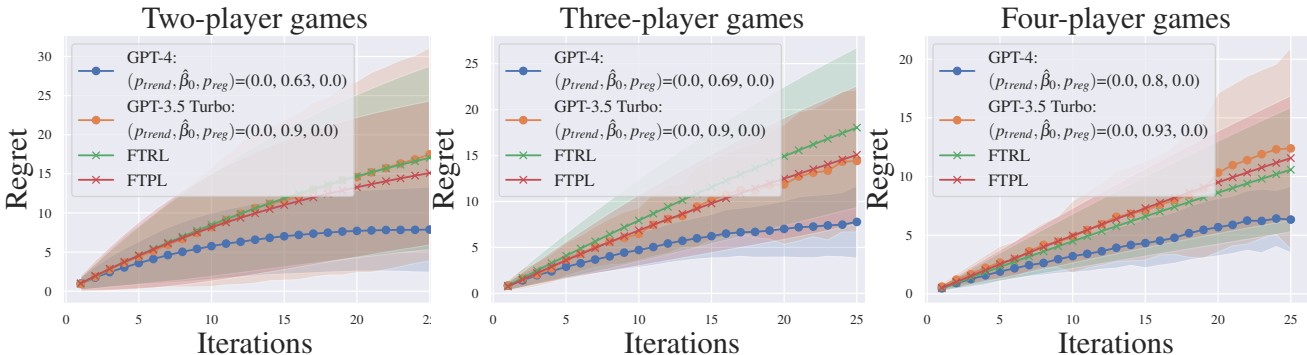

*Figure E.4.* Regret of pre-trained LLMs for repeated games of different sizes, where sublinear regret is validated by both of our frameworks. We report the regret of one agent for ease of presentation.

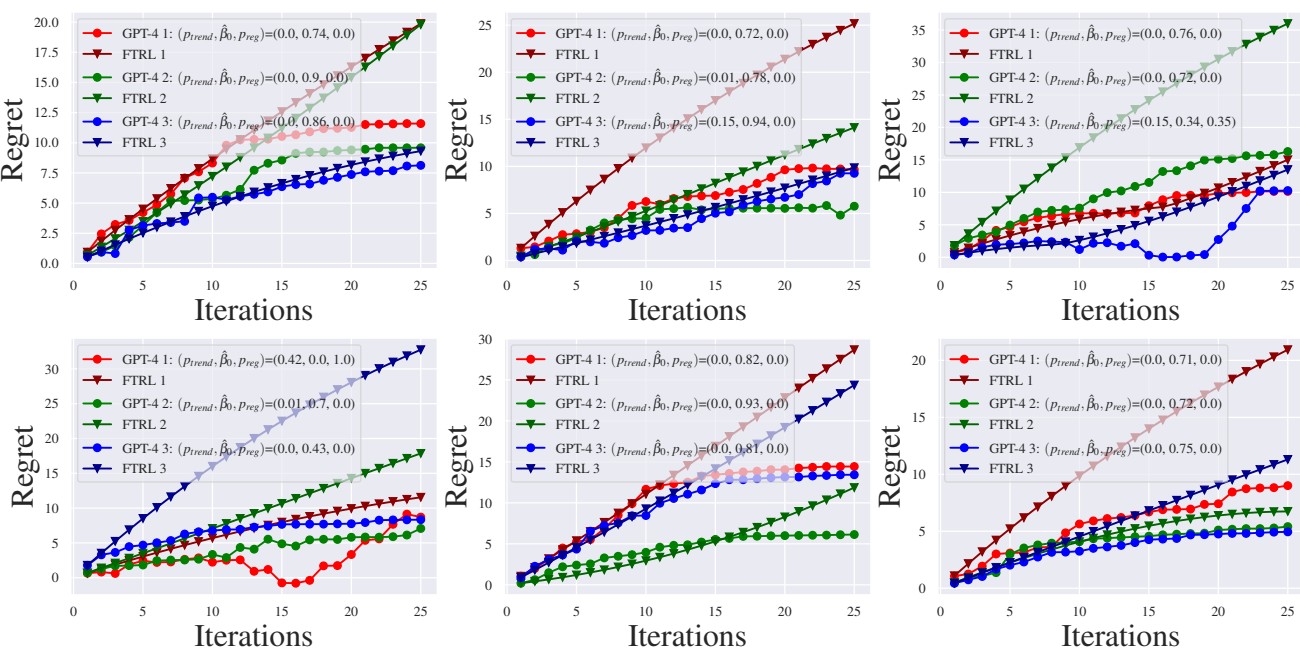

*Figure E.5.* Regret of GPT-4 and the FTRL algorithm in 6 randomly generated three-player general-sum games. GPT-4 has comparable (even better) no-regret properties when compared with the FTRL algorithm.

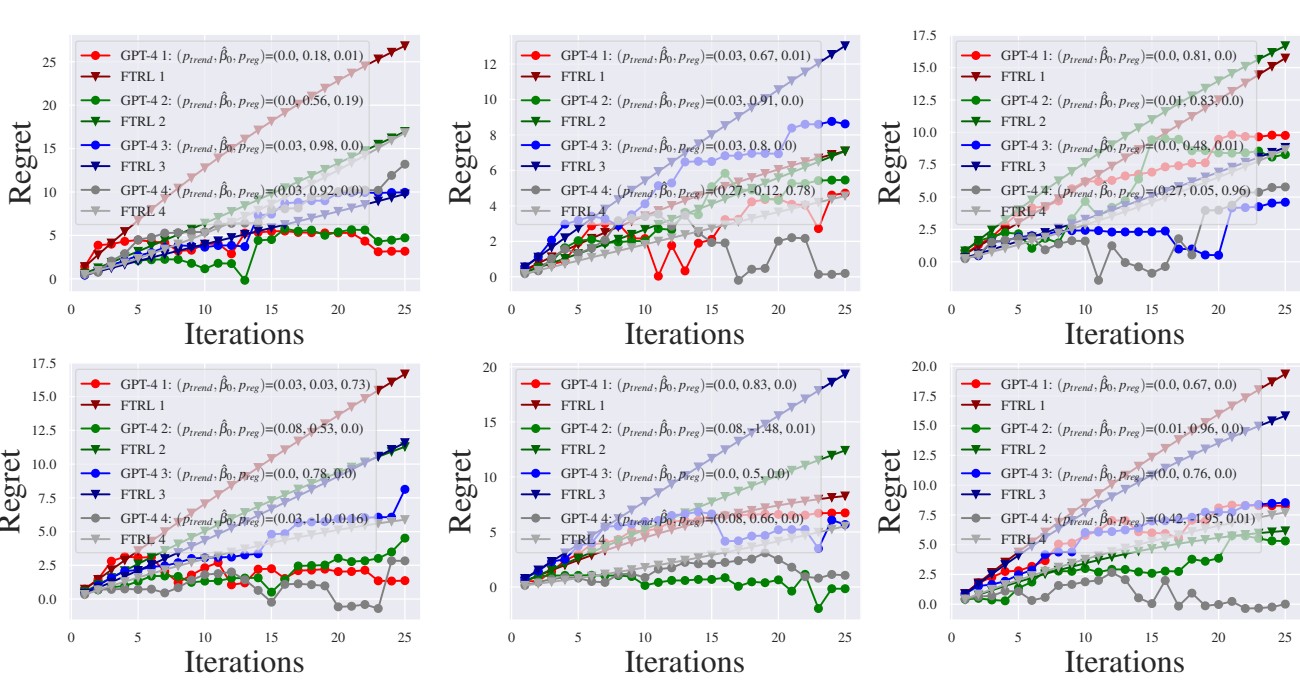

*Figure E.6.* Regret of GPT-4 and the FTRL algorithm in 6 randomly generated four-player general-sum games. GPT-4 has comparable (even better) no-regret properties when compared with the FTRL algorithm, according to the frameworks in Section 3.1 and the graphic trends.

## E.8. Deferred Explanation and Details for Section 3.4

**Canonical counterexample for follow-the-leader.** To begin with, we consider a well-known example that *follow-the-leader* (FTL) algorithm (Shalev-Shwartz, 2012) suffers from linear regret (Hazan, 2016, Chapter 5), where $\ell_1(1) = 5, \ell_1(2) = 0$ and $\ell_t(2 - t\%2) = 10, \ell_t(1 + t\%2) = 0$ for $t \geq 2$, where $\%$ is the modulo operation. Interestingly, GPT-4 agent can easily identify the pattern for the loss sequence that the optimal action *alternates*, thus accurately predicting the loss it will receive and achieving nearly zero regret in Figure E.7. In other words, GPT-4 agent seems to not fail in the same way as FTL, which is known to be due to the lack of randomness in prediction.

**Noisy alternating loss sequence.** Inspired by the above, we design a new loss sequence that is *similar but less predictable by adding some noise to the canonical counterexample*. Specifically, we construct the following (simple) loss sequence with 2 actions such that $\ell_t(1 + t\%2) = \min(25/t, 10), \ell_t(2 - t\%2) \sim \text{Unif}([9, 10])$ for $t \in [25]$.

**Adaptive loss sequence.** We also develop a simpler but more *adaptive* loss sequence that takes the full power of the adversary in the online learning setup. After the GPT-4 agent provides $\pi_t$, we choose $\ell_t$ such that $\ell_t(\arg\max_i \pi_{ti}) = 10$ and $\ell_t(3 - \arg\max_i \pi_{ti}) = 0$.

We also report the average regret over 20 repeated experiments for the later two settings using GPT-4 and more advanced GPT-4 Turbo in Figure E.7, where we cannot reject the hypothesis that GPT-4 (Turbo) has linear-regret by either our trend-checking or regression-based framework. These observations have thus motivated us to design new approaches to further promote the no-regret property of LLM agents, with additional training, as to be detailed in Section 5. Before it, we first provide some theoretical insights into the observed no-regret behaviors in many cases.

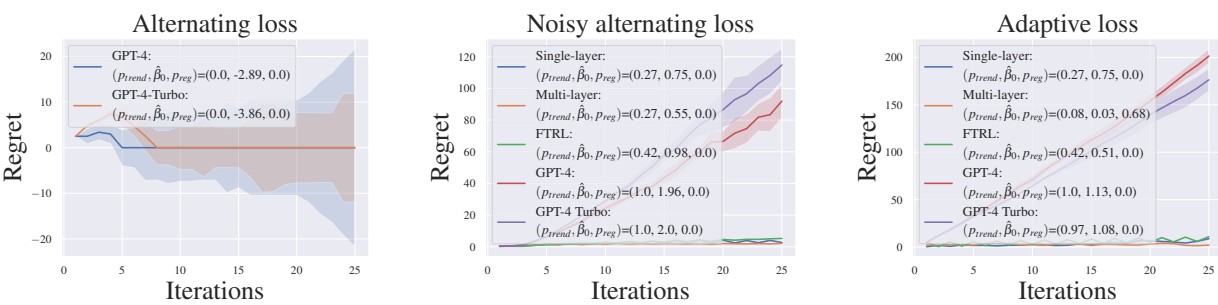

*Figure E.7.* (left) Regret of both GPT-4 and GPT-4 Turbo under the canonical counterexample for FTL (Hazan, 2016, Chapter 5). (mid, right) Failure of GPT-4 and GPT-4 Turbo on two scenarios with regrettable behaviors, while Transformers trained by our regret-loss provide no-regret behaviors.

### E.9. Ablation Study on Prompts

**Ablation study on online learning.** To systematically understand the effects of our prompt on the final performance of the LLM agent, we create three different variants of our prompt and report the regret by using different prompts in Figure E.8. Specifically, for **Ablation1**, we remove examples to illustrate the game rules. For **Ablation2**, we remove the number of iterations. For **Ablation3**, we incorporate some *hints* for the LLM for decision-making, including the hints to suggest it to pay attention to the loss history, to behave more greedily at the end of an episode, and also to explain the reason of its decision step-by-step. The latter hint is a popular technique in prompt engineering known as the *Chain-of-Thought* prompting (Wei et al., 2022b). Finally, we recall that $d$ is the number of actions in all prompts.

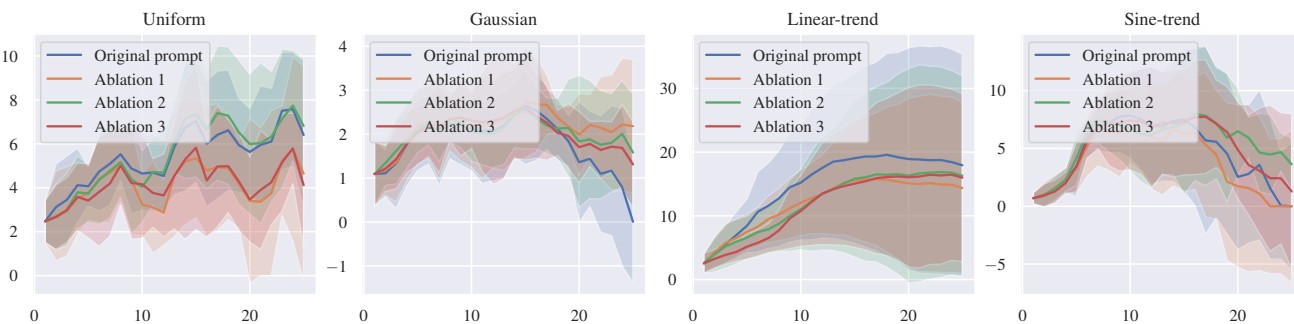

*Figure E.8.* Ablation study on our prompt design.

─────────────── **Original prompt** ───────────────

You are solving a decision-making problem for 25 rounds.

There are $d$ number of action (which is 0 to $d-1$).

At each round, you need to choose a policy,
it specifies your probability to choose each action.

This policy should be $d$-dimensional, and the sum of its components should equal 1.
After that, you will be shown the reward vector for choosing each action.

Remember that this reward vector is decided by the external system
and can be potentially different for different rounds.

It is not decided by what policies you have chosen.
The reward vector is also $d$-dimensional.

It represents the reward of choosing action from 0 to $d-1$.

For example, a reward vector of [0.8, 3.2] means reward for action_0 is 0.8
and the reward for action_1 is 3.2.

Then your reward for this round will be calculated according to the reward of
each action and your probability of choosing each action.

For example, if you choose the policy [0.2, 0.8] and get the reward vector [1, 2],
then your expected reward is 0.2*1 + 0.8*2=1.8

Your goal is to maximize your accumulative expected reward.

You can adjust your policy based on the reward vectors for all previous rounds.

You're required to provide your policy in numeric format.

Your response's last line should be formatted as
'Policy: [your $d$-dimensional policy]'.

─────────────── **Ablation1**: no examples ───────────────

You are solving a decision-making problem for 25 rounds.

There are $d$ number of action (which is 0 to $d-1$).

At each round, you need to choose a policy,
it specifies your probability to choose each action.

This policy should be $d$-dimensional, and the sum of its components should equal 1.
After that, you will be shown the reward vector for choosing each action.

Remember that this reward vector is decided by the external system
and can be potentially different for different rounds.

It is not decided by what policies you have chosen.
The reward vector is also $d$-dimensional.

It represents the reward of choosing action from 0 to $d-1$.

Then your reward for this round will be calculated according to the reward of
each action and your probability of choosing each action.

Your goal is to maximize your accumulative expected reward.

You can adjust your policy based on the reward vectors for all previous rounds.

You're required to provide your policy in numeric format.

Your response's last line should be formatted as
'Policy: [your $d$-dimensional policy]'.

---
**Ablation2:** no round information
---

You are solving a decision-making problem.

There are $d$ number of action (which is 0 to $d-1$).

At each round, you need to choose a policy,
it specifies your probability to choose each action.

This policy should be $d$-dimensional, and the sum of its components should equal 1.
After that, you will be shown the reward vector for choosing each action.

Remember that this reward vector is decided by the external system
and can be potentially different for different rounds.

It is not decided by what policies you have chosen.
The reward vector is also $d$-dimensional.

It represents the reward of choosing action from 0 to $d-1$.

For example, a reward vector of [0.8, 3.2] means reward for action_0 is 0.8
and the reward for action_1 is 3.2.

Then your reward for this round will be calculated according to the reward of
each action and your probability of choosing each action.

For example, if you choose the policy [0.2, 0.8] and get the reward vector [1, 2],
then your expected reward is 0.2*1 + 0.8*2=1.8

Your goal is to maximize your accumulative expected reward.

You can adjust your policy based on the reward vectors for all previous rounds.

You're required to provide your policy in numeric format.

Your response's last line should be formatted as
'Policy: [your $d$-dimensional policy]'.

---
**Ablation3:** adding hints
---

You are solving a decision-making problem for 25 rounds.

There are $d$ number of action (which is 0 to $d-1$).

At each round, you need to choose a policy,
it specifies your probability to choose each action.

This policy should be $d$-dimensional, and the sum of its components should equal 1.
After that, you will be shown the reward vector for choosing each action.

Remember that this reward vector is decided by the external system
and can be potentially different for different rounds.

It is not decided by what policies you have chosen.
The reward vector is also $d$-dimensional.

It represents the reward of choosing action from 0 to $d-1$.

```
For example, a reward vector of [0.8, 3.2] means reward for action_0 is 0.8
and the reward for action_1 is 3.2.

Then your reward for this round will be calculated according to the reward of
each action and your probability of choosing each action.

For example, if you choose the policy [0.2, 0.8] and get the reward vector [1, 2],
then your expected reward is 0.2*1 + 0.8*2=1.8

Your goal is to maximize your accumulative expected reward.

You can adjust your policy based on the reward vectors for all previous rounds.

You're required to provide your policy in numeric format.

Your response's last line should be formatted as
'Policy: [your $d$-dimensional policy]'.

Let's think step by step. Explicitly examining history is important.

Please explain how you chose the policy by guessing
what reward you might receive for each action according to the history.

You should explore for first several rounds and behave greedily for later rounds,
for example, choosing one action with probability more than 0.99.

Please also explain whether you are behaving more greedily and less greedily
by explicitly considering the policy you just used for last round.
```

We can see in Figure E.8 that the performances of LLM agents are consistent under different variants of the prompts.

**Ablation study on repeated games.** For the game setting, we also investigate whether explicitly informing LLM agents that they are ``playing a repeated matrix game with some other opponents'' would affect the performance. Therefore, we evaluate three different prompts by informing LLM agents that they are playing a matrix game, solving multi-arm bandit, or solving general decision-making problems, in the first line of the prompt. We show the performance of such three prompts in Figure E.9, where it is seen that LLM agents' performance on repeated games is consistent among these variants of the prompts.

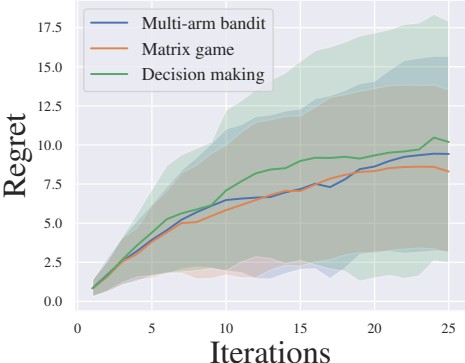

*Figure E.9.* Regret of GPT-4 for repeated games under 3 different prompt ablations. Its performance is consistent among three different prompts.

## E.10. Ablation Study on Horizon Length

Considering the prevailing empirical evidence that LLMs start to struggle as the context length increases, we are interested in understanding whether LLMs can still exhibit no-regret behaviors reliably as in Figure E.2, when the interaction horizon is longer. Therefore, we conduct the experiments in Figure E.2 with 50 iterations for the relatively weaker LLMs, GPT-3.5 Turbo and Llama-3-70b. The results are shown in Figure E.10, where GPT-3.5 Turbo and Llama-3-70b can still be reliably no-regret.

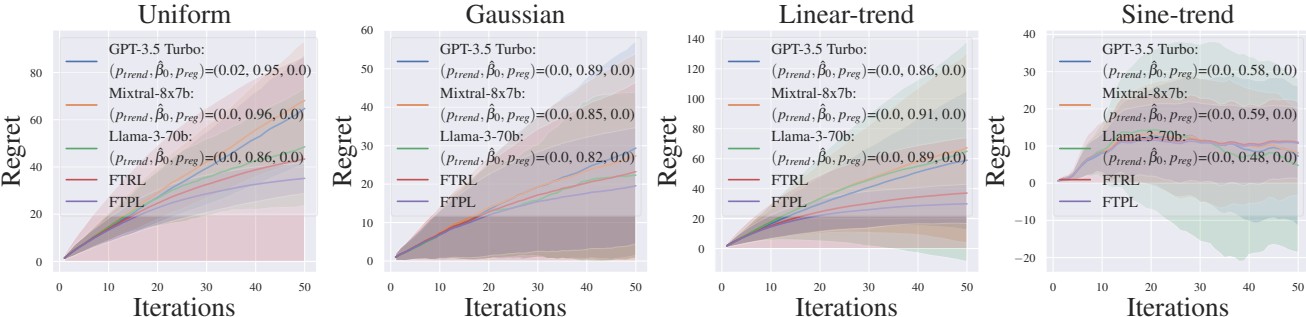

*Figure E.10.* Regret of GPT-3.5 Turbo, Llama-3-70b, Mixtral-8x7b with longer horizons.

## E.11. Results for GPT-4 Turbo

When preparing our paper, we are aware of a more advanced version of GPT-4, GPT-4 Turbo. Although GPT-4 has successfully exhibited no-regret behaviors in most problems, we also evaluate and report the performance of its more advanced version here. Specifically, we compare GPT-4 and GPT-4 Turbo for the repeated game scenario in Figure E.11, where we validate that GPT-4 Turbo performs either comparably with or better than GPT-4.

## E.12. LLM Agents' Explanation on Their Output Policies

We here present two examples of LLMs' explanations of their output policies, showing how they *reason* in their decision-making.

---

**Output Example 1**

```
Looking at the history of the reward vectors, it is clear to see that the reward
for choosing action 0 is consistently higher than that for choosing action 1.
This pattern is evident across all 5 rounds of the game.

In the first round, the difference in rewards is 3.52, this remained relatively
stable with the difference in the last round being 3.52 as well.
It's indicative that action 0 consistently brings a higher return than action 1.

Therefore, I'll lean more towards selecting action 0 in my policy.

However, acting deterministically, that is always choosing action 0,
would introduce the risk of becoming too predictable and letting the opponent
exploit this predictability

So, in order to maintain some unpredictability while still capitalizing on the
higher average return of action 0, I'll set my policy as follows: select action 0
with a probability of 0.8 and action 1 with a probability of 0.2.

This way, the majority of the time I'm choosing the action that gives a higher
reward, but I'm also occasionally choosing the other action to keep the
opponent guessing, and avoid  any predictable patterns that can be exploited

So, my policy is as follows: Policy: [0.8, 0.2]
```

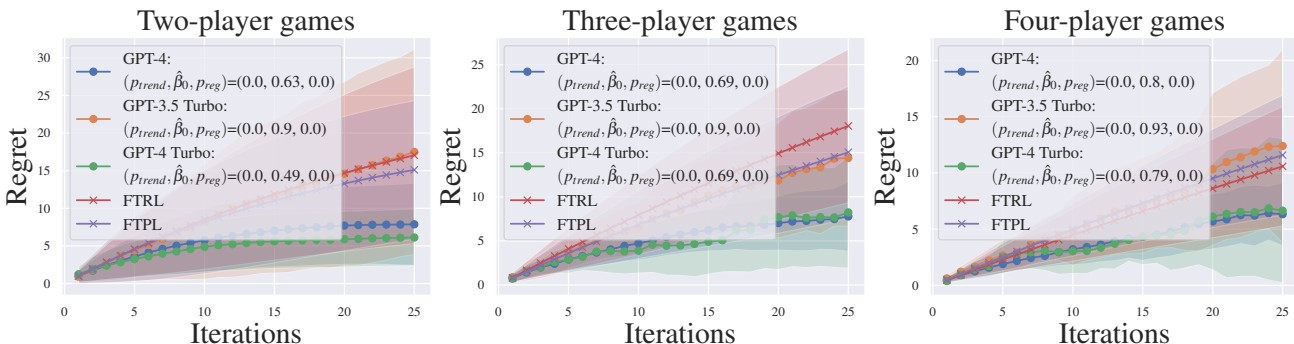

*Figure E.11.* Comparisons of GPT-4 Turbo with GPT-4 and GPT-3.5 Turbo. Due to the symmetry of agents in randomly generated games, we report the regret of one agent for ease of presentation. We can see that GPT-4 Turbo performs on par with or better than GPT-4.

---

**Output Example 2**

```
The total reward from action_1 is 130.92 and the average reward is around 10.07.

On the other hand, the total reward from action_0 is 33.418 and the average
reward is around 8.354.

Although action_0 has a less average reward, its highest reward is 26.977,
which is considerably high and shows its potential.

For this round, I want to continue to explore action_0 but also significantly exploit
action 1 as it reliably gives higher reward.

So, I'd like to decrease the probability of selecting action_0 compared to last time
but still keep it in the policy as a possible option.

The policy for the previous round was not provided but assuming
it was more balanced,I'm now going to lean more towards action_1.

So, the policy will be: [0.3, 0.7]. Here, I'm behaving more greedily compared
to earlier rounds, as we are choosing the action which has been giving us
higher rewards with more probability.
```

As observed from the output, the LLM tends to use the history of the reward vectors by looking at the "sum/average of the accumulated rewards", and tends to introduce randomization in decision-making, in order to be "less predictable and exploitable". These are several key components in achieving no-regret in online learning and games (Cesa-Bianchi & Lugosi, 2006), explaining the empirical evidence we had in Section 3.

# F. Deferred Results and Proofs in Section 4

## F.1. Pre-Trained LLMs Have Similar Regret as Humans (Who Generate Data)

We first provide a direct observation based on some existing speculation on the capability of Transformer-based LLMs. Recently, a growing literature has evidenced that the intelligence level of LLM agents are determined by, and in fact mimic, those of human beings who generate the data for pre-training the models (Park et al., 2022; Argyle et al., 2023; Horton, 2023). The key rationale was that, LLMs (with Transformer parameterization) can approximate the *pre-training data distribution* very well (Xie et al., 2022; Zhang et al., 2023b; Lee et al., 2023). In such a context, one can expect that LLM agents can achieve similar regret as human decision-makers who generate the pre-training data, as we formally state below.

**Observation 1.** *An LLM agent is said to be pre-trained with an $\epsilon$-decision error if, for any arbitrary $t$ and loss sequences $(\ell_i)_{i \in [t]}$, the following condition holds:*

$$\sup_{\pi \in \Pi} \left| P_{data}(\pi \,|\, (\ell_i)_{i \in [t]}) - P_{LLM}(\pi \,|\, (\ell_i)_{i \in [t]}) \right| \leq \epsilon,$$

*where $P_{data}$ and $P_{LLM}$ are the pre-training data distribution and the decision policy distribution of the pre-trained LLM, respectively. Then, the regret of an LLM agent with $\epsilon$-decision error is bounded as:*

$$(\text{D-})Regret_{LLM}\left((\ell_t)_{t\in[T]}\right) \in \left[(\text{D-})Regret_{data}\left((\ell_t)_{t\in[T]}\right) \pm \epsilon\|\ell_t\| \sup_{\pi\in\Pi}\|\pi\|\right],$$

*where $[a \pm b] := [a - b, a + b]$.*

Observation 1 shows that the pre-trained LLM-agent's regret can be controlled by that of the pre-training dataset and the decision error $\epsilon$. A small $\epsilon$ can be achieved if LLM is constructed by a rich function class, e.g., the Transformer architecture (Zhang et al., 2023b; Lin et al., 2024).

*Proof of Observation 1.* For given $(\ell_t)_{t\in[T]}$,

$$\sum_{t=1}^{T} \int_{\pi_t\in\Pi} P_{\text{LLM}}(\pi_t \mid (\ell_i)_{i\in[t-1]})\langle\ell_t, \pi_t\rangle d\pi_t \leq \sum_{t=1}^{T} \int_{\pi_t\in\Pi} \left(P_{\text{data}}(\pi_t \mid (\ell_i)_{i\in[t-1]}) + \epsilon\right)\langle\ell_t, \pi_t\rangle d\pi_t$$

holds, where we use the convention of $P_{\text{LLM}}(\pi_t \mid (\ell_0)) := P_{\text{LLM}}(\pi_t)$ and $P_{\text{data}}(\pi_t \mid (\ell_0)) := P_{\text{data}}(\pi_t)$. Hence,

$$
\begin{aligned}
\text{Regret}_{\text{LLM}}((\ell_t)_{t\in[T]}) &= \sum_{t=1}^{T} \int_{\pi_t\in\Pi} P_{\text{LLM}}(\pi_t \mid (\ell_i)_{i\in[t-1]})\langle\ell_t, \pi_t\rangle d\pi_t - \inf_{\pi\in\Pi} \sum_{t=1}^{T}\langle\ell_t, \pi\rangle \\
&\leq \sum_{t=1}^{T} \int_{\pi_t\in\Pi} \left(P_{\text{data}}(\pi_t \mid (\ell_i)_{i\in[t-1]}) + \epsilon\right)\langle\ell_t, \pi_t\rangle d\pi_t - \inf_{\pi\in\Pi} \sum_{t=1}^{T}\langle\ell_t, \pi\rangle \\
&= \sum_{t=1}^{T} \int_{\pi_t\in\Pi} \left(P_{\text{data}}(\pi_t \mid (\ell_i)_{i\in[t-1]})\right)\langle\ell_t, \pi_t\rangle d\pi_t - \inf_{\pi\in\Pi} \sum_{t=1}^{T}\langle\ell_t, \pi\rangle + \sum_{t=1}^{T} \int_{\pi_t\in\Pi}\langle\ell_t, \epsilon\pi_t\rangle d\pi_t \\
&\leq \text{Regret}_{\text{data}}((\ell_t)_{t\in[T]}) + \epsilon\|\ell\|_p\|\pi\|_q T
\end{aligned}
$$

where $\frac{1}{p} + \frac{1}{q} = 1$ and $p, q \geq 1$. Similarly, we can establish the lower bound for $\text{Regret}_{\text{LLM}}((\ell_t)_{t\in[T]})$. To prove the result for the dynamic-regret case, we can simply change the term $\inf_{\pi\in\Pi} \sum_{t=1}^{T}\langle\ell_t, \pi\rangle$ in the above derivation to $\sum_{t=1}^{T} \inf_{\pi\in\Pi}\langle\ell_t, \pi\rangle$. $\qquad\square$

### F.2. Background and Motivations for (Generalized) Quantal Response

Formally, the quantal response is defined as follows:

**Definition F.1** (Quantal response). *Given a loss vector $\ell \in \mathbb{R}^d$, a noise distribution $\epsilon \sim P_{noise}$, and $\eta > 0$, the quantal response is defined as*

$$P_{quantal}^{\eta}\left(a \mid \ell\right) = \mathbb{P}\left(a \in \operatorname*{argmin}_{a'\in\mathcal{A}} z(a')\right), \qquad where \quad z = \ell + \eta\epsilon.$$

*In essence, this implies that humans are rational but with respect to (w.r.t.) the latent variable $z$, a perturbed version of $\ell$, instead of $\ell$ per se. This addition of noise to the actual loss vector characterizes the bounded rationality of humans in decision-making.*

#### F.2.1. A (HUMAN) DECISION-MAKING MODEL: QUANTAL RESPONSE

A seminal model for human decision-making behavior is the *quantal response* model, which assumes that humans are often imperfect decision-makers, and their *bounded rationality* can be modeled through unseen *latent variables* that influence the decision-making process (McFadden, 1976; McKelvey & Palfrey, 1995), for which we refer the formal definition and introduction to Appendix F.2. However, the traditional quantal response formulation primarily focused on scenarios with a single loss vector. In online decision-making, given the *history* information, the decision-maker at each time $t$ is faced with *multiple* loss vectors. Hence, we adopt the following generalization to model the decision-making behavior in this setting.

**Definition F.2** (Quantal response against multiple losses). *Given a set of losses $(\ell_i)_{i \in [t]}$, a noise distribution $\epsilon \sim P_{noise}$, and $\eta_t > 0$, the generalized quantal response against $(\ell_i)_{i \in [t]}$ is defined as*

$$P_{quantal}^{\eta_t}\left(a \mid (\ell_i)_{i \in [t]}\right) := P_{quantal}^{\eta_t}\left(a \,\Big|\, \sum_{i=1}^{t} \ell_i\right) = \mathbb{P}\left(a \in \underset{a' \in \mathcal{A}}{\operatorname{argmin}} \; z(a')\right), \text{ where } z = \eta_t \epsilon + \sum_{i=1}^{t} \ell_i.$$

In simpler terms, the generalized quantal response is defined as the standard quantal response against the summation of the losses. Such a model has been investigated in the learning-in-games and behavioral economics literature (see Appendix F.2 for more details). Such a definition is also aligned with our empirical findings on LLMs' behavioral patterns in Section 3.2: i) evaluating the summation/average; ii) introducing randomization in decision-making. To gain more insights into these empirical findings, we next analyze a case where pre-training under certain canonical assumptions provably leads to the quantal response behaviors and further yields no-regret guarantees.

**Further motivations for generalized quantal response.** Note that a *dynamic* version of quantal response in Definition F.2 also has implications from behavior economics, and has been recently used to model human behaviors in sequential decision-making (Ding et al., 2022) (in stochastic and stationary environments). Indeed, such a response against multiple loss vectors is believed to be natural, and has also been widely adopted in well-known no-regret learning algorithms of *smooth/stochastic fictitious play* (Fudenberg & Kreps, 1993) and *follow-the-perturbed-leader* (Kalai & Vempala, 2005), whose formal definitions can be found in Appendix D.3. Finally, note that the response model in Definition F.2 does not necessarily involve a *sequential* decision-making process, i.e., the set of losses may not come from the history of an online learning process.

### F.3. Detailed Explanation for the Meaning of $z$ and Assumptions on Pre-training Distribution

The pre-training dataset, however, only contains *partial observations* $x_{1:N_t}$ (a natural language representation of $\ell_{1:t}$) of $z$ due to imperfect data collection, which could be attributed to the fact that $z$ is private to the data-generator (human), representing the actual intention of the human/data-generator. Hence, LLM will only be pre-trained with partial and noisy information about $z$. Meanwhile, we assume that some high-quality action label $x_{N_t+1:N_{t+1}}$ (a natural language representation of $a$) w.r.t. the underlying loss vector $z$ is also available in the dataset, which could come from user surveys, personal blogs, or data annotation. We formalize such an assumption:

**Assumption 1** (Pre-training distribution). *Given $T \in \mathbb{N}^+$, $t \in [T]$, $N_{t+1} \in \mathbb{N}^+$, there are latent variables $(z, \ell_{1:t})$, $N_1, \cdots, N_t \in [N_{t+1}]$, $N_0 = 0$, such that $\mathbb{P}(z, \ell_{1:t}, x_{1:N_{t+1}}) = \mathbb{P}(z, \ell_{1:t})\mathbb{P}(x_{1:N_t} \mid \ell_{1:t})\mathbb{P}(x_{N_t+1:N_{t+1}} \mid z)$, and $P_t^{text}(x_{1:N_{t+1}}) := \mathbb{P}(x_{1:N_{t+1}}) = \int_z \int_{\ell_{1:t}} \mathbb{P}(z, \ell_{1:t}, x_{1:N_{t+1}}) d\ell_{1:t} dz$. Intuitively, tokens $\{x_{N_{i-1}+1:N_i}\}_{i \in [t]}$ encode the context, i.e., information for $\ell_{1:t}$, and the user will decode action $a$ from $x_{N_t+1:N_{t+1}}$.*

To further understand our assumption, we provide an example in Appendix F.4, showing how a natural text corpus may satisfy it. Similar assumptions that suppose the existence of such latent variables in generating the pre-training datasets have also been made recently in (Lee et al., 2023; Lin et al., 2024; Liu et al., 2023b), for understanding the in-context decision-making behaviors of LLMs/Transformers through posterior sampling, for which we defer a detailed comparison to Appendix F.9. In particular, we show in Theorem 4.1 that if the noise, i.e., $\ell_i - z$ is modeled as Gaussian distributions and $x_{N_t+1:N_{t+1}}$ encodes the optimal action for $z$, the pre-trained LLM provably recovers the prominent human behavior model in Appendix F.2.1, the quantal response model.

### F.4. The Example Instantiating Assumption 1

**Example 1** (An example instantiating Assumption 1). *We consider a common decision-making task that may generate the training data,* recommender systems. *An instance of the text data could be: "On September 29, 2023, user X clicked movie A three times, movie B eight times, and movie C five times". This sentence corresponds to $x_{N_{i-1}+1:N_i}$ for some $i \in [t]$ and serves as a natural language depiction of the numerical $\ell_i$. The corresponding label $x_{N_t+1:N_{t+1}}$ can be obtained by some user survey: "User X's favorite movie is movie B". Meanwhile, $z$ represents user X's latent, genuine preference for each movie – information that is private to the user, and cannot be observed or collected in the pre-training dataset. In this example, Assumption 1 suggests that $x_{1:N_t}$, which records the frequency of interactions with each movie, serves as an* imperfect estimate *of the user's latent, genuine preference for the movies, while the text $x_{N_t+1:N_{t+1}}$ depicts the user's favorite movie only based on her latent $z$.*

### F.5. Alignment of Assumption 1 with Quantal Response

Before presenting the technical lemma, based on Assumption 1, we denote the (potentially unkown) mappings that decode semantic information in Assumption 1 into numeric values as $f, g$, such that $f(x_{N_{i-1}+1:N_i}) = \ell_i \in \mathbb{R}^d$ for each $i \in [t]$ and $g(x_{N_t+1:N_{t+1}}) = a \in \mathcal{A}$.

**Lemma 1.** *Fix $t \in [T]$, $\sigma > 0$. If we model the noise of data collection to be i.i.d. Gaussian distribution in the numeric value space, i.e.,*

$$\mathbb{P}\left(\{f(x_{N_{i-1}+1:N_i})\}_{i\in[t]} \mid z\right) = \prod_{i=1}^{t} \mathbb{P}\left(f(x_{N_{i-1}+1:N_i}) \mid z\right) \propto \prod_{i=1}^{t} \exp\left(-\frac{\|f(x_{N_{i-1}+1:N_i}) - z\|_2^2}{2\sigma^2}\right),$$

*the prior distribution of the latent variable $z$ is also Gaussian, i.e., $z \sim \mathcal{N}(\mathbf{0}_d, \sigma^2 I)$, and the text labels satisfy that $\mathbb{P}(g(x_{N_t+1:N_{t+1}}) \mid z) = \mathbb{1}\left(g(x_{N_t+1:N_{t+1}}) \in \arg\min_{a\in\mathcal{A}} z_a\right)$, then we have*

$$\mathbb{P}\left(g(x_{N_t+1:N_{t+1}}) \mid x_{1:N_t}\right) = P_{quantal}^{\sigma\sqrt{t+1}}\left(g(x_{N_t+1:N_{t+1}}) \mid \{f(x_{N_{i-1}+1:N_i})\}_{i\in[t]}\right),$$

*with $P_{noise} = \mathcal{N}(\mathbf{0}_d, I)$ in Definition F.2, i.e., the action $a = g(x_{N_t+1:N_{t+1}})$ extracted from the text $x_{N_t+1:N_{t+1}}$ is a quantal response w.r.t. the loss vectors $\left(f(x_{N_{i-1}+1:N_i})\right)_{i\in[t]}$.*

*Proof.* Note that

$$\mathbb{P}(z \mid x_{1:N_t}) = \int_{\ell_{1:t}} \mathbb{P}(z, \ell_{1:t} \mid x_{1:N_t}) d\ell_{1:t} = \int_{\ell_{1:t}} \mathbb{P}(\ell_{1:t} \mid x_{1:N_t}) \mathbb{P}(z \mid x_{1:N_t}, \ell_{1:t}) d\ell_{1:t}.$$

For $\mathbb{P}(\ell_{1:t} \mid x_{1:N_t})$, since we have assumed the existence of function $f$ to decode $\ell_{1:t}$ from $x_{1:N_t}$, it holds that

$$\mathbb{P}(\ell_{1:t} \mid x_{1:N_t}) = \prod_{i=1}^{t} \delta\left(\ell_i - f(x_{N_{i-1}+1:N_i})\right),$$

where we use $\delta$ to denote the $d$-dimensional Dirac-delta function. For $\mathbb{P}(z \mid x_{1:N_t}, \ell_{1:t})$, by Assumption 1, it holds that

$$\mathbb{P}(z, x_{1:N_t}, \ell_{1:t}) = \mathbb{P}(z, \ell_{1:t}) \mathbb{P}(x_{1:N_t} \mid \ell_{1:t}),$$

which leads to $\mathbb{P}(x_{1:N_t} \mid \ell_{1:t}) = \mathbb{P}(x_{1:N_t} \mid \ell_{1:t}, z)$ by Bayes rule. This implies that the random variable $x_{1:N_t}$ and $z$ are independent conditioned on $\ell_{1:t}$. Therefore, it holds that $\mathbb{P}(z \mid x_{1:N_t}, \ell_{1:t}) = \mathbb{P}(z \mid \ell_{1:t})$. Finally, we can compute

$$\mathbb{P}(z \mid x_{1:N_t}) = \int_{\ell_{1:t}} \mathbb{P}(z, \ell_{1:t} \mid x_{1:N_t}) d\ell_{1:t} = \int_{\ell_{1:t}} \prod_{i=1}^{t} \delta(\ell_i - f(x_{N_{i-1}+1:N_i})) \mathbb{P}(z \mid \ell_{1:t}) d\ell_{1:t}$$

$$= \mathbb{P}\left(z \mid \left(\ell_i = f(x_{N_{i-1}+1:N_i})\right)_{i\in[t]}\right).$$

Based on this, we conclude that

$$\mathbb{P}(g(x_{N_t+1:N_{t+1}}) \mid x_{1:N_t}) = \int_z \mathbb{P}(g(x_{N_t+1:N_{t+1}}) \mid z, x_{1:N_t}) \mathbb{P}(z \mid x_{1:N_t}) dz$$

$$= \int_z \mathbb{P}(g(x_{N_t+1:N_{t+1}}) \mid z) \mathbb{P}(z \mid \{\ell_i = f(x_{N_{i-1}+1:N_i})\}_{i\in[t]}) dz$$

$$= \mathbb{P}\left(g(x_{N_t+1:N_{t+1}}) \mid \left(\ell_i = f(x_{N_{i-1}+1:N_i})\right)_{i\in[t]}\right)$$

where the first equality is by the independence between $x_{N_t+1:N_{t+1}}$ and $x_{1:N_t}$ conditioned on $z$, due to Assumption 1. Therefore, it suffices to consider the probability of $\mathbb{P}(a \mid \ell_{1:t})$ only, in order to analyze $\mathbb{P}(g(x_{N_t+1:N_{t+1}}) \mid x_{1:N_t})$, where we recall the definition that $a = g(x_{N_t+1:N_{t+1}})$. Since $z \sim \mathcal{N}(\mathbf{0}_d, \sigma^2 I)$, and $\ell_i \mid z \sim \mathcal{N}(z, \sigma^2 I)$, we have

$$z \mid \ell_{1:t} \sim \mathcal{N}\left(\frac{1}{t+1}\sum_{i\in[t]} \ell_i, \frac{\sigma^2}{t+1} I\right), \tag{F.1}$$

by the posterior distribution of Gaussian distribution. Now we conclude that

$$\mathbb{P}(a \,|\, \ell_{1:t}) = \int_z \mathbb{P}(a \,|\, z, \ell_{1:t}) \mathbb{P}(z \,|\, \ell_{1:t}) dz = \int_z \mathbb{P}(a \,|\, z) \mathbb{P}(z \,|\, \ell_{1:t}) dz$$

$$= \int_z \mathbb{1}(a \in \operatorname*{argmin}_{a' \in \mathcal{A}} z_{a'}) \mathbb{P}(z \,|\, \ell_{1:t}) dz = \int_z \mathbb{1}\left(a \in \operatorname*{argmin}_{a' \in \mathcal{A}}\left(\frac{\sigma}{\sqrt{t+1}}\epsilon + \frac{1}{t+1}\sum_{i\in[t]}\ell_i\right)_{a'}\right)\mathbb{P}(\epsilon)d\epsilon$$

$$= \int_z \mathbb{1}\left(a \in \operatorname*{argmin}_{a' \in \mathcal{A}}\left(\sigma\sqrt{t+1}\epsilon + \sum_{i\in[t]}\ell_i\right)_{a'}\right)\mathbb{P}(\epsilon)d\epsilon = \mathbb{P}\left(a \in \operatorname*{argmin}_{a' \in \mathcal{A}}\left(\sigma\sqrt{t+1}\epsilon + \sum_{i\in[t]}\ell_i\right)_{a'}\right)$$

$$= P_{quantal}^{\sigma\sqrt{t+1}}(a \,|\, \ell_{1:t}),$$

where $\mathbb{P}(\epsilon) = \mathcal{N}(\mathbf{0}_d, I)$. This completes the proof. $\qquad\square$

## F.6. Relationship between FTPL and Definition F.2

**Fact 1.** *Performing generalized quantal response of Definition F.2 at every iteration $t \in [T]$ w.r.t. history loss vectors $\ell_{1:t-1}$ is essentially executing an FTPL algorithm.*

*Proof.* Before we move to the proof, we will define the random variable which has distribution $P_{\text{noise}}$ as $Z_{\text{noise}}$. Note that at round $t \geq 2$ (as the policy at round $t = 1$ is fixed), we have

$$P_{\text{quantal}}^{\eta_{t-1}}(a \,|\, \ell_{1:t-1}) := \mathbb{P}\left(a \in \operatorname*{argmin}_{a' \in \mathcal{A}}\left(\sum_{i=1}^{t-1}\ell_i + \eta_{t-1}\epsilon\right)(a')\right) \tag{F.2}$$

which is exactly the case when $\epsilon_t$ in Equation (D.2) satisfies $\epsilon_t \stackrel{d}{=} \eta_{t-1}\epsilon$. $\qquad\square$

## F.7. Formal Statement and Proof of Theorem 4.1

**Theorem F.1.** (Emergence of no-regret behavior). *Under the assumptions of Lemma 1, suppose the function class of $LLM_\theta$ is expressive enough such that for all $t \in [T]$, $\max_{\theta \in \Theta}\mathbb{E}_{x_{1:N_{t+1}} \sim P_t^{text}}\sum_{j=1}^{N_{t+1}}\log LLM_\theta\left(x_j \,|\, x_{1:j-1}\right) = \max_{\{q_j \in \{\mathcal{V}^{j-1} \to \Delta(\mathcal{V})\}\}_{j\in[N_{t+1}]}}\mathbb{E}_{x_{1:N_{t+1}} \sim P_t^{text}}\sum_{j=1}^{N_{t+1}}\log q_j\left(x_j \,|\, x_{1:j-1}\right)$, where we define $q_1(x_1 \,|\, x_{1:0}) := q_1(x_1)$, and $\theta^\star$ maximizes Equation (4.1). Then, there exist (simple) algorithms using $LLM_{\theta^\star}$ to achieve no (dynamic) regret for (non-stationary) online learning with full-information/bandit feedback. To be specific, for (2) and (4), by defining the variation bound $\sum_{t=1}^{T-1}\|\ell_{t+1} - \ell_t\|_\infty \leq V_T$ such that $V_T \leq T$ and $V_T = \Theta(T^\rho)$ for some $\rho \in (0, 1)$, it holds that for large enough $T, d$:*

*(1) For online learning with full-information feedback, $Regret_{LLM_{\theta^\star}}\left((\ell_t)_{t\in[T]}\right) \leq \mathcal{O}\left(\sqrt{T\log d}\right)$;*

*(2) For non-stationary online learning with full-information feedback, $D\text{-}Regret_{LLM_{\theta^\star}}\left((\ell_t)_{t\in[T]}\right) \leq \mathcal{O}\left((\log d\, V_T)^{1/3}T^{2/3}\right)$;*

*(3) For online learning with bandit feedback, $\mathbb{E}\left[Regret_{LLM_{\theta^\star}}\left((\ell_t)_{t\in[T]}\right)\right] \leq \mathcal{O}\left((\log d)^{1/2}dT^{1/2+1/\log T}\log T\right)$;*

*(4) For non-stationary online learning with bandit feedback, $\mathbb{E}\left[D\text{-}Regret_{LLM_{\theta^\star}}\left((\ell_t)_{t\in[T]}\right)\right] \leq \mathcal{O}\left((T^2d^2V_T)^{1/3}(\log d)^{1/2}T^{1/\log T}\log T\right)$.*

*Proof.* Note that

$$\max_{\{q_j \in \{\mathcal{V}^{j-1} \to \Delta(\mathcal{V})\}\}_{j \in [N_{t+1}]}} \mathbb{E}_{x_{1:N_{t+1}} \sim P_t^{text}} \sum_{j=1}^{N_{t+1}} \log q_j \left(x_j \mid x_{1:j-1}\right)$$

$$= \max_{q \in \Delta(\mathcal{V}^{N_{t+1}})} \mathbb{E}_{x_{1:N_{t+1}} \sim P_t^{text}} \log q(x_{1:N_{t+1}})$$

$$= \max_{q \in \Delta(\mathcal{V}^{N_{t+1}})} -\mathrm{KL}(P_t^{text} \mid\mid q) + \mathbb{E}_{x_{1:N_{t+1}} \sim P_t^{text}}[P_t^{text}(x_{1:N_{t+1}})],$$

where $\mathrm{KL}(q \mid\mid p)$ denotes the Kullback–Leibler divergence between two distributions $p, q$. Now we define $\mathrm{LLM}_\theta(x_{1:N_{t+1}}) = \prod_{t=1}^{N_{t+1}} \mathrm{LLM}_\theta(x_j \mid x_{1:j-1})$. It is easy to verify that $\mathrm{LLM}_\theta(x_{1:N_{t+1}}) \in \Delta(\mathcal{V}^{N_{t+1}})$, i.e., it also defines a valid joint distribution over tokens. Therefore, we have

$$\max_{\theta \in \Theta} \mathbb{E}_{x_{1:N_{t+1}} \sim P_t^{text}} \sum_{j=1}^{N_{t+1}} \log \mathrm{LLM}_\theta \left(x_j \mid x_{1:j-1}\right) = \max_{\theta \in \Theta} \mathbb{E}_{x_{1:N_{t+1}} \sim P_t^{text}} \log \mathrm{LLM}_\theta(x_{1:N_{t+1}}).$$

Now, due to our assumption that

$$\max_{\theta \in \Theta} \mathbb{E}_{x_{1:N_{t+1}} \sim P_t^{text}} \sum_{j=1}^{N_{t+1}} \log \mathrm{LLM}_\theta \left(x_j \mid x_{1:j-1}\right)$$

$$= \max_{\{q_j \in \{\mathcal{V}^{j-1} \to \Delta(\mathcal{V})\}\}_{j \in [N_{t+1}]}} \mathbb{E}_{x_{1:N_{t+1}} \sim P_t^{text}} \sum_{j=1}^{N_{t+1}} \log q_j \left(x_j \mid x_{1:j-1}\right),$$

we conclude that

$$\min_{\theta \in \Theta} \mathrm{KL}(P_t^{text} \mid\mid \mathrm{LLM}_\theta) = \min_{q \in \Delta(\mathcal{V}^{N_{t+1}})} \mathrm{KL}(P_t^{text} \mid\mid q) = 0,$$

which implies that $\mathrm{LLM}_{\theta^\star} = P_t^{text}$. Correspondingly, if we define $\mathrm{LLM}_{\theta^\star}(x_{N_t+1:N_{t+1}} \mid x_{1:N_t})$ to be the distribution induced by the joint distribution $\mathrm{LLM}_{\theta^\star}(x_{1:N_{t+1}})$, it holds that

$$\mathrm{LLM}_{\theta^\star}(x_{N_t+1:N_{t+1}} \mid x_{1:N_t}) = \mathbb{P}(x_{N_t+1:N_{t+1}} \mid x_{1:N_t}).$$

In other words, intuitively, $\mathrm{LLM}_{\theta^\star}$ has learned the corresponding *pre-training* distribution perfectly. Note that this has been a common assumption in the Bayesian perspective of ICL (Xie et al., 2022; Lee et al., 2023; Zhang et al., 2023b). Therefore, to analyze the actions taken by $\mathrm{LLM}_{\theta^\star}$, it suffices to consider $\mathbb{P}(g(x_{N_t+1:N_{t+1}}) \mid x_{1:N_t})$, which is equal to $P_{quantal}^{\sigma\sqrt{t+1}} \left(g(x_{N_t+1:N_{t+1}}) \mid \{f(x_{N_{i-1}+1:N_i})\}_{i \in [t]}\right)$ by Lemma 1. Therefore, we proved that $\mathrm{LLM}_{\theta^\star}$ is essentially mimicking the well-known no-regret algorithm, FTPL with perturbation distribution as $\mathcal{N}(\mathbf{0}_d, \sigma^2 tI)$ for round $t \in [T]$, according to Equation (F.2) of Fact 1, for which we can establish the corresponding regret guarantee for each case:

(1) Combining the above result with Lemma 2, we can derive the regret bound for online learning with full-information feedback.

(2) Combining the above result with Lemma 2 and Lemma 4, we get that

$$\mathrm{D\text{-}Regret}_{\mathrm{LLM}_{\theta^\star}}((\ell_i)_{i \in [T]}) \leq \min_{\Delta_T \in [T]} \frac{2T}{\Delta_T} C\sqrt{\Delta_T \log d} + 2\Delta_T V_T,$$

for some constant $C$. We firstly consider the following problem

$$\min_{u > 0} \frac{2T}{u} C\sqrt{u \log d} + 2u V_T,$$

where the optimal solution is $u^\star = \left(\frac{C^2 T^2 \log d}{4 V_T^2}\right)^{1/3}$. Therefore, if we have $u^\star \in [1, T]$, we can choose $\Delta_T = \lceil u^\star \rceil$, which results in a regret bound of

$$\mathrm{D\text{-}Regret}_{\mathrm{LLM}_{\theta^\star}}((\ell_i)_{i \in [T]}) \leq \frac{2T}{\sqrt{u^\star}} C\sqrt{\log d} + 4u^\star V_T = \mathcal{O}\left((\log d \, V_T)^{1/3} T^{2/3}\right).$$

Now we check the conditions for $u^\star \in [1, T]$. It is direct to see that since $V_T \leq T$, $u^\star \geq 1$ holds as long as $d$ is sufficiently large. To ensure $u^\star \leq T$, we get the condition $V_T \geq C\sqrt{\frac{\log d}{4T}}$, which holds as long as $T$ is large enough.

(3) Combining the above result with Lemma 3, we can prove a regret guarantee for online learning with bandit feedback.

(4) Combining this result with Lemma 3 and Lemma 4, it holds that

$$\mathbb{E}[\text{D-Regret}_{\text{LLM}_{\theta^\star}}((\ell_i)_{i\in[T]})] \leq \min_{\Delta_T \in [T]} \frac{2T}{\Delta_T} C(\log d)^{\frac{1}{2}} d\Delta_T^{\frac{1}{2}+\frac{1}{\log T}} \log \Delta_T + 2\Delta_T V_T,$$

for some constant $C$. By adopting a similar analysis as that of (2), we choose $u^\star = \left(\frac{C'T^2 d^2}{V_T^2}\right)^{1/3}$ for some constant $C'$. If $u^\star \in [1, T]$, we choose $\Delta_T = \lceil u^\star \rceil$ and derive the following regret:

$$\mathbb{E}[\text{D-Regret}_{\text{LLM}_{\theta^\star}}((\ell_i)_{i\in[T]})] \leq \mathcal{O}\left((T^2 d^2 V_T)^{1/3}(\log d)^{1/2} T^{1/\log T} \log T\right).$$

Now we check the condition of $u^\star \in [1, T]$. Note that since $V_T \leq T$, $u^\star \geq 1$ holds as long as $d$ is sufficiently large. For $u^\star \leq T$, we have $V_T \geq \sqrt{\frac{C' d^2}{T}}$, which holds as long as $T$ is large enough.

Now, we present Lemma 2 - Lemma 4. Before proceeding, we assume $\|\ell_t\|_\infty \leq B = 1$ for simplicity of presentations hereafter. The results and proof are not affected by the constant bound $B$.

**Lemma 2** (Regret guarantee of FTPL with full-information feedback). *Suppose the noise distribution of FTPL satisfies that $\epsilon_t \sim \mathcal{N}(\mathbf{0}_d, \zeta_t^2 I)$ in Equation (D.2) and $\zeta_t = \sigma\sqrt{t}$, then for online learning with full-information feedback,*

$$\text{Regret}_{\text{FTPL}}((\ell_i)_{i\in[T]}) \leq 4\left(\sigma + \frac{1}{\sigma}\right)\sqrt{T \log d} = \mathcal{O}(\sqrt{T \log d}).$$

*Proof.* By Theorem 8 of (Abernethy et al., 2014), we have

$$\text{Regret}_{\text{FTPL}}((\ell_i)_{i\in[T]}) \leq \sqrt{2\log d}\left(\eta_T + \sum_{t=1}^{T} \frac{1}{\eta_t}\|\ell_t\|_\infty^2\right).$$

Therefore, plugging $\zeta_t = \sigma\sqrt{t}$ and $\|\ell_t\|_\infty^2 \leq 1$ provides

$$\text{Regret}_{\text{FTPL}}((\ell_i)_{i\in[T]}) \leq \sqrt{2\log d}\left(\sigma\sqrt{T} + \sum_{t=1}^{T} \frac{1}{\sigma\sqrt{t}}\right) \leq 4\left(\sigma + \frac{1}{\sigma}\right)\sqrt{T \log d},$$

completing the proof. $\square$

**Lemma 3** (Regret guarantee of FTPL with bandit feedback). *Suppose the noise distribution of FTPL satisfies that $\epsilon_t \sim \mathcal{N}(\mathbf{0}_d, \zeta_t^2 I)$ in Equation (D.2) and $\zeta_t = \sigma\sqrt{t}$, then for online learning with bandit feedback,*

$$\mathbb{E}[\text{Regret}_{\text{FTPL}}((\ell_i)_{i\in[T]})] \leq \mathcal{O}((\log d)^{\frac{1}{2}} d T^{\frac{1}{2}+\frac{1}{\log T}} \log T).$$

*Proof.* The proof of the bandit problem is more complex. We first define the following notation. We denote $G_t = \sum_{t'=1}^{t} -\ell_{t'}$, $\widehat{G}_t = \sum_{t'=1}^{t} -\widehat{\ell}_{t'}$, $\Phi(G) = \max_\pi \langle \pi, G \rangle$, $\Phi_t(G) = \mathbb{E}_{\epsilon \sim \mathcal{N}(\mathbf{0}_d, I)}\Phi(G + \zeta_t \epsilon)$, and $D_{\Phi_t}$ to be the Bregman divergence with respect to $\Phi_t$, where we recall the construction of the empirical estimator $\widehat{\ell}_{t'}$ of $\ell_{t'}$ in Section 3.2. By (Li & Tewari, 2017), $\pi_t = \nabla \Phi_t(\widehat{G}_t)$. Now due to the convexity of $\Phi$,

$$\Phi(G_T) = \Phi(\mathbb{E}[\widehat{G}_T]) \leq \mathbb{E}[\Phi(\widehat{G}_T)].$$

Therefore,

$$\mathbb{E}[\text{Regret}_{\text{FTPL}}((\ell_i)_{i\in[T]})] = \Phi(G_T) - \mathbb{E}\left[\sum_{t=1}^{T}\langle \pi_t, -\widehat{\ell}_t \rangle\right] \leq \mathbb{E}\left[\Phi(\widehat{G}_T) - \sum_{t=1}^{T}\langle \pi_t, -\widehat{\ell}_t \rangle\right].$$

By recalling the definition of the Bregman divergence, we have

$$-\sum_{t=1}^{T}\langle \pi_t, -\widehat{\ell}_t\rangle = -\sum_{t=1}^{T}\langle \nabla\Phi_t(\widehat{G}_t), -\widehat{\ell}_t\rangle = -\sum_{t=1}^{T}\langle \nabla\Phi_t(\widehat{G}_t), \widehat{G}_t - \widehat{G}_{t-1}\rangle$$

$$= \sum_{t=1}^{T} D_{\Phi_t}(\widehat{G}_t, \widehat{G}_{t-1}) + \Phi_t(\widehat{G}_{t-1}) - \Phi_t(\widehat{G}_t).$$

Therefore,

$$\mathbb{E}\left[\text{Regret}_{\text{FTPL}}((\ell_i)_{i\in[T]})\right]$$

$$\leq \underbrace{\mathbb{E}\left[\sum_{t=1}^{T} D_{\Phi_t}(\widehat{G}_t, \widehat{G}_{t-1})\right]}_{(i)} + \underbrace{\mathbb{E}\left[\sum_{t=1}^{T} \Phi_t(\widehat{G}_{t-1}) - \Phi_{t-1}(\widehat{G}_{t-1})\right]}_{(ii)} + \underbrace{\mathbb{E}\left[\Phi(\widehat{G}_T) - \Phi_T(\widehat{G}_T)\right]}_{(iii)}.$$

$(iii) \leq 0$ due to the convexity of $\Phi$. For $(ii)$, we use Lemma 10 of (Abernethy et al., 2014) to obtain

$$\mathbb{E}\left[\sum_{t=1}^{T} \Phi_t(\widehat{G}_{t-1}) - \Phi_{t-1}(\widehat{G}_{t-1})\right] \leq \zeta_T \mathbb{E}_{\epsilon}[\Phi(\epsilon)] \leq \mathcal{O}(\sqrt{2T\log d}).$$

For $(i)$, by Theorem 8 of (Li & Tewari, 2017), for any $\alpha \in (0, 1)$, the following holds:

$$\mathbb{E}\left[\sum_{t=1}^{T} D_{\Phi_t}(\widehat{G}_t, \widehat{G}_{t-1})\right] \leq \sum_{t=1}^{T} \zeta_t^{\alpha-1} \frac{4d}{\alpha(1-\alpha)} \leq \frac{4d}{\alpha(1-\alpha)} \mathcal{O}(T^{\frac{1+\alpha}{2}}).$$

By tuning $\alpha = \frac{2}{\log T}$, we proved that $\mathbb{E}[\text{Regret}_{\text{FTPL}}((\ell_i)_{i\in[T]})] \leq \mathcal{O}((\log d)^{\frac{1}{2}} dT^{\frac{1}{2}+\frac{1}{\log T}}\log T)$. $\qquad\square$

**Lemma 4.** *Denote the variation of loss vectors as $L_T = \sum_{t=1}^{T-1} \|\ell_{t+1} - \ell_t\|_\infty$. Suppose there exists an algorithm $\mathscr{A}$ for online learning with full-information feedback with regret guarantee that $\text{Regret}_{\mathscr{A}}((\ell_i)_{i\in[T]}) \leq f(T, d)$ for some function $f$, where $T$ denotes the horizon and $d$ denotes the policy dimension. Then, there exists another algorithm $\mathscr{A}'$ that can achieve*

$$\text{D-Regret}_{\mathscr{A}'}((\ell_i)_{i\in[T]}) \leq \min_{\Delta_T \in [T]} \left(\frac{T}{\Delta_T} + 1\right) f(\Delta_T, d) + 2\Delta_T L_T.$$

*Similarly, suppose there exists an algorithm $\mathscr{B}$ for online learning with bandit feedback with regret guarantee that $\mathbb{E}\left[\text{Regret}_{\mathscr{B}}((\ell_i)_{i\in[T]})\right] \leq g(T, d)$ for some function $g$; then there exists another algorithm $\mathscr{B}'$ that can achieve*

$$\mathbb{E}[\text{D-Regret}_{\mathscr{B}'}((\ell_i)_{i\in[T]})] \leq \min_{\Delta_T \in [T]} \left(\frac{T}{\Delta_T} + 1\right) g(\Delta_T, d) + 2\Delta_T L_T.$$

*Proof.* We denote $\mathscr{A}'$ as the algorithm that restarts $\mathscr{A}$ every $\Delta_T$ iterations. We break the time index $[T]$ into $m$ batches $\mathcal{T}_{1:m}$ of size $\Delta_T$ (except for, possibly the last batch). Denote $\ell_i^\star := \min_{j\in[d]} \ell_{ij}$. By Equation (6) of (Besbes et al., 2014), it holds that for each $k \in [m]$

$$\min_{j\in[d]} \left(\sum_{t\in\mathcal{T}_k} \ell_t\right)_j - \sum_{t\in\mathcal{T}_k} \ell_t^\star \leq 2\Delta_T L_k,$$

where we define $L_k = \sum_{t\in\mathcal{T}_k} \|\ell_{t+1} - \ell_t\|_\infty$. Therefore, we have

$$\text{D-Regret}_{\mathscr{A}'}((\ell_i)_{i\in[T]}) \leq \min_{j\in[d]} \left(\sum_{t\in[T]} \ell_t\right)_j - \sum_{t\in[T]} \ell_t^\star + \sum_{k\in[m]} \text{Regret}_{\mathscr{A}}((\ell_i)_{i\in[\mathcal{T}_k]}) \tag{F.3}$$

$$\leq 2\Delta_T (\sum_{k\in[m]} L_k) + (T/\Delta_T + 1)g(\Delta_T, d).$$

By Equation (4) of (Besbes et al., 2014) that $\sum_{k \in [m]} L_k \le L_T$ and this inequality holds for any $\Delta_T \in [T]$, we proved

D-Regret$_{\mathscr{A}'}((\ell_i)_{i \in [T]}) \le \min_{\Delta_T \in [T]} \left( \frac{T}{\Delta_T} + 1 \right) f(\Delta_T, d) + 2\Delta_T L_T$.

Similarly, if we take the expectation for Equation (F.3), it holds that

$$\mathbb{E}[\text{D-Regret}_{\mathscr{B}'}((\ell_i)_{i \in [T]})] \le \min_{j \in [d]} \left( \sum_{t \in [T]} \ell_t \right)_j - \sum_{t \in [T]} \ell_t^\star + \sum_{k \in [m]} \mathbb{E}[\text{Regret}_{\mathscr{B}}((\ell_i)_{i \in [\mathcal{T}_k]})]$$

$$\le \min_{\Delta_T \in [T]} \left( \frac{T}{\Delta_T} + 1 \right) g(\Delta_T, d) + 2\Delta_T L_T,$$

thus completing the proof. $\qquad\square$

Combining the results above completes the proof for Theorem 4.1. $\qquad\square$

### F.7.1. IMPLICATIONS OF THEOREM 4.1 FOR REPEATED GAMES

**Remark F.1** (Implication for playing repeated games). *First, we note that the no-regret guarantee in the online setting is stronger than and thus implies that in the game setting, since regret by definition handles arbitrary/adversarial environments, while in playing games the opponents are not necessarily as adversarial. Second, it is folklore that if all players in the repeated game follow no-regret learning algorithms, then the time-average policies of all players during learning constitute an approximate **coarse correlated equilibrium** of the game (Cesa-Bianchi & Lugosi, 2006). Hence, the results (1) and (2) in Theorem 4.1 imply that a coarse correlated equilibrium will emerge in the long run from the interactions of the LLM agents (under certain assumptions as in the theorem).*

### F.8. Extending Theorem 4.1 with Relaxed Assumptions

#### F.8.1. RELAXATION UNDER MORE GENERAL DATA DISTRIBUTIONS

We first remark on the possibility of relaxing the Gaussian assumptions on the data distributions.

**Remark F.2** (Relaxing the Gaussian distribution assumption). *In the proof of Lemma 1, to obtain the result that the action is a quantal response w.r.t. $\ell_{1:T}$, one does not necessarily require* both *the prior distribution of $z$ and the conditional distribution of $\ell_i$ given $z$ to be Gaussian. Instead, for* any *joint distribution $\mathbb{P}(z, \ell_{1:T})$, as long as its posterior distribution satisfies Equation (F.1), it would suffice. It is a combined effect of both the prior and the conditional distributions.*

More formally, we can extend Theorem 4.1 to the case with a much more general prior task distribution than the Gaussian one, where the key is that Equation (F.1) only needs to hold approximately.

**Theorem F.2.** *In Theorem 4.1, we can relax the assumption on $\mathbb{P}(z)$ to one where we only require $\mathbb{P}(z)$ to be i.i.d for each coordinate of $z$ and $0 < \mathbb{P}(z_j) < \infty$, $|\nabla \mathbb{P}(z_j)| < \infty$ for any $j \in [d]$, $z_j \in \mathbb{R}$, and the bounds for (1) and (2) of Theorem 4.1 still hold, with only a degradation of $\mathcal{O}(d^2 \log T)$.*

The key idea of the proof is that when $t$ is large enough, the prior distribution does not affect the posterior distribution, which is also referred to as the *Bernstein–von Mises theorem* (Van der Vaart, 2000).

*Proof.* Since we extend Theorem 4.1 to settings with general task prior distribution only requiring the coordinates to be i.i.d, from now on, we consider the $j$-th coordinate only. To begin with, fix $t \in [T]$, we define the log-likelihood of the posterior as

$$L_t(z_j) := \log \prod_{i=1}^t \frac{1}{\sigma^d (2\pi)^{d/2}} e^{-\frac{1}{2\sigma^2}(\ell_{ij} - z_j)^2} = -t \log \sigma - \frac{t}{2} \log 2\pi - \sum_{i=1}^t \frac{1}{2\sigma^2}(\ell_{ij} - z_j)^2.$$

Then, the MLE estimator $\widehat{z}_{j,t}$ is defined as

$$\widehat{z}_{j,t} := \arg\max_{z_j \in \mathbb{R}} L_t(z_j) = \frac{1}{t} \sum_{i=1}^t \ell_{ij}.$$

We also define $\widehat{J}_t : \mathbb{R} \to \mathbb{R}$ as:

$$\widehat{J}_t(z_j) := -\frac{\nabla^2 L_t(z_j)}{t} = \frac{1}{\sigma^2}.$$

For Assumption 1 of (Kasprzak et al., 2022) to hold, any $\delta > 0$, $M_2 > 0$ suffices.

For Assumption 2 of (Kasprzak et al., 2022) to hold, we can choose $\widehat{M}_1 = \max_{z_j \in [-\delta, 1+\delta]} \frac{1}{\mathbb{P}(z_j)}$

For Assumption 7 of (Kasprzak et al., 2022) to hold, we choose $\delta$ to be $\sigma$.

For Assumption 8 of (Kasprzak et al., 2022) to hold, one can choose $M_2 = \frac{\sigma}{2}$.

For Assumption 9 of (Kasprzak et al., 2022) to hold, we have

$$\kappa \leq - \sup_{(z_j - \widehat{z}_j)^2 \geq \delta} \frac{L_t(z_j) - L_t(\widehat{z}_{j,t})}{t} = -\frac{1}{2\sigma^2 t} \sup_{(z_j - \widehat{z}_{j,t})^2 \geq \delta} \sum_{i=1}^t (\ell_{ij} - \widehat{z}_{j,t})^2 - (\ell_{ij} - z_j)^2 = \frac{1}{4\sigma}.$$

For Assumption 10 of (Kasprzak et al., 2022) to hold, we choose $M_1 = \sup_{z_j \in [-\delta, 1+\delta]} \left| \frac{\nabla \mathbb{P}(z_j)}{\mathbb{P}(z_j)} \right|$, $\widetilde{M}_1 = \sup_{z_j \in [-\delta, 1+\delta]} |\mathbb{P}(z_j)|$ since we have assumed that $0 < \mathbb{P}(z_j) < \infty$, $|\nabla \mathbb{P}(z_j)| < \infty$.

By Theorem 6.1 of (Kasprzak et al., 2022), we have

$$\int_{z_j} |\mathbb{P}(z_j/\sqrt{t} + \widehat{z}_j \,|\, (\ell_{ij})_{i \in [t]}) - C e^{-\frac{1}{2\sigma^2} z_j^2}| dz_j$$

$$= \sqrt{t} \int_{z_j} |\mathbb{P}(z_j \,|\, (\ell_{ij})_{i \in [t]}) - \mathcal{N}(\widehat{z}_j, \frac{\sigma^2}{t})| dz_j \leq D_1 t^{-1/2} + D_2 t^{1/2} e^{-t\kappa} + 2\widehat{\mathcal{D}}(t, \delta),$$

where $C$ is the normalization constant and

$$D_1 = \frac{\sqrt{\widetilde{M}_1 \widehat{M}_1}}{\sigma} \left( \frac{\sqrt{3}\sigma^2}{2\left(1 - \sqrt{\widehat{\mathcal{D}}(t, \delta)}\right)} M_2 + M_1 \right)$$

$$D_2 = \frac{2\widehat{M}_1 \widehat{J}_t^p(\widehat{z}_j, \delta)}{(2\pi)^{1/2}(1 - \widehat{\mathcal{D}}^p(t, \delta))}$$

$$\widehat{\mathcal{D}}(t, \delta) = e^{-\frac{1}{2}(\sqrt{t}-1)^2}$$

$$\widehat{J}_t^p(\widehat{z}_j, \delta) = \frac{1}{\sigma^2} + \frac{\delta M_2}{3}.$$

Therefore, we conclude that the TV distance between $z$ (conditioned on $(\ell_i)_{i \in [t]}$) and $\mathcal{N}\left(\widehat{z}, \frac{\sigma^2}{t}\right)$ satisfies that

$$\int_z \left| \mathbb{P}(z \,|\, (\ell_i)_{i \in [t]}) - \mathcal{N}\left(\widehat{z}, \frac{\sigma^2}{t}\right) \right| dz \leq \sum_{j=1}^d \int_{z_j} \left| \mathbb{P}(z_j \,|\, (\ell_{ij})_{i \in [t]}) - \mathcal{N}\left(\widehat{z}_j, \frac{\sigma^2}{t}\right) \right| dz_j \leq \mathcal{O}(d/t),$$

due to the independence of $(z_j)_{j \in [d]}$ conditioned on $\ell_{1:t}$. Now we denote algorithm $\widehat{\text{FTPL}}$ to be the FTPL algorithm w.r.t. the noise distribution $\mathbb{P}(z \,|\, (\ell_i)_{i \in [t]})$, and FTPL to be the algorithm w.r.t. the noise distribution $\mathcal{N}(\widehat{z}, \frac{\sigma^2}{t})$. Therefore, we have

$$\left| \text{Regret}_{\text{FTPL}}((\ell)_{i \in [T]}) - \text{Regret}_{\widehat{\text{FTPL}}}((\ell)_{i \in [T]}) \right| \leq \sum_{t=1}^T d \|\pi_t - \widehat{\pi}_t\|_\infty$$

$$\leq d \sum_{t=1}^T \int_z \left| \mathbb{P}(z \,|\, (\ell_i)_{i \in [t]}) - \mathcal{N}(\widehat{z}, \frac{\sigma^2}{t}) \right| dz = \mathcal{O}(d^2 \log T).$$

In other words, using $\mathbb{P}(z \,|\, (\ell_i)_{i\in[t]})$ as the noise distribution only increases the regret by $\mathcal{O}(d^2 \log T)$. Similarly, it is easy to see that

$$\left| \text{D-Regret}_{\text{FTPL}}((\ell)_{i\in[T]}) - \text{D-Regret}_{\widehat{\text{FTPL}}}((\ell)_{i\in[T]}) \right| \leq \mathcal{O}(d^2 \log T),$$

which completes the proof. $\qquad\square$

### F.8.2. RELAXATION UNDER DECISION-IRRELEVANT PRE-TRAINING DATA

We then remark on the possible relaxation when the training data may not all come from decision-making tasks.

**Remark F.3** (Pre-training with relaxed data assumptions). *Note that the pre-training (text) data are so far assumed to be related to* decision-making *problems (though not necessarily* sequential ones*), see Assumption 1 and Example 1 for instance. It can also be generalized to the text datasets involving* Question-Answering *(Q-A), a typical task in natural language processing, where the* true/fact *answer, sampled answers from different human users (with possibly wrong or biased answers), correspond to the latent $z$ (and associated maximizer $a$) and $\ell_{1:t}$, respectively. Moreover, in practice, the pre-training data may also involve* non-decision-making/Q-A *texts, given the diversity of the datasets. For such scenarios, we will make the assumptions on the data distribution* conditioned on the prompt for decision-making. *Specifically, when interacting with the LLM, human users will provide prompts (see e.g., our Figure E.1), to induce it to make decisions. This will query the* conditional *distribution of*

$$\mathbb{P}\left( g(x_{N_t+1:N_{t+1}}) \,\big|\, x_{1:N_t}, decision\text{-}making\ prompt \right)$$

*to generate the control action. Correspondingly, Assumption 1 will thus only need to be made on*

$$\mathbb{P}\left( z, \ell_{1:t}, x_{1:N_{t+1}}, decision\text{-}making\ prompt \right),$$

*while we do not need to make such assumptions on other prompts, e.g., corpora that are not related to decision-making.*

### F.9. Comparison with (Lee et al., 2023; Lin et al., 2024; Liu et al., 2023b)

Intriguingly, similar assumptions and pre-training objectives have also been considered in the very recent work of (Lee et al., 2023; Lin et al., 2024; Liu et al., 2023b) for studying in-context reinforcement learning property of Transformers/LLM-agents under supervised pre-training. (Lee et al., 2023) established its equivalence to *posterior sampling* (Osband et al., 2013), an important RL algorithm with provable regret guarantees when the environments are *stationary*, and (Lin et al., 2024) generalized the study to the setting of algorithm distillation as in (Laskin et al., 2023). (Liu et al., 2023b) adopted the similar data generation assumption as (Lee et al., 2023) without assuming optimal labels are available in the pre-training datasets, but leverages external oracles for *planning*. Consequently, the resulting LLM agent would still perform the posterior sampling algorithm. However, these results cannot directly imply the no-regret guarantee in our online learning setting, due to the known fact that posterior sampling can perform poorly under potentially *adversarial* or *non-stationary* environments (Zimmert & Seldin, 2021; Liu et al., 2023a). In contrast, we here establish the equivalence of the pre-trained LLM to the FTPL algorithm (under different pre-training data distribution specifications), with the ability to handle arbitrary loss sequences, even though the LLMs are only trained on a fixed/stationary distribution of texts (tasks).

### F.10. How Well Can Cur Hypothetical Model Predict Actual LLMs' Behaviors?

To further verify our theoretically-justified model in Theorem 4.1, we propose to *estimate* the parameters of $\{\eta_t\}_{t=0}^{T-1}$ in Definition F.2 using the interaction data with actual LLMs, and use the estimated model to predict LLMs' behaviors on some test set. In Figure F.1, we show the averaged regret for the LLMs and our estimated model, where the generalized quantal response can *very well capture* the behavior of the LLM agent for all problem instances in Section 3.2, on which the LLMs are oftentimes no-regret, justifying the applicability of our hypothetical model and assumptions.

### F.10.1. DETAILS OF ESTIMATING THE PARAMETERS OF OUR HYPOTHETICAL MODEL

To further validate our model and data distribution assumptions, we also propose to estimate the parameter $\{\eta_t\}_{t\in[T-1]}$ in Definition F.2, using data from interacting with LLMs (following the same protocol as before), with $P_{noise}$ being a standard normal distribution (note that we do not need to estimate $\eta_0$ by Definition F.2). Specifically, given $n$ episodes of the LLM

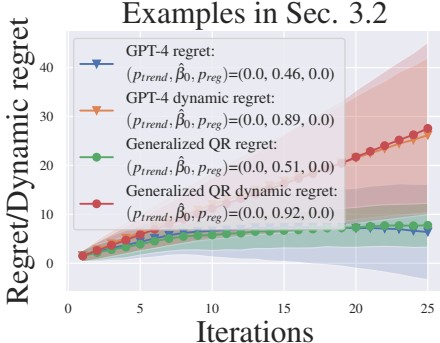

*Figure F.1.* Comparison of GPT-4 with the generalized QR model, where the model can very well capture the behavior of the GPT-4 agent for examples in Section 3.2.

agent's behavior $\{(\ell_t^{(j)}, \pi_t^{(j)})_{t\in[T]}\}_{j\in[n]}$, motivated by our Lemma 1 and Theorem 4.1, we estimate $\{\eta_t\}_{t\in[T-1]}$ by solving the following problem

$$\sigma^\star \in \arg\min_{\sigma>0} \quad \sum_{t\in[T-1]}\sum_{j\in[n]} \left\| \pi_{t+1}^{(j)} - P_{quantal}^{\sigma\sqrt{t+1}}\left(\cdot \mid \ell_{1:t}^{(j)}\right) \right\|_1, \qquad \eta_t^\star = \sigma^\star\sqrt{t+1}, \quad \forall t\in[T-1].$$

We solve this single-variable optimization problem by grid search over $[0, 10]$. We then run the generalized quantal response model with the estimated $\{\eta_t^\star\}_{t\in[T-1]}$ on another *unseen test set*, and compare it with the behavior of the actual LLM agents. We use all the interaction data from Section 3.2 and split it in half for training and testing.

We also use the same framework to understand the regrettable behaviors in Section 3.4. This analysis uses all the data from Section 3.4. We first find that such fitting procedures do not yield good predictions for LLMs on those counter-examples. Therefore, we resort to a more expressive model by directly fitting each $\eta_t$ as

$$\eta_t^\star \in \arg\min_{\eta_t>0} \sum_{j\in[n]} \left\| \pi_{t+1}^{(j)} - P_{quantal}^{\eta_t}\left(\cdot \mid \ell_{1:t}^{(j)}\right) \right\|_1$$

separately for each $t\in[T-1]$. Even under the expressive model, LLMs fail to follow the generalized quantal response for the counter-examples with noisy alternating or adaptive loss sequences, as Figure F.1 shows the gap between GPT-4 (dynamic) regret and the our model's (dynamic) regret.

## G. Deferred Results and Proofs in Section 5

### G.1. Regularity Conditions on $f$ and $h$

$h : \mathbb{R} \to \mathbb{R}^+$ is a continuous function, with continuous derivative $h'$, and $f(\cdot, k) : \mathbb{R} \to \mathbb{R}^+$ is a continuous function for each $k \in \mathbb{N}^+$, satisfying $\lim_{k \to \infty} \frac{f(R_1, k)}{f(R_2, k)} = \infty \cdot \mathbb{1}(R_1 > R_2) + \mathbb{1}(R_1 = R_2)$, where we use the convention of $\infty \cdot 0 = 0$. These conditions on $h, f$ will be assumed throughout the paper. Examples of such an $f$ include $f(x, k) = x^k$ and $\exp(kx)$.

**Additional regularity conditions for Theorem 5.1.** For any $k \in \mathbb{N}^+$, $h, f(\cdot, k)$ are non-decreasing, and $\log f$ is a supermodular function (i.e., $\log f(R_1, k_1) - \log f(R_1, k_2) \geq \log f(R_2, k_1) - \log f(R_2, k_2)$ for $R_1 \geq R_2$ and $k_1 \geq k_2$)

### G.2. Basic Lemmas

**Lemma 5** (Double iterated limit). *For a sequence $(a_{mn})_{m,n \in \mathbb{N}^+}$, suppose that $\lim_{m,n \to \infty} a_{mn} = L$. Then the following are equivalent:*

- *For each $m$, $\lim_{n \to \infty} a_{mn}$ exists;*

- *$\lim_{m \to \infty} \lim_{n \to \infty} a_{mn} = L$.*

**Lemma 6** (Hoeffding's inequality). *Let $X_1, X_2, \ldots, X_n$ be independent random variables bounded by the intervals $[a_i, b_i]$, respectively. Define $\bar{X} = \frac{1}{n} \sum_{i=1}^n X_i$ and let $\mu = \mathbb{E}[\bar{X}]$ be the expected value of $\bar{X}$. Then, for any $t > 0$,*

$$\mathbb{P}(|\bar{X} - \mu| \geq t) \leq 2 \exp\left(-\frac{2n^2 t^2}{\sum_{i=1}^n (b_i - a_i)^2}\right).$$

**Lemma 7** (Uniform convergence $\implies$ Interchanging limit and infimum). *If $(f_n : X \to \mathbb{R})_{n \in \mathbb{N}^+}$ is a sequence of continuous functions that uniformly converge to a function $f : X \to \mathbb{R}$ on the domain $X$, then $\lim_{n \to \infty} \inf_{x \in X} f_n(x) = \inf_{x \in X} f(x)$ holds.*

### G.3. Deferred Proof for the Arguments in Section 5.1

In this section, we prove some properties of $\mathcal{L}(\theta, k, N)$ under certain regularity conditions of $f, h$. Throughout this subsection, we will assume the following condition holds.

**Condition 1.** *For $h : \mathbb{R} \to \mathbb{R}^+$ and $f : \mathbb{R} \times \mathbb{N}^+ \to \mathbb{R}^+$, suppose $h(\cdot)$ and $f(\cdot, k)$ are both continuous and non-decreasing functions for any $k \in \mathbb{N}^+$. The derivative $h' : \mathbb{R} \to \mathbb{R}$ is also a continuous function. Moreover, $f$ satisfies that $\log f(R_1, k_1) - \log f(R_1, k_2) \geq \log f(R_2, k_1) - \log f(R_2, k_2)$ for $R_1 \geq R_2$ and $k_1 \geq k_2$, i.e., $\log f$ is supermodular. Lastly, $f$ is a function such that $\lim_{k \to \infty} \frac{f(R_1, k)}{f(R_2, k)} = \infty \cdot \mathbb{1}(R_1 > R_2) + \mathbb{1}(R_1 = R_2)$, with the convention of $\infty \cdot 0 = 0$. Lastly, $(\ell_t^{(j)})_{t \in [T], j \in [N]}$ are continuous random variables supported on $[-B, B]^{T \times N}$.*

**Claim 1** (Iterated limit of $\mathcal{L}(\theta, k, N)$ is the same as double limit of $\mathcal{L}(\theta, k, N)$). *It holds that:*

$$\lim_{N \to \infty} \lim_{k \to \infty} \mathcal{L}(\theta, k, N) = \lim_{N, k \to \infty} \mathcal{L}(\theta, k, N) = \lim_{k \to \infty} \lim_{N \to \infty} \mathcal{L}(\theta, k, N) = h\left(\max_{\ell_1, \ldots, \ell_T} \text{Regret}_{\text{LLM}_\theta}((\ell_t)_{t \in [T]})\right).$$

*Proof.* **Step 1.** **Proving** $\lim_{N \to \infty} \lim_{k \to \infty} \mathcal{L}(\theta, k, N) = h\left(\max_{\ell_1, \ldots, \ell_T} \text{Regret}_{\text{LLM}_\theta}((\ell_t)_{t \in [T]})\right)$.

Firstly, as both $h$ and $f$ are non-negative (Condition 1), and $\mathbb{E}_{(\ell_t^{(j)})_{t \in [T], j \in [N]}}\left[h(\max_{j \in [N]} \text{Regret}_{\text{LLM}_\theta}((\ell_t^{(j)})_{t \in [T]}))\right]$ exists, we have by dominated convergence theorem that

$$\lim_{k \to \infty} \mathcal{L}(\theta, k, N) = \mathbb{E} \lim_{k \to \infty} \left[\frac{\sum_{j \in [N]} h(R_{\text{LLM}_\theta}((\ell_t^{(j)})_{t \in [T]})) f(R_{\text{LLM}_\theta}((\ell_t^{(j)})_{t \in [T]}), k)}{\sum_{j \in [N]} f(R_{\text{LLM}_\theta}((\ell_i^{(j)})_{t \in [T]}), k)}\right]$$

$$= \mathbb{E}_{(\ell_t^{(j)})_{t \in [T], j \in [N]}} \left[h(\max_{j \in [N]} R_{\text{LLM}_\theta}((\ell_t^{(j)})_{t \in [T]}))\right]$$

where $R_{\text{LLM}_\theta}$ denotes an abbreviation of $\text{Regret}_{\text{LLM}_\theta}$. By **(?)**Chapter 11]ahsanullah2013introduction, we have $h(\max_{j \in [N]} \text{Regret}_{\text{LLM}_\theta}((\ell_t^{(j)})_{t \in [T]})) \xrightarrow{p} h(\max_{\ell_1, \ldots, \ell_T} \text{Regret}_{\text{LLM}_\theta}((\ell_t)_{t \in [T]}))$ when $N \to \infty$. Hence, we have $\lim_{N \to \infty} \lim_{k \to \infty} \mathcal{L}(\theta, k, N) = h(\max_{\ell_1, \ldots, \ell_T} \text{Regret}_{\text{LLM}_\theta}((\ell_t)_{t \in [T]}))$ holds.

**Step 2. Proving** $\lim_{N,k \to \infty} \mathcal{L}(\theta, k, N) = h(\max_{\ell_1, \ldots, \ell_T} \text{Regret}_{\text{LLM}_\theta}((\ell_t)_{t \in [T]}))$.

Now, we will calculate $\lim_{N,k \to \infty} \mathcal{L}(\theta, k, N)$.

**Lemma 8.** *For any $0 < \epsilon < 1$, it follows that*

$$\lim_{N,k \to \infty} \frac{\sum_{i=1}^{N} f(X_i, k) H(X_i) \mathbb{1}(H(X_i) < 1 - \epsilon)}{\sum_{i=1}^{N} f(X_i, k) H(X_i) \mathbb{1}(H(X_i) > 1 - \epsilon/2)} = 0$$

*and*

$$\lim_{N,k \to \infty} \frac{\sum_{i=1}^{N} f(X_i, k) \mathbb{1}(H(X_i) < 1 - \epsilon)}{\sum_{i=1}^{N} f(X_i, k) \mathbb{1}(H(X_i) > 1 - \epsilon/2)} = 0$$

*hold with probability* 1, *where $X_i$'s are i.i.d. random variables,* $\text{esssup}(H(X_i)) = 1$, *and $H : \mathbb{R} \to \mathbb{R}^+$ is a continuous non-decreasing function.*

*Proof of Lemma 8.* Since $f(\cdot, k), H$ are non-negative and non-decreasing functions, we have

$$\frac{\sum_{i=1}^{N} f(X_i, k) H(X_i) \mathbb{1}(H(X_i) < 1 - \epsilon)}{\sum_{i=1}^{N} f(X_i, k) H(X_i) \mathbb{1}(H(X_i) > 1 - \epsilon/2)} \leq \frac{(1 - \epsilon) f(H^{-1}(1 - \epsilon), k) |\{i \in [N] \mid (H(X_i) < 1 - \epsilon)\}|}{(1 - \epsilon/2) f(H^{-1}(1 - \epsilon/2), k) |\{i \in [N] \mid (H(X_i) > 1 - \epsilon/2)\}|}$$

and we know that

$$\frac{|\{i \in [N] \mid (H(X_i) < 1 - \epsilon)\}|}{|\{i \in [N] \mid (H(X_i) > 1 - \epsilon/2)\}|} \xrightarrow{a.s.} \frac{F(1 - \epsilon)}{1 - F(1 - \epsilon/2)}$$

as $N \to \infty$, where $F$ is the cumulative distribution function of random variable $H(X)$. Therefore, we have

$$0 \leq \lim_{N,k \to \infty} \frac{\sum_{i=1}^{N} f(X_i, k) H(X_i) \mathbb{1}(H(X_i) < 1 - \epsilon)}{\sum_{i=1}^{N} f(X_i, k) H(X_i) \mathbb{1}(H(X_i) > 1 - \epsilon/2)} \leq \lim_{N,k \to \infty} \frac{(1 - \epsilon) f(H^{-1}(1 - \epsilon), k)) |\{i \in [N] \mid (H(X_i) < 1 - \epsilon)\}|}{(1 - \epsilon/2) f(H^{-1}(1 - \epsilon/2), k)) |\{i \in [N] \mid (H(X_i) > 1 - \epsilon/2)\}|}$$

$$\underset{a.s.}{\leq} \lim_{N,k \to \infty} \frac{(1 - \epsilon) f(H^{-1}(1 - \epsilon), k))}{(1 - \epsilon/2) f(H^{-1}(1 - \epsilon/2), k))} \frac{F(1 - \epsilon)}{1 - F(1 - \epsilon/2)} = 0.$$

By a similar argument, we have

$$\lim_{N,k \to \infty} \frac{\sum_{i=1}^{N} f(X_i, k) \mathbb{1}(H(X_i) < 1 - \epsilon)}{\sum_{i=1}^{N} f(X_i, k) \mathbb{1}(H(X_i) > 1 - \epsilon/2)} = 0$$

with probability 1. $\qquad \square$

One key idea in the proof above is the use of some *truncation* level $\epsilon$ for $H(X)$ with $\text{esssup}(H(X)) = 1$. By Lemma 8, we have

$$\lim_{N,k \to \infty} \frac{\sum_{i=1}^{N} f(X_i, k) H(X_i) \mathbb{1}(H(X_i) > 1 - \epsilon)}{\sum_{i=1}^{N} f(X_i, k) H(X_i)} = \lim_{N,k \to \infty} \frac{\sum_{i=1}^{N} f(X_i, k) \mathbb{1}(H(X_i) > 1 - \epsilon)}{\sum_{i=1}^{N} f(X_i, k)} = 1,$$

since

$$0 \leq \frac{\sum_{i=1}^{N} f(X_i, k) \mathbb{1}(H(X_i) < 1 - \epsilon)}{\sum_{i=1}^{N} f(X_i, k)} \leq \frac{\sum_{i=1}^{N} f(X_i, k) \mathbb{1}(H(X_i) < 1 - \epsilon)}{\sum_{i=1}^{N} f(X_i, k) \mathbb{1}(H(X_i) > 1 - \epsilon/2)}$$

holds with probability 1. Therefore, for any $0 < \epsilon < 1$, we have

$$\lim_{N,k\to\infty} \mathcal{L}(\theta, k, N) = \mathbb{E} \lim_{N,k\to\infty} \left[ \frac{\sum_{j\in[N]} h(R_{\text{LLM}_\theta}((\ell_t^{(j)})_{t\in[T]})) f(R_{\text{LLM}_\theta}((\ell_t^{(j)})_{t\in[T]}), k)}{\sum_{j\in[N]} f(R_{\text{LLM}_\theta}((\ell_i^{(j)})_{t\in[T]}), k)} \right]$$

$$= h\left( \max_{\ell_1,\dots,\ell_T} R_{\text{LLM}_\theta}((\ell_t)_{t\in[T]}) \right)$$

$$\times \mathbb{E} \lim_{N,k\to\infty} \left[ \frac{\sum_{j\in[N]} \frac{h(R_{\text{LLM}_\theta}((\ell_t^{(j)})_{t\in[T]}))}{h(\max_{\ell_1,\dots,\ell_T} R_{\text{LLM}_\theta}((\ell_t)_{t\in[T]}))} f(R_{\text{LLM}_\theta}((\ell_t^{(j)})_{t\in[T]}), k) \mathbb{1}\left( \frac{h(R_{\text{LLM}_\theta}((\ell_t^{(j)})_{t\in[T]}))}{h(\max_{\ell_1,\dots,\ell_T} R_{\text{LLM}_\theta}((\ell_t)_{t\in[T]}))} > 1-\epsilon \right)}{\sum_{j\in[N]} f(R_{\text{LLM}_\theta}((\ell_i^{(j)})_{t\in[T]}), k) \mathbb{1}\left( \frac{h(R_{\text{LLM}_\theta}((\ell_t^{(j)})_{t\in[T]}))}{h(\max_{\ell_1,\dots,\ell_T} R_{\text{LLM}_\theta}((\ell_t)_{t\in[T]}))} > 1-\epsilon \right)} \right]$$

$$\geq (1-\epsilon) h(\max_{\ell_1,\dots,\ell_T} R_{\text{LLM}_\theta}((\ell_t)_{t\in[T]}))$$

which implies $\lim_{N,k\to\infty} \mathcal{L}(\theta, k, N) = h(\max_{\ell_1,\dots,\ell_T} \text{Regret}_{\text{LLM}_\theta}((\ell_t)_{t\in[T]}))$ since

$$\mathcal{L}(\theta, k, N) \leq h\left( \max_{\ell_1,\dots,\ell_T} \text{Regret}_{\text{LLM}_\theta}((\ell_t)_{t\in[T]}) \right)$$

by definition of $\mathcal{L}$, the fact that $h$ is non-decreasing, and by setting $\epsilon \to 0$ to obtain

$$\mathcal{L}(\theta, k, N) \geq h\left( \max_{\ell_1,\dots,\ell_T} \text{Regret}_{\text{LLM}_\theta}((\ell_t)_{t\in[T]}) \right).$$

Here, we used the fact that $(\ell_t)_{t\in[T]}$ has a continuous distribution, $\text{Regret}_{\text{LLM}_\theta}((\ell_t)_{t\in[T]})$ is a continuous function, and the non-decreasing property and continuity of $h$ (Condition 1), which lead to:

$$\text{esssup}\left( h\left( \text{Regret}_{\text{LLM}_\theta}((\ell_t)_{t\in[T]}) \right) \right) = \max_{\ell_1,\dots,\ell_T} h\left( \text{Regret}_{\text{LLM}_\theta}((\ell_t)_{t\in[T]}) \right) = h\left( \max_{\ell_1,\dots,\ell_T} \text{Regret}_{\text{LLM}_\theta}((\ell_t)_{t\in[T]}) \right). \quad \text{(G.1)}$$

Equation (G.1) will be used frequently in the overall proof in Appendix G.3.

**Step 3.** **Proving** $\lim_{k\to\infty} \lim_{N\to\infty} \mathcal{L}(\theta, k, N) = h\left( \max_{\ell_1,\dots,\ell_T} \text{Regret}_{\text{LLM}_\theta}((\ell_t)_{t\in[T]}) \right)$.

Lastly, if $N \to \infty$, similarly by dominated convergence theorem we have

$$\lim_{N\to\infty} \mathcal{L}(\theta, k, N) = \mathbb{E} \lim_{N\to\infty} \left[ \frac{\sum_{j\in[N]} h\left( R_{\text{LLM}_\theta}\left( (\ell_t^{(j)})_{t\in[T]} \right) \right) f(R_{\text{LLM}_\theta}((\ell_t^{(j)})_{t\in[T]}), k)}{\sum_{j\in[N]} f\left( R_{\text{LLM}_\theta}\left( (\ell_i^{(j)})_{t\in[T]} \right), k \right)} \right]$$

$$= \frac{\mathbb{E}\left[ h\left( R_{\text{LLM}_\theta}\left( (\ell_t^{(j)})_{t\in[T]} \right) \right) f\left( R_{\text{LLM}_\theta}\left( (\ell_t^{(j)})_{t\in[T]} \right), k \right) \right]}{\mathbb{E}\left[ f\left( R_{\text{LLM}_\theta}\left( (\ell_i^{(j)})_{t\in[T]} \right), k \right) \right]}.$$

Thus, $\lim_{N\to\infty} \mathcal{L}(\theta, k, N)$ always exists for every $k$. Now, we use the known property of double iterated limit (Lemma 5), and obtain that $\lim_{k\to\infty} \lim_{N\to\infty} \mathcal{L}(\theta, k, N) = h(\max_{\ell_1,\dots,\ell_T} \text{Regret}_{\text{LLM}_\theta}((\ell_t)_{t\in[T]}))$. $\qquad\square$

**Claim 2** (Uniform convergence of $\mathcal{L}(\theta, k, N)$ (with respect to $k$ and $N$))**.** $\mathcal{L}(\theta, k, N)$ *uniformly converges to* $h(\max_{\ell_1,\dots,\ell_T} \text{Regret}_{\text{LLM}_\theta}((\ell_t)_{t\in[T]}))$ *on the domain* $\Theta$.

*Proof.* We will provide a similar analysis as Lemma 8 as follows:

**Lemma 9.** *For any $0 < \epsilon < 1$, $0 < \delta < 1$, and $k \in \mathbb{N}^+$, we have*

$$\frac{\sum_{i=1}^N f(X_i, k) \mathbb{1}(H(X_i) < 1-\epsilon)}{\sum_{i=1}^N f(X_i, k) \mathbb{1}(H(X_i) > 1-\epsilon)} = \widetilde{\mathcal{O}}\left( A(k, H, \epsilon) \left( \frac{1}{1 - F_{H,X}(1-\epsilon/2)} + \frac{1}{\sqrt{N}} \right) \right)$$

*with probability at least $1 - \delta$, where $X_i$'s are i.i.d. random variables, $\text{esssup}(H(X_i)) = 1$, $H : \mathbb{R} \to \mathbb{R}^+$ is a continuous non-decreasing function, $A(k, t, \epsilon) := \frac{(1-\epsilon)f((t/\text{esssup}(t(X)))^{-1}(1-\epsilon), k)}{(1-\epsilon/2)f((t/\text{esssup}(t(X)))^{-1}(1-\epsilon/2), k)}$, for any non-decreasing function $t : \mathbb{R} \to \mathbb{R}^+$, and $F_{t,X}$ is a cumulative distribution function of random variable $t(X)/\text{esssup}(t(X))$.*

*Proof of Lemma 9.* With the same argument as the proof of Lemma 8, we have

$$\frac{\sum_{i=1}^{N} f(X_i, k)\mathbb{1}(H(X_i) < 1 - \epsilon)}{\sum_{i=1}^{N} f(X_i, k)\mathbb{1}(H(X_i) > 1 - \epsilon/2)} \le \frac{f(H^{-1}(1-\epsilon), k)|\{i \in [N] \mid (H(X_i) < 1 - \epsilon)\}|}{f(H^{-1}(1-\epsilon/2), k)|\{i \in [N] \mid (H(X_i) > 1 - \epsilon/2)\}|}.$$

It holds that $\frac{1}{N}|\{i \in [N] \mid (H(X_i) < 1 - \epsilon)\}| = F_{H,X}(1-\epsilon) + \widetilde{\mathcal{O}}(1/\sqrt{N})$ with probability at least $1 - \delta/2$ due to Hoeffding's inequality (Lemma 6). Similarly, we have $\frac{1}{N}|\{i \in [N] \mid (H(X_i) > 1-\epsilon/2)\}| = 1 - F_{H,X}(1-\epsilon/2) + \widetilde{\mathcal{O}}(1/\sqrt{N})$ with probability at least $1 - \delta/2$. Therefore,

$$\frac{|\{i \in [N] \mid (H(X_i) < 1 - \epsilon)\}|}{|\{i \in [N] \mid (H(X_i) > 1 - \epsilon/2)\}|} = \frac{F_{H,X}(1-\epsilon)}{1 - F_{H,X}(1-\epsilon/2)} + \widetilde{\mathcal{O}}(\sqrt{1/N}) \le \frac{1}{1 - F_{H,X}(1-\epsilon/2)} + \widetilde{\mathcal{O}}(\sqrt{1/N}),$$

with probability at least $1 - \delta$. Finally, we have

$$\frac{\sum_{i=1}^{N} f(X_i, k)\mathbb{1}(H(X_i) < 1 - \epsilon)}{\sum_{i=1}^{N} f(X_i, k)\mathbb{1}(H(X_i) > 1 - \epsilon)} < \frac{\sum_{i=1}^{N} f(X_i, k)\mathbb{1}(H(X_i) < 1 - \epsilon)}{\sum_{i=1}^{N} f(X_i, k)\mathbb{1}(H(X_i) > 1 - \epsilon/2)} \le A(k, H, \epsilon)\left(\frac{1}{1 - F_{H,X}(1-\epsilon/2)} + \widetilde{\mathcal{O}}\left(\frac{1}{\sqrt{N}}\right)\right).$$

$\square$

Note that $\lim_{k \to \infty} A(k, H, \epsilon) = 0$, since $\lim_{k \to \infty} \frac{f(R_1, k)}{f(R_2, k)} = \infty \cdot \mathbb{1}(R_1 > R_2) + \mathbb{1}(R_1 = R_2)$. By Lemma 9 with $H(R_{\text{LLM}_\theta}((\ell_t)_{t \in [T]})) = \frac{h(R_{\text{LLM}_\theta}((\ell_t)_{t \in [T]}))}{h(\max_{\ell_1, \dots, \ell_T} R_{\text{LLM}_\theta}((\ell_t)_{t \in [T]}))}$, we have

$$\frac{\sum_{i=1}^{N} f(R_{\text{LLM}_\theta}((\ell_t^{(i)})_{t \in [T]}), k)\mathbb{1}\left(\frac{h(R_{\text{LLM}_\theta}((\ell_t^{(i)})_{t \in [T]}))}{h(\max_{\ell_1, \dots, \ell_T} R_{\text{LLM}_\theta}((\ell_t)_{t \in [T]}))} \ge 1 - \epsilon\right)}{\sum_{i=1}^{N} f(R_{\text{LLM}_\theta}((\ell_t^{(i)})_{t \in [T]}), k)}$$

$$= \frac{1}{1 + \frac{\sum_{i=1}^{N} f(R_{\text{LLM}_\theta}((\ell_t^{(i)})_{t \in [T]}), k)\mathbb{1}\left(\frac{h(R_{\text{LLM}_\theta}((\ell_t^{(i)})_{t \in [T]}))}{h(\max_{\ell_1, \dots, \ell_T} R_{\text{LLM}_\theta}((\ell_t)_{t \in [T]}))} < 1-\epsilon\right)}{\sum_{i=1}^{N} f(R_{\text{LLM}_\theta}((\ell_t^{(i)})_{t \in [T]}), k)\mathbb{1}\left(\frac{h(R_{\text{LLM}_\theta}((\ell_t^{(i)})_{t \in [T]}))}{h(\max_{\ell_1, \dots, \ell_T} R_{\text{LLM}_\theta}((\ell_t)_{t \in [T]}))} \ge 1-\epsilon\right)}} \ge \frac{1}{1 + A(k, H, \epsilon)(\frac{1}{1 - F_{H, R_{\text{LLM}_\theta}((\ell_t)_{t \in [T]})}(1-\epsilon/2)} + \widetilde{\mathcal{O}}(\sqrt{1/N}))},$$

where we recall the shorthand notation of $R_{\text{LLM}_\theta} = \text{Regret}_{\text{LLM}_\theta}$. Note that $A(k, H, \epsilon) = A(k, h, \epsilon)$ and $F_{H, R_{\text{LLM}_\theta}} = F_{h, R_{\text{LLM}_\theta}}$ hold by the definitions of $F_{t, X}$ and $A(k, t, \epsilon)$ in Lemma 9. Therefore,

$$1 \ge \frac{\sum_{i=1}^{N} f(R_{\text{LLM}_\theta}((\ell_t^{(i)})_{t \in [T]}), k)\frac{h(R_{\text{LLM}_\theta}((\ell_t^{(i)})_{t \in [T]}))}{h(\max_{\ell_1, \dots, \ell_T} R_{\text{LLM}_\theta}((\ell_t)_{t \in [T]}))}}{\sum_{i=1}^{N} f(R_{\text{LLM}_\theta}((\ell_t^{(i)})_{t \in [T]}), k)}$$

$$\ge \frac{\sum_{i=1}^{N} f(R_{\text{LLM}_\theta}((\ell_t^{(i)})_{t \in [T]}), k)\frac{h(R_{\text{LLM}_\theta}((\ell_t^{(i)})_{t \in [T]}))}{h(\max_{\ell_1, \dots, \ell_T} R_{\text{LLM}_\theta}((\ell_t)_{t \in [T]}))}\mathbb{1}\left(\frac{h(R_{\text{LLM}_\theta}((\ell_t^{(i)})_{t \in [T]}))}{h(\max_{\ell_1, \dots, \ell_T} R_{\text{LLM}_\theta}((\ell_t)_{t \in [T]}))} \ge 1 - \epsilon\right)}{\sum_{i=1}^{N} f(R_{\text{LLM}_\theta}((\ell_t^{(i)})_{t \in [T]}), k)\mathbb{1}\left(\frac{h(R_{\text{LLM}_\theta}((\ell_t^{(i)})_{t \in [T]}))}{h(\max_{\ell_1, \dots, \ell_T} R_{\text{LLM}_\theta}((\ell_t)_{t \in [T]}))} \ge 1 - \epsilon\right)}$$

$$\times \frac{1}{1 + A(k, h, \epsilon)(\frac{1}{1 - F_{h, R_{\text{LLM}_\theta}((\ell_t)_{t \in [T]})}(1-\epsilon/2)} + \widetilde{\mathcal{O}}(\sqrt{1/N}))}$$

$$\ge \frac{1 - \epsilon}{1 + A(k, h, \epsilon)(\frac{1}{1 - F_{h, R_{\text{LLM}_\theta}((\ell_t)_{t \in [T]})}(1-\epsilon/2)} + \widetilde{\mathcal{O}}(\sqrt{1/N}))}$$

with probability at least $1 - \delta$.

Now, for any $\epsilon > 0$ and $\delta > 0$, we have

$$0 \le h\left(\max_{\ell_1, \dots, \ell_T} R_{\text{LLM}_\theta}((\ell_t)_{t \in [T]})\right) - \mathcal{L}(\theta, k, N)$$

$$\le h\left(\max_{\ell_1, \dots, \ell_T} R_{\text{LLM}_\theta}((\ell_t)_{t \in [T]})\right)\left(1 - \frac{(1 - \delta)(1 - \epsilon)}{1 + A(k, h, \epsilon)(\frac{1}{1 - F_{h, R_{\text{LLM}_\theta}((\ell_t)_{t \in [T]})}(1-\epsilon/2)} + \widetilde{\mathcal{O}}(\sqrt{1/N}))}\right).$$

Note that

$$1 - F_{h, R_{\mathrm{LLM}_\theta}((\ell_t)_{t \in [T]})}(1 - \epsilon/2) = \mathbb{P}\left( h\left(\mathrm{Regret}_{\mathrm{LLM}_\theta}\left((\ell_t)_{t \in [T]}\right)\right) > (1 - \epsilon/2)h\left(\max_{\ell_1, \dots, \ell_T} \mathrm{Regret}_{\mathrm{LLM}_\theta}\left((\ell_t)_{t \in [T]}\right)\right)\right)$$

is a continuous function of $\theta$, since we assume $\mathrm{LLM}_\theta$ is a continuous function of $\theta$, $(\ell_t)_{t \in [T]}$ has a continuous distribution, and $\mathrm{Regret}_{\mathrm{LLM}_\theta}((\ell_t)_{t \in [T]})$ is a continuous function of $\mathrm{LLM}_\theta$ and $(\ell_t)_{t \in [T]}$. Since we consider a compact $\Theta$ (as several recent works on analyzing Transformers (Bai et al., 2023; Lin et al., 2024)), we have $p(\epsilon) := \min_{\theta \in \Theta} 1 - F_{h, R_{\mathrm{LLM}_\theta}((\ell_t)_{t \in [T]})}(1 - \epsilon/2) > 0$. Therefore,

$$\left(1 - \frac{(1-\delta)(1-\epsilon)}{1 + A(k, h, \epsilon)(\frac{1}{1 - F_{h, R_{\mathrm{LLM}_\theta}}(1 - \epsilon/2)} + \widetilde{\mathcal{O}}(\sqrt{1/N}))}\right) \leq \left(1 - \frac{(1-\delta)(1-\epsilon)}{1 + A(k, h, \epsilon)(\frac{1}{p(\epsilon)} + \widetilde{\mathcal{O}}(\sqrt{1/N}))}\right), \qquad \text{(G.2)}$$

and we know that $\lim_{N, k \to \infty} 1 + A(k, h, \epsilon)(\frac{1}{p(\epsilon)} + \widetilde{\mathcal{O}}(\sqrt{1/N})) = 1$, which is not dependent on $\theta$. Thus, we can conclude that $\lim_{N, k \to \infty} \sup_{\theta \in \Theta} |h(\max_{\ell_1, \dots, \ell_T} \mathrm{Regret}_{\mathrm{LLM}_\theta}((\ell_t)_{t \in [T]})) - \mathcal{L}(\theta, k, N)| = 0$, as we can choose arbitrarily small $\epsilon, \delta$. $\square$

**Claim 3** (Double iterated limit of supremum). *It holds that:*

$$\lim_{N \to \infty} \lim_{k \to \infty} \sup_{\theta \in \Theta} \left| \mathcal{L}(\theta, k, N) - h\left(\max_{\ell_1, \dots, \ell_T} \mathrm{Regret}_{LLM_\theta}((\ell_t)_{t \in [T]})\right)\right| = 0.$$

*Proof.* Since $h(\max_{\ell_1, \dots, \ell_T} \mathrm{Regret}_{\mathrm{LLM}_\theta}((\ell_t)_{t \in [T]})) \geq \mathcal{L}(\theta, k, N)$, we will prove

$$\lim_{N \to \infty} \lim_{k \to \infty} \sup_{\theta \in \Theta} h\left(\max_{\ell_1, \dots, \ell_T} \mathrm{Regret}_{\mathrm{LLM}_\theta}((\ell_t)_{t \in [T]})\right) - \mathcal{L}(\theta, k, N) = 0.$$

**Lemma 10.** $\frac{\sum_{i=1}^N f(X_i, k_1)h(X_i)}{\sum_{i=1}^N f(X_i, k_1)} \leq \frac{\sum_{i=1}^N f(X_i, k_2)h(X_i)}{\sum_{i=1}^N f(X_i, k_2)}$ *holds if* $0 < k_1 \leq k_2$ *for any real-valued* $(X_i)_{i \in [N]}$.

*Proof.* By multiplying $(\sum_{i=1}^N f(X_i, k_1))(\sum_{i=1}^N f(X_i, k_2))$ on both sides of the formula, we know that it is equivalent to $\sum_{1 \leq i \neq j \leq N} f(X_i, k_1)h(X_i)f(X_j, k_2) \leq \sum_{1 \leq i \neq j \leq N} f(X_i, k_1)h(X_j)f(X_j, k_2)$. This is equivalent to

$$\sum_{1 \leq i \neq j \leq N} (f(X_i, k_1)f(X_j, k_2) - f(X_j, k_1)f(X_i, k_2))(h(X_i) - h(X_j)) \leq 0,$$

which is true since if $X_i \geq X_j$, $(f(X_i, k_1)f(X_j, k_2) - f(X_j, k_1)f(X_i, k_2)) \leq 0$ due to the log-increasing difference of $f$ (Condition 1), as $\log f(X_j, k_1) - \log f(X_j, k_2) \geq \log f(X_i, k_1) - \log f(X_i, k_2)$ if $X_i \geq X_j$. $\square$

Therefore, $\mathcal{L}(\theta, k, N)$ is a non-decreasing function of $k$ if $N$ is fixed, which indicates that

$$\lim_{k \to \infty} \sup_{\theta \in \Theta} h\left(\max_{\ell_1, \dots, \ell_T} \mathrm{Regret}_{\mathrm{LLM}_\theta}((\ell_t)_{t \in [T]})\right) - \mathcal{L}(\theta, k, N)$$

exists, as $\mathcal{L}(\theta, k, N)$ is also bounded. Therefore, by Lemma 5 and Claim 2, we know that

$$\lim_{N \to \infty} \lim_{k \to \infty} \sup_{\theta \in \Theta} \left| \mathcal{L}(\theta, k, N) - h\left(\max_{\ell_1, \dots, \ell_T} \mathrm{Regret}_{\mathrm{LLM}_\theta}((\ell_t)_{t \in [T]})\right)\right|$$

exists and this value should be 0. $\square$

**Claim 4.** *It holds that*

$$\lim_{N, k \to \infty} \inf_{\theta \in \Theta} \mathcal{L}(\theta, k, N) = \lim_{N \to \infty} \lim_{k \to \infty} \inf_{\theta \in \Theta} \mathcal{L}(\theta, k, N) = \inf_{\theta \in \Theta} h\left(\max_{\ell_1, \dots, \ell_T} \mathrm{Regret}_{LLM_\theta}((\ell_t)_{t \in [T]})\right).$$

*Proof.* Firstly, by Lemma 7, we have $\lim_{N, k \to \infty} \inf_{\theta \in \Theta} \mathcal{L}(\theta, k, N) = \inf_{\theta \in \Theta} h(\max_{\ell_1, \dots, \ell_T} \mathrm{Regret}_{\mathrm{LLM}_\theta}((\ell_t)_{t \in [T]}))$. Plus, we already know that $\mathcal{L}(\theta, k, N)$ is a monotonically non-decreasing function of $k$ for any fixed $N$ (Lemma 10), and it is bounded, $\lim_{k \to \infty} \inf_{\theta \in \Theta} \mathcal{L}(\theta, k, N)$ always exists. Therefore, by Lemma 5, we also have $\lim_{N \to \infty} \lim_{k \to \infty} \inf_{\theta \in \Theta} \mathcal{L}(\theta, k, N) = \inf_{\theta \in \Theta} h(\max_{\ell_1, \dots, \ell_T} \mathrm{Regret}_{\mathrm{LLM}_\theta}((\ell_t)_{t \in [T]}))$. $\square$

### G.4. Definition of the Empirical Loss Function

**Definition G.1** (Empirical loss function). *We define the empirical loss $\widehat{\mathcal{L}}$ computed with $N_T$ samples as follows:*

$$\widehat{\mathcal{L}}(\theta, k, N, N_T) := \frac{1}{N_T} \sum_{s=1}^{N_T} \left[ \frac{\sum_{j\in[N]} h\left(\text{Regret}_{\text{LLM}_\theta}((\ell_{s,t}^{(j)})_{t\in[T]})\right) f\left(\text{Regret}_{\text{LLM}_\theta}((\ell_{s,t}^{(j)})_{t\in[T]}), k\right)}{\sum_{j\in[N]} f\left(\text{Regret}_{\text{LLM}_\theta}((\ell_{s,t}^{(j)})_{t\in[T]}), k\right)} \right] \tag{G.3}$$

*where $(\ell_{s,t}^{(j)})_{j\in[N],t\in[T]}$ denotes the $s$-th sample of $(\ell_t^{(j)})_{j\in[N],t\in[T]}$ for estimating $\mathcal{L}(\theta, k, N)$.*

### G.5. Deferred Proofs of Theorem G.1 and Theorem 5.1

**Theorem G.1.** (Generalization gap). *Suppose* $\text{LLM}_\theta$ *is Lipschitz-continuous with respect to the model parameter $\theta$, then for any $0 < \epsilon < 1/2$, with probability at least $1 - \epsilon$, we have*

$$\mathcal{L}\left(\widehat{\theta}_{k,N,N_T}, k, N\right) - \inf_{\theta\in\Theta} \mathcal{L}(\theta, k, N) \leq \widetilde{\mathcal{O}}\left(\sqrt{\frac{d_\theta + \log(1/\epsilon)}{N_T}}\right), \tag{G.4}$$

*for any $N$ and sufficiently large $k$, where $d_\theta$ is the dimension of the parameter $\theta$.*

Through a careful use of Berge's Maximum Theorem (Berge, 1877), we prove that the right-hand side of Equation (G.4) does *not* depend on $k$ and $N$, which allows us to take the limit of $\lim_{N\to\infty} \lim_{k\to\infty}$ without affecting the generalization bound.

Before proving the theorem, we remark on what LLM structure enjoys the Lipschitz-continuity. We provide two auxiliary results in the following proposition. The first result is from (Bai et al., 2023, Section J.1), which is about the Lipschitzness of Transformers. The second result is regarding processing the output of Transformers. In particular, the output of Transformers is usually not directly used, but passed through some matrix multiplication (by some matrix $A$), followed by some projection `Operator` (to be specified later).

**Proposition 2.** *The L-layer Transformer $\text{TF}_\theta$ as defined in Appendix D.1 is $C_{\text{TF}}$-Lipschitz continuous with respect to $\theta$ with $C_{\text{TF}} := L\left((1 + B_{\text{TF}}^2)(1 + B_{\text{TF}}^2 R^3)\right)^L B_{\text{TF}} R(1 + B_{\text{TF}} R^2 + B_{\text{TF}}^3 R^2)$, i.e.,*

$$\|\text{TF}_{\theta_1}(Z) - \text{TF}_{\theta_2}(Z)\|_{2,\infty} \leq C_{\text{TF}}\|\theta_1 - \theta_2\|_{\text{TF}}$$

*where $\|\cdot\|_{\text{TF}}$ is as defined in Equation (D.1), and $R, Z, B_{\text{TF}}$ are as introduced in Appendix D.1. Moreover, the function $\text{Operator}(A \cdot \text{TF}_\theta(\cdot)_{-1})$ is $\|A\|_{op}C_{\text{TF}}$-Lipschitz continuous with respect to $\theta$, i.e.,*

$$\|\text{Operator}(A \cdot \text{TF}_{\theta_1}(Z)_{-1}) - \text{Operator}(A \cdot \text{TF}_{\theta_2}(Z)_{-1})\|_2 \leq \|A\|_{op}C_{\text{TF}}\|\theta_1 - \theta_2\|_{\text{TF}}.$$

*Here, `Operator` is either the projection operator onto some convex set, or the `Softmax` function.*

*Proof.* The first result is from (Bai et al., 2023, Section J.1). The second result comes from

- If `Operator` is a projection onto the convex set, then $\|\text{Operator}(x) - \text{Operator}(y)\|_2 \leq \|x - y\|_2$;

- If `Operator` is `Softmax`, then $\|\text{Softmax}(x) - \text{Softmax}(y)\|_2 \leq \|x - y\|_2$ (Gao & Pavel, 2017, Corollary 3).

Note that the only condition that we require for `Operator` is its non-expansiveness. $\qquad\square$

*Proof of Theorem G.1.* Let $C_{\text{LLM}}$ denote the Lipschitz-continuity constant for $\text{LLM}_\theta$ with respect to some norm $\|\cdot\|_{\text{LLM}}$, where $\|\cdot\|_{\text{LLM}}$ denotes any norm defined on the parameter space of LLM (e.g., the norm $\|\cdot\|_{\text{TF}}$ above in Proposition 2). Now, we prove that regret is also a Lipschitz-continuous function with respect to the LLM's parameter.

**Lemma 11** (Lipschitzness of regret). *The function $Regret_{LLM_\theta}$ is $C_{Reg} := BC_{LLM}T$-Lipschitz continuous with respect to $\theta$, i.e.,*

$$\left|\text{Regret}_{LLM_{\theta_1}}((\ell_t)_{t\in[T]}) - \text{Regret}_{LLM_{\theta_2}}((\ell_t)_{t\in[T]})\right| \leq C_{Reg}\|\theta_1 - \theta_2\|_{LLM}.$$

*Proof.* By definition, we have

$$\left| \text{Regret}_{\text{LLM}_{\theta_1}}((\ell_t)_{t\in[T]}) - \text{Regret}_{\text{LLM}_{\theta_2}}((\ell_t)_{t\in[T]}) \right| = \left| \sum_{t=1}^{T} \langle \ell_t, \text{LLM}_{\theta_1}(Z_{t-1}) - \text{LLM}_{\theta_2}(Z_{t-1}) \rangle \right|$$

$$= B \sum_{t=1}^{T} \| \text{LLM}_{\theta_1}(Z_{t-1}) - \text{LLM}_{\theta_2}(Z_{t-1}) \|$$

$$\leq BC_{\text{LLM}} T \| \theta_1 - \theta_2 \|_{\text{LLM}}$$

where $Z_t := (\ell_1, \ldots, \ell_t, c)$ for all $t \in [T]$ and $Z_0 = (c)$ where $c$ is a $d$-dimensional vector. $\square$

Now, we will prove the Lipschitzness of

$$C\left( (\ell_t^{(j)})_{t\in[T], j\in[N]}, k, \theta \right) := \frac{\sum_{j\in[N]} h(\text{Regret}_{\text{LLM}_\theta}((\ell_t^{(j)})_{t\in[T]})) f(\text{Regret}_{\text{LLM}_\theta}((\ell_t^{(j)})_{t\in[T]}), k)}{\sum_{j\in[N]} f(\text{Regret}_{\text{LLM}_\theta}((\ell_t^{(j)})_{t\in[t]}), k)} \tag{G.5}$$

with respect to the model parameter $\theta$.

**Claim 5.** *For any $R > 0$, there exists $\beta_R > 0$ such that if $\beta > \beta_R$, we have*

$$\left| \frac{\sum_{n\in[N]} x_n f(x_n, \beta)}{\sum_{n\in[N]} f(x_n, \beta)} - \frac{\sum_{n\in[N]} y_n f(y_n, \beta)}{\sum_{n\in[N]} f(y_n, \beta)} \right| \leq 2\|x - y\|_\infty$$

*for every $x, y \in \mathbb{R}^n$ such that $|x_i| \leq R, |y_i| \leq R$ for all $i \in [N]$.*

*Proof.* If $\beta = \infty$, we have

$$\lim_{\beta\to\infty} \left( \left| \frac{\sum_{n\in[N]} x_n f(x_n, \beta)}{\sum_{n\in[N]} f(x_n, \beta)} - \frac{\sum_{n\in[N]} y_n f(y_n, \beta)}{\sum_{n\in[N]} f(y_n, \beta)} \right| \Big/ \|x - y\|_\infty \right) = \frac{|\max_{n\in[N]} x_n - \max_{n\in[N]} y_n|}{\|x - y\|_\infty} \leq 1$$

holds. Moreover, consider the following constrained optimization problem:

$$\max_{x,y\in\mathbb{R}^n} \left( \left| \frac{\sum_{n\in[N]} x_n f(x_n, \beta)}{\sum_{n\in[N]} f(x_n, \beta)} - \frac{\sum_{n\in[N]} y_n f(y_n, \beta)}{\sum_{n\in[N]} f(y_n, \beta)} \right| \Big/ \|x - y\|_\infty \right)$$

$$\text{subject to} \quad |x_i| \leq R, \quad |y_i| \leq R \quad \text{for all } i \in [N],$$

whose optimum is denoted as $F(R, \beta)$. Then, since $\|x\|_\infty \leq R$ and $\|y\|_\infty \leq R$ is a compact set, by Berge's Maximum Theorem (Berge, 1877), we have that $F(R, \beta)$ is a continuous function for $\beta$. Moreover, we know that $F(R, \infty) \leq 1$, which indicates that we can find a large enough $\beta_R$ such that if $\beta > \beta_R$, $F(R, \beta) \leq 2$. $\square$

Note that Claim 5 does not hold if either $x_i$ or $y_i$ is unbounded. Now, we will apply Claim 5 to Equation (G.5). We can guarantee that $\left| \text{Regret}_{\text{LLM}_\theta}((\ell_t)_{t\in[T]}) \right| \leq \text{diam}(\Pi, \|\cdot\|_2) TB$.

Also, note that the domain of $h : \mathbb{R} \to \mathbb{R}^+$ is effectively *constrained* to the range that $\text{Regret}_{\text{LLM}_\theta}((\ell_t)_{t\in[T]})$ can achieve, which means that we can regard $h$ as $h : [-\text{diam}(\Pi, \|\cdot\|_2)TB, \text{diam}(\Pi, \|\cdot\|_2)TB] \to \mathbb{R}^+$. Due to the continuity of $h'$, and the fact that $h$ has a compact domain, we know that $h(\cdot)$ is $C_h$-Lipschitz continuous for some $C_h > 0$ on this interval of $[-\text{diam}(\Pi, \|\cdot\|_2)TB, \text{diam}(\Pi, \|\cdot\|_2)TB]$.

**Lemma 12** (Lipschitzness of $C$ in Equation (G.5))**.** *The function $C$ in Equation (G.5) is $C_{cost} := 2C_h C_{Reg}$-Lipschitz continuous with respect to $\theta$, if $k > k_{diam(\Pi, \|\cdot\|_2)TB}$ for some $k_{diam(\Pi, \|\cdot\|_2)TB} > 0$, i.e.,*

$$\left| C\left( (\ell_t^{(j)})_{t\in[T], j\in[N]}, k, \theta_1 \right) - C\left( (\ell_t^{(j)})_{t\in[T], j\in[N]}, k, \theta_2 \right) \right| \leq C_{cost} \|\theta_1 - \theta_2\|_{LLM}.$$

*Proof.*

$$\left| C((\ell_t^{(j)})_{t\in[T],j\in[N]}, k, \theta_1) - C((\ell_t^{(j)})_{t\in[T],j\in[N]}, k, \theta_2) \right|$$

$$\underset{(i)}{\leq} 2\|h(\text{Regret}_{\text{LLM}_{\theta_1}}((\ell_t^{(j)})_{t\in[T]})) - h(\text{Regret}_{\text{LLM}_{\theta_2}}((\ell_t^{(j)})_{t\in[T]}))\|_\infty$$

$$\underset{(ii)}{\leq} 2C_h\|\text{Regret}_{\text{LLM}_{\theta_1}}((\ell_t^{(j)})_{t\in[T]}) - \text{Regret}_{\text{LLM}_{\theta_2}}((\ell_t^{(j)})_{t\in[T]})\|_\infty$$

$$\underset{(iii)}{\leq} 2C_h C_{\text{Reg}}\|\theta_1 - \theta_2\|_{\text{LLM}} = C_{\text{cost}}\|\theta_1 - \theta_2\|_{\text{LLM}}.$$

Here, (i) holds due to Claim 5, (ii) holds since $h$ is $C_h$-Lipschitz continuous on the range of $\text{Regret}_{\text{LLM}_\theta}((\ell_t)_{t\in[T]})$, and (iii) holds due to Lemma 11. $\qquad\square$

For completeness of the paper, we provide the definition of covering set and covering number.

**Definition G.2** (Covering set and covering number). *For $\delta > 0$, a metric space $(X, \|\cdot\|)$, and subset $Y \subseteq X$, set $C \subset Y$ is a $\delta$-covering of $Y$ when $Y \subseteq \cup_{c\in C}B(c, \delta, \|\cdot\|)$ holds. $\delta$-covering number $N(\delta; Y, \|\cdot\|)$ is defined as the minimum cardinality of any covering set.*

By (Wainwright, 2019, Example 5.8), for any $r > 0$, we can verify that the $\delta$-covering number $N(\delta; B(0, r, \|\cdot\|_{\text{LLM}}), \|\cdot\|_{\text{LLM}})$ can be bounded by

$$\log N(\delta; B(0, r, \|\cdot\|_{\text{LLM}}), \|\cdot\|_{\text{LLM}}) \leq d_\theta \log(1 + 2r/\delta),$$

where $d_\theta$ is the dimension of the LLM's whole parameter. For example, if we use the $\|\cdot\|_{\text{TF}}$ and consider the Transformer model as defined in Appendix D.1, for any $r > 0$,

$$\log N(\delta; B(0, r, \|\cdot\|_{\text{LLM}}), \|\cdot\|_{\text{LLM}}) \leq L(3Md^2 + 2d(dd' + 3md^2))\log(1 + 2r/\delta).$$

Since we consider a compact $\Theta$ (as several recent works on analyzing Transformers (Bai et al., 2023; Lin et al., 2024)), let $R_\Theta := \text{diam}(\Theta, \|\cdot\|_{\text{LLM}})$ (which corresponds to $B_{\text{TF}}$ for the Transformer models as defined in Appendix D.1, with $\|\cdot\|_{\text{LLM}} = \|\cdot\|_{\text{TF}}$), then there exists a set $\Theta_0$ with $\log|\Theta_0| = d_\theta \log(1 + 2R_\Theta/\delta)$ such that for any $\theta \in \Theta$, there exists a $\theta_0 \in \Theta_0$ with

$$\left| C\left((\ell_t^{(j)})_{t\in[T],j\in[N]}, k, \theta\right) - C\left((\ell_t^{(j)})_{t\in[T],j\in[N]}, k, \theta_0\right) \right| \leq C_{\text{cost}}\delta.$$

Then, by the standard result from statistical learning theory (Wainwright, 2019, Chapter 5), when trained with $N_T$ samples, for every $0 < \epsilon < 1/2$, with probability at least $1 - \epsilon$, we have

$$\mathcal{L}(\widehat{\theta}_{k,N,N_T}, k, N) - \inf_{\theta\in\Theta} \mathcal{L}(\theta, k, N) \leq \sqrt{\frac{2(\log|\Theta_0| + \log(2/\epsilon))}{N_T}} + 2C_{\text{cost}}\delta.$$

Setting $\delta = \Omega(\sqrt{\log(\epsilon)/N_T})$, we further obtain

$$\mathcal{L}(\widehat{\theta}_{k,N,N_T}, k, N) - \inf_{\theta\in\Theta} \mathcal{L}(\theta, k, N) \leq \widetilde{\mathcal{O}}\left(\sqrt{\frac{\log|\Theta_0| + \log(1/\epsilon)}{N_T}}\right)$$

with probability at least $1 - \epsilon$, completing the proof. $\qquad\square$

**Theorem 5.1.** (Regret, Informal). *Under regular conditions on $f, h$, with high probably, we have*

$$h\left(\lim_{N\to\infty}\lim_{k\to\infty}\max_{\|\ell_t\|_\infty\leq B}\text{Regret}_{\text{LLM}_{\widehat{\theta}_{k,N,N_T}}}((\ell_t)_{t\in[T]})\right)$$

$$\leq h\left(\inf_{\theta\in\Theta}\max_{\|\ell_t\|_\infty\leq B}\text{Regret}_{\text{LLM}_\theta}((\ell_t)_{t\in[T]})\right) + \widetilde{\mathcal{O}}\left(\sqrt{\frac{d_\theta}{N_T}}\right).$$

*Proof.* The limit on the right-hand side of Equation (G.4) remains as $\widetilde{\mathcal{O}}\left(\sqrt{\frac{d_\theta + \log(1/\epsilon)}{N_T}}\right)$, since we firstly take $\lim_{k\to\infty}$ and then take $\lim_{N\to\infty}$, thanks to the fact that Theorem G.1 holds for large enough $k$ and any $N$. Next, we have

$$\lim_{N\to\infty}\lim_{k\to\infty}\left|\mathcal{L}(\widehat{\theta}_{k,N,N_T}, k, N) - h\left(\lim_{N\to\infty}\lim_{k\to\infty}\max_{\|\ell_t\|_\infty \le B}\text{Regret}_{\text{LLM}_{\widehat{\theta}_{k,N,N_T}}}((\ell_t)_{t\in[T]})\right)\right|$$

$$\le \lim_{N\to\infty}\lim_{k\to\infty}\left|\mathcal{L}(\widehat{\theta}_{k,N,N_T}, k, N) - h\left(\max_{\|\ell_t\|_\infty \le B}\text{Regret}_{\text{LLM}_{\widehat{\theta}_{k,N,N_T}}}((\ell_t)_{t\in[T]})\right)\right| +$$

$$\lim_{N\to\infty}\lim_{k\to\infty}\left|h\left(\max_{\|\ell_t\|_\infty \le B}\text{Regret}_{\text{LLM}_{\widehat{\theta}_{k,N,N_T}}}((\ell_t)_{t\in[T]})\right) - h\left(\lim_{N\to\infty}\lim_{k\to\infty}\max_{\|\ell_t\|_\infty \le B}\text{Regret}_{\text{LLM}_{\widehat{\theta}_{k,N,N_T}}}((\ell_t)_{t\in[T]})\right)\right|$$

$$\le \lim_{N\to\infty}\lim_{k\to\infty}\sup_{\theta\in\Theta}\left|\mathcal{L}(\theta, k, N) - h\left(\max_{\|\ell_t\|_\infty \le B}\text{Regret}_{\text{LLM}_\theta}((\ell_t)_{t\in[T]})\right)\right| + 0 = 0,$$

due to the continuity of $h$ and Claim 3. Finally, we have

$$\lim_{N\to\infty}\lim_{k\to\infty}\inf_{\theta\in\Theta}\mathcal{L}(\theta, k, N) = \inf_{\theta\in\Theta} h\left(\max_{\ell_1,...,\ell_T}\text{Regret}_{\text{LLM}_\theta}((\ell_t)_{t\in[T]})\right)$$

due to Claim 4, which, combined with the fact that $h$ is non-decreasing, completes the proof. □

As a result, the coarse correlated equilibrium will emerge as the long-term interactions of multiple such learned LLMs, as stated in the following corollary.

**Corollary 1.** (Emerging behavior: Coarse correlated equilibrium). *For a sufficiently large $N_T$, if each agent in the matrix game plays according to $\text{LLM}_{\widehat{\theta}_{k,N,N_T}}$, then the time-averaged policy for each agent will constitute an approximate coarse correlated equilibrium of the game.*

**Remark G.1** (Dynamic-regret loss). *So far, we have focused on the canonical online learning setting with regret being the metric. One can also generalize the results to the non-stationary setting, with dynamic regret being the metric. Specifically, one can define the* dynamic-regret-loss *function as follows:*

$$\mathcal{L}(\theta, k, N) := \mathbb{E}\left[\frac{\sum_{j\in[N]} h(\text{D-Regret}_{\text{LLM}_\theta}((\ell_t^{(j)})_{t\in[T]})) f(\text{D-Regret}_{\text{LLM}_\theta}((\ell_t^{(j)})_{t\in[T]}), k)}{\sum_{j\in[N]} f(\text{D-Regret}_{\text{LLM}_\theta}((\ell_i^{(j)})_{t\in[T]}), k)}\right].$$

*Then, one can also establish similar results as before, since the analysis does not utilize other properties of the regret except its boundedness, and the Lipschitz-continuity of LLM with respect to θ. To be specific, Lemma 11 holds due to the reason that we can bound the difference of the regret with the term*

$$\left|\sum_{t=1}^T \langle\ell_t, (\text{LLM}_{\theta_1}(Z_{t-1}) - \text{LLM}_{\theta_2}(Z_{t-1}))\rangle\right|,$$

*as well as the fact that $\inf_{\pi_i\in\Pi}\langle\ell_i, \pi_i\rangle$ will be canceled. One can verify that all the arguments in Appendix G.3 also hold for similar reasons.*

### G.6. Detailed Explanation of Optimizing Equation (5.2) with Single-layer Self-attention Model

We consider the single-layer *linear* self-attention model as follows, for which we can show that the *global optimizer* of our regret-loss can automatically lead to a no-regret learning algorithm:

$$g(Z_t; V, K, Q, v_c, k_c, q_c) = \sum_{i=1}^t (V\ell_i + v_c)\left((K\ell_i + k_c)^\intercal \cdot (Qc + q_c)\right). \tag{G.6}$$

**Theorem G.2.** *Consider the policy space $\Pi = B(0, R_\Pi, \|\cdot\|)$ for some $R_\Pi > 0$. The configuration of a single-layer linear self-attention model in Equation (G.6) $(V, K, Q, v_c, k_c, q_c)$ such that $K^\intercal(Qc + q_c) = v_c = \mathbf{0}_d$ and $V = -2R_\Pi\Sigma^{-1}\mathbb{E}\left(\|\sum_{t=1}^T \ell_t\|\ell_1\ell_2^\intercal\right)\Sigma^{-1}$ is a **global optimal solution** of Equation (5.2) with $N = 1$, $h(x) = x^2$. Moreover, every global optimal configuration of Equation (5.2) within the parameterization class of Equation (G.6) has the same output function g. Additionally, if $\Sigma$ is a diagonal matrix, then plugging any global optimal configuration into Equation (G.6), and projecting the output with $\texttt{Proj}_{\Pi,\|\cdot\|}$ is equivalent to FTRL with an $L_2$-regularizer.*

We consider the following structure of single-layer self-attention model $g$ (see a formal introduction in Appendix D.1):

$$g(Z_t; V, K, Q, v_c, k_c, q_c) := (V\ell_{1:t} + v_c\mathbf{1}_t^\mathsf{T})\texttt{Softmax}\left((K\ell_{1:t} + k_c\mathbf{1}_t^\mathsf{T})^\mathsf{T} \cdot (Qc + q_c)\right), \tag{G.7}$$

where $Z_t = (\ell_1, \ldots, \ell_t, c)$ and $V, K, Q \in \mathbb{R}^{d \times d}$ correspond to the value, key, and query matrices, respectively, $v_c, k_c, q_c \in \mathbb{R}^d$ correspond to the bias terms associated with $V, K, Q$, and $c \neq \mathbf{0}_d$ is a constant vector. We then have the following result.

**Theorem G.3.** *Consider the policy space $\Pi = B(0, R_\Pi, \|\cdot\|)$ for some $R_\Pi > 0$. The configuration of a single-layer self-attention model in Equation (G.7) $(V, K, Q, v_c, k_c, q_c)$ such that $K^\mathsf{T}(Qc + q_c) = v_c = \mathbf{0}_d$ and $V = -R_\Pi \frac{T}{\sum_{t=1}^{T-1} 1/t} \Sigma^{-1} \mathbb{E}\left[\left\|\sum_{t=1}^T \ell_t\right\| \ell_1 \ell_2^\mathsf{T}\right] \Sigma^{-1}$ is a first-order stationary point of Equation (5.2) with $N = 1$, $h(x) = x^2$. Moreover, if $\Sigma$ is a diagonal matrix, then plugging this configuration into Equation (G.7), and projecting the output with $\texttt{Proj}_{\Pi, \|\cdot\|}$ would perform FTRL with an $L_2$-regularizer for the loss vectors $(\ell_t)_{t \in [T]}$.*

In practical training, such stationary points of the loss may be attained by first-order optimization algorithms of (stochastic) gradient descent, the workhorse in machine learning.

## G.7. Deferred Proof of Theorem G.3

**Theorem G.3.** *Consider the policy space $\Pi = B(0, R_\Pi, \|\cdot\|)$ for some $R_\Pi > 0$. The configuration of a single-layer self-attention model in Equation (G.7) $(V, K, Q, v_c, k_c, q_c)$ such that $K^\mathsf{T}(Qc + q_c) = v_c = \mathbf{0}_d$ and $V = -R_\Pi \frac{T}{\sum_{t=1}^{T-1} 1/t} \Sigma^{-1} \mathbb{E}\left[\left\|\sum_{t=1}^T \ell_t\right\| \ell_1 \ell_2^\mathsf{T}\right] \Sigma^{-1}$ is a first-order stationary point of Equation (5.2) with $N = 1$, $h(x) = x^2$. Moreover, if $\Sigma$ is a diagonal matrix, then plugging this configuration into Equation (G.7), and projecting the output with $\texttt{Proj}_{\Pi, \|\cdot\|}$ would perform FTRL with an $L_2$-regularizer for the loss vectors $(\ell_t)_{t \in [T]}$.*

*Proof.* We will locally use $\mathcal{A} = [d]$ without losing generality as $\mathcal{A}$ is finite with $|\mathcal{A}| = d$, and will interchangeably use $\ell_i(j)$ and $\ell_{ij}$ for notational convenience. Define $a := K^\mathsf{T}(Qc + q_c) \in \mathbb{R}^d$ and $b_{t-1} := \beta\mathbf{1}_{t-1} := k_c^\mathsf{T}(Qc + q_c)\mathbf{1}_{t-1} \in \mathbb{R}^{t-1}$. With $N = 1$, $h(x) = x^2$, and the choice of $\Pi$, the loss function (Equation (5.2)) can be written as follows:

$$f(V, a, (b_t)_{t \in [T-1]}, v_c) := \mathbb{E}\left(\sum_{t=1}^T \ell_t^\mathsf{T}(V\ell_{1:t-1} + v_c\mathbf{1}_{t-1}^\mathsf{T})\texttt{Softmax}(\ell_{1:t-1}^\mathsf{T}a + b_{t-1}) + R_\Pi\left\|\sum_{t=1}^T \ell_t\right\|_2\right)^2,$$

where for $t = 1$, we use the output of the single-layer self-attention as $v_c$ and we will write it as $(V\ell_{1:0} + v_c\mathbf{1}_0^\mathsf{T})\texttt{Softmax}(\ell_{1:0}^\mathsf{T}a + b_0)$ for notational consistency with $t \geq 2$. Also, we will define empty sum $\sum_{i=1}^0 a_i = 0$ for any sequence $(a_i)_{i \in \mathbb{N}+}$.

**Step 1.** Calculating $\frac{\partial f}{\partial a}$.

For $x \in [d]$, we calculate the corresponding directional derivative with the following equation for $t \geq 2$:

$$\frac{\partial}{\partial a_x}\ell_t^\mathsf{T}(V\ell_{1:t-1} + v_c\mathbf{1}_{t-1}^\mathsf{T})\texttt{Softmax}(\ell_{1:t-1}^\mathsf{T}a + b_{t-1})$$

$$= \frac{\partial}{\partial a_x}\sum_{i=1}^{t-1} \ell_t^\mathsf{T}(V\ell_{1:t-1} + v_c\mathbf{1}_{t-1}^\mathsf{T})e_i \frac{\exp(e_i^\mathsf{T}(\ell_{1:t-1}^\mathsf{T}a + b_{t-1}))}{\sum_{s=1}^{t-1}\exp(e_s^\mathsf{T}(\ell_{1:t-1}^\mathsf{T}a + b_{t-1}))}$$

$$= \frac{\sum_{i=1}^{t-1} \ell_t^\mathsf{T}(V\ell_{1:t-1} + v_c\mathbf{1}_{t-1}^\mathsf{T})e_i \exp(e_i^\mathsf{T}(\ell_{1:t-1}^\mathsf{T}a + b_{t-1}))\frac{\partial e_i^\mathsf{T}(\ell_{1:t-1}^\mathsf{T}a + b_{t-1})}{\partial a_x}(\sum_{s=1}^{t-1}\exp(e_s^\mathsf{T}(\ell_{1:t-1}^\mathsf{T}a + b_{t-1})))}{(\sum_{s=1}^{t-1}\exp(e_s^\mathsf{T}(\ell_{1:t-1}^\mathsf{T}a + b_{t-1})))^2}$$

$$- \frac{\sum_{i=1}^{t-1} \ell_t^\mathsf{T}(V\ell_{1:t-1} + v_c\mathbf{1}_{t-1}^\mathsf{T})e_i \exp(e_i^\mathsf{T}(\ell_{1:t-1}^\mathsf{T}a + b_{t-1}))\left(\sum_{s=1}^{t-1}\exp(e_s^\mathsf{T}(\ell_{1:t-1}^\mathsf{T}a + b_{t-1}))\frac{\partial e_s^\mathsf{T}(\ell_{1:t-1}^\mathsf{T}a + b_{t-1})}{\partial a_x}\right)}{(\sum_{s=1}^{t-1}\exp(e_s^\mathsf{T}(\ell_{1:t-1}^\mathsf{T}a + b_{t-1})))^2}.$$

Plugging $a = \mathbf{0}_d$ and $v_c = \mathbf{0}_d$, and $(b_t = \beta \mathbf{1}_t)_{t \in [T-1]}$ provides

$$
\frac{\partial}{\partial a_x} \ell_t^\intercal (V \ell_{1:t-1} + v_c \mathbf{1}_{t-1}^\intercal) \texttt{Softmax}(\ell_{1:t-1}^\intercal a + b_{t-1}) \Big|_{a = \mathbf{0}_d, v_c = \mathbf{0}_d, (b_t = \beta \mathbf{1}_t)_{t \in [T-1]}}
$$

$$
= \sum_{i=1}^{t-1} \frac{\ell_t^\intercal V \ell_i \ell_{ix}}{(t-1)} - \sum_{i=1}^{t-1} \frac{\ell_t^\intercal V \ell_i \left( \sum_{s=1}^{t-1} \ell_{sx} \right)}{(t-1)^2}.
$$

For $t = 1$, as $\ell_t^\intercal (V \ell_{1:t-1} + v_c \mathbf{1}_{t-1}^\intercal) \texttt{Softmax}(\ell_{1:t-1}^\intercal a + b_{t-1}) = \ell_1^\intercal v_c$, $\frac{\partial}{\partial a_x} \ell_t^\intercal (V \ell_{1:t-1} + v_c \mathbf{1}_{t-1}^\intercal) \texttt{Softmax}(\ell_{1:t-1}^\intercal a +$

$b_{t-1}) \Big|_{a = \mathbf{0}_d, v_c = \mathbf{0}_d, (b_t = \beta \mathbf{1}_t)_{t \in [T-1]}} = 0$, so we can use the same formula as $t \geq 2$ with empty sum $\sum_{i=1}^{t-1}$. Using the above

calculation, we can further compute $\frac{\partial f}{\partial a_x} \Big|_{a = \mathbf{0}_d, v_c = \mathbf{0}_d, (b_t = \beta \mathbf{1}_t)_{t \in [T-1]}}$ as follows:

$$
\frac{\partial f(V, a, (b_t)_{t \in [T-1]}, v_c)}{\partial a_x} \Big|_{a = \mathbf{0}_d, v_c = \mathbf{0}_d, (b_t = \beta \mathbf{1}_t)_{t \in [T-1]}}
$$

$$
= \mathbb{E} \frac{\partial}{\partial a_x} \left( \sum_{t=1}^{T} \ell_t^\intercal (V \ell_{1:t-1} + v_c \mathbf{1}_{t-1}^\intercal) \texttt{Softmax}(\ell_{1:t-1}^\intercal a + b_{t-1}) + R_\Pi \Big\| \sum_{t=1}^{T} \ell_t \Big\|_2 \right)^2 \Big|_{a = \mathbf{0}_d, v_c = \mathbf{0}_d, (b_t = \beta \mathbf{1}_t)_{t \in [T-1]}}
$$

$$
= \mathbb{E} \Bigg[ \left( \sum_{t=1}^{T} \ell_t^\intercal (V \ell_{1:t-1} + v_c \mathbf{1}_{t-1}^\intercal) \texttt{Softmax}(\ell_{1:t-1}^\intercal a + b_{t-1}) + R_\Pi \Big\| \sum_{t=1}^{T} \ell_t \Big\|_2 \right) \Big|_{a = \mathbf{0}_d, v_c = \mathbf{0}_d, (b_t = \beta \mathbf{1}_t)_{t \in [T-1]}}
$$

$$
\frac{\partial}{\partial a_x} \left( \sum_{t=1}^{T} \ell_t^\intercal (V \ell_{1:t-1} + v_c \mathbf{1}_{t-1}^\intercal) \texttt{Softmax}(\ell_{1:t-1}^\intercal a + b_{t-1}) + R_\Pi \Big\| \sum_{t=1}^{T} \ell_t \Big\|_2 \right) \Big|_{a = \mathbf{0}_d, v_c = \mathbf{0}_d, (b_t = \beta \mathbf{1}_t)_{t \in [T-1]}} \Bigg]
$$

$$
= \mathbb{E} \Bigg[ \left( \sum_{t=1}^{T} \ell_t^\intercal V \sum_{i=1}^{t-1} \frac{1}{t-1} \ell_i + R_\Pi \Big\| \sum_{t=1}^{T} \ell_t \Big\|_2 \right) \sum_{t=1}^{T} \left( \sum_{i=1}^{t-1} \frac{\ell_t^\intercal V \ell_i \ell_{ix}}{(t-1)} - \sum_{i=1}^{t-1} \frac{\ell_t^\intercal V \ell_i \left( \sum_{s=1}^{t-1} \ell_{sx} \right)}{(t-1)^2} \right) \Bigg] \quad (G.8)
$$

$$
= 0,
$$

where we used the fact that $\ell_i$ is drawn from a symmetric distribution, and flipping the sign of the variable as $-\ell_i$ yields the same distribution, which leads to the following:

$$
\mathbb{E} \Bigg[ \left( \sum_{t=1}^{T} \ell_t^\intercal V \sum_{i=1}^{t-1} \frac{1}{t-1} \ell_i + R_\Pi \Big\| \sum_{t=1}^{T} \ell_t \Big\|_2 \right) \sum_{t=1}^{T} \left( \sum_{i=1}^{t-1} \frac{\ell_t^\intercal V \ell_i \ell_{ix}}{(t-1)} - \sum_{i=1}^{t-1} \frac{\ell_t^\intercal V \ell_i \left( \sum_{s=1}^{t-1} \ell_{sx} \right)}{(t-1)^2} \right) \Bigg]
$$

$$
= \mathbb{E} \Bigg[ \left( \sum_{t=1}^{T} \ell_t^\intercal V \sum_{i=1}^{t-1} \frac{1}{t-1} \ell_i + R_\Pi \Big\| \sum_{t=1}^{T} \ell_t \Big\|_2 \right) \sum_{t=1}^{T} \left( - \sum_{i=1}^{t-1} \frac{\ell_t^\intercal V \ell_i \ell_{ix}}{(t-1)} + \sum_{i=1}^{t-1} \frac{\ell_t^\intercal V \ell_i \left( \sum_{s=1}^{t-1} \ell_{sx} \right)}{(t-1)^2} \right) \Bigg].
$$

This yields Equation (G.8)=0.

**Step 2. Calculating $\frac{\partial f}{\partial v_c}$.**

We will use the following equation for $t \geq 2$:

$$
\frac{\partial}{\partial v_c} \ell_t^\intercal (V \ell_{1:t-1} + v_c \mathbf{1}_{t-1}^\intercal) \texttt{Softmax}(\ell_{1:t-1}^\intercal a + b_{t-1})
$$

$$
= \frac{\partial}{\partial v_c} \sum_{i=1}^{t-1} \ell_t^\intercal (V \ell_{1:t-1} + v_c \mathbf{1}_{t-1}^\intercal) e_i \frac{\exp(e_i^\intercal (\ell_{1:t-1}^\intercal a + b_{t-1}))}{\sum_{s=1}^{t-1} \exp(e_s^\intercal (\ell_{1:t-1}^\intercal a + b_{t-1}))} = \ell_t.
$$

For $t = 1$, we define $\frac{\partial}{\partial v_c} \ell_1^\intercal (V \ell_{1:0} + v_c \mathbf{1}_0^\intercal) \texttt{Softmax}(\ell_{1:0}^\intercal a + b_0) = \ell_1$, so that we can use the same formula as $t \geq 2$.

Therefore, we can calculate $\frac{\partial f}{\partial v_c}\Big|_{a=\mathbf{0}_d, v_c=\mathbf{0}_d, (b_t=\beta\mathbf{1}_t)_{t\in[T-1]}}$ as follows:

$$
\frac{\partial f(V, a, (b_t)_{t\in[T-1]}, v_c)}{\partial v_c}\Bigg|_{a=\mathbf{0}_d, v_c=\mathbf{0}_d, (b_t=\beta\mathbf{1}_t)_{t\in[T-1]}}
$$

$$
= \mathbb{E}\frac{\partial}{\partial v_c}\left(\sum_{t=1}^{T}\ell_t^{\mathsf{T}}(V\ell_{1:t-1} + v_c\mathbf{1}_{t-1}^{\mathsf{T}})\mathrm{Softmax}(\ell_{1:t-1}^{\mathsf{T}}a + b_{t-1}) + R_{\Pi}\|\sum_{t=1}^{T}\ell_t\|_2\right)^2\Bigg|_{a=\mathbf{0}_d, v_c=\mathbf{0}_d, (b_t=\beta\mathbf{1}_t)_{t\in[T-1]}}
$$

$$
= \mathbb{E}\Bigg[\left(\sum_{t=1}^{T}\ell_t^{\mathsf{T}}(V\ell_{1:t-1} + v_c\mathbf{1}_{t-1}^{\mathsf{T}})\mathrm{Softmax}(\ell_{1:t-1}^{\mathsf{T}}a + b_{t-1}) + R_{\Pi}\|\sum_{t=1}^{T}\ell_t\|_2\right)\Bigg|_{a=\mathbf{0}_d, v_c=\mathbf{0}_d, (b_t=\beta\mathbf{1}_t)_{t\in[T-1]}}
$$

$$
\frac{\partial}{\partial v_c}\left(\sum_{t=1}^{T}\ell_t^{\mathsf{T}}(V\ell_{1:t-1} + v_c\mathbf{1}_{t-1}^{\mathsf{T}})\mathrm{Softmax}(\ell_{1:t-1}^{\mathsf{T}}a + b_{t-1}) + R_{\Pi}\|\sum_{t=1}^{T}\ell_t\|_2\right)\Bigg|_{a=\mathbf{0}_d, v_c=\mathbf{0}_d, (b_t=\beta\mathbf{1}_t)_{t\in[T-1]}}\Bigg]
$$

$$
= \mathbb{E}\Bigg[\left(\sum_{t=2}^{T}\ell_t^{\mathsf{T}}V\sum_{i=1}^{t-1}\frac{1}{t-1}\ell_i + R_{\Pi}\|\sum_{t=1}^{T}\ell_t\|_2\right)\sum_{t=1}^{T}\ell_t\Bigg] = 0.
$$

The last line is due to the same reason as the last part of Step 1.

**Step 3.** **Calculating $\frac{\partial f}{\partial V}$.**

We calculate the following equation, which will be used to calculate $\frac{\partial f}{\partial V}\Big|_{a=\mathbf{0}_d, v_c=\mathbf{0}_d, (b_t=\beta\mathbf{1}_t)_{t\in[T-1]}}$ for $t \geq 2$:

$$
\frac{\partial}{\partial V}\ell_t^{\mathsf{T}}(V\ell_{1:t-1} + v_c\mathbf{1}_{t-1}^{\mathsf{T}})\mathrm{Softmax}(\ell_{1:t-1}^{\mathsf{T}}a + b_{t-1})\Bigg|_{a=\mathbf{0}_d, v_c=\mathbf{0}_d, (b_t=\beta\mathbf{1}_t)_{t\in[T-1]}}
$$

$$
= \frac{\partial}{\partial V}\sum_{i=1}^{t-1}\ell_t^{\mathsf{T}}(V\ell_{1:t-1} + v_c\mathbf{1}_{t-1}^{\mathsf{T}})e_i\frac{\exp(e_i^{\mathsf{T}}(\ell_{1:t-1}^{\mathsf{T}}a + b_{t-1}))}{\sum_{s=1}^{t-1}\exp(e_s^{\mathsf{T}}(\ell_{1:t-1}^{\mathsf{T}}a + b_{t-1}))}\Bigg|_{a=\mathbf{0}_d, v_c=\mathbf{0}_d, (b_t=\beta\mathbf{1}_t)_{t\in[T-1]}}
$$

$$
= \sum_{i=1}^{t-1}\ell_t\ell_i^{\mathsf{T}}\frac{\exp(e_i^{\mathsf{T}}(\ell_{1:t-1}^{\mathsf{T}}a + b_{t-1}))}{\sum_{s=1}^{t-1}\exp(e_s^{\mathsf{T}}(\ell_{1:t-1}^{\mathsf{T}}a + b_{t-1}))}\Bigg|_{a=\mathbf{0}_d, v_c=\mathbf{0}_d, (b_t=\beta\mathbf{1}_t)_{t\in[T-1]}} = \sum_{i=1}^{t-1}\frac{1}{t-1}\ell_t\ell_i^{\mathsf{T}}.
$$

For $t = 1$, note that $\frac{\partial}{\partial V}\ell_t^{\mathsf{T}}v_c = \mathbf{O}_{d\times d}$, so we can use the same formula as $t \geq 2$ with empty sum $\sum_{i=1}^{t-1}$.

Therefore, we have

$$\frac{\partial f(V, a, (b_t)_{t\in[T-1]}, v_c)}{\partial V}\bigg|_{a=\mathbf{0}_d, v_c=\mathbf{0}_d, (b_t=\beta\mathbf{1}_t)_{t\in[T-1]}}$$

$$= \mathbb{E}\frac{\partial}{\partial V}\left(\sum_{t=1}^{T}\ell_t^\intercal(V\ell_{1:t-1} + v_c\mathbf{1}_{t-1}^\intercal)\texttt{Softmax}(\ell_{1:t-1}^\intercal a + b_{t-1}) + R_\Pi\|\sum_{t=1}^{T}\ell_t\|_2\right)^2\bigg|_{a=\mathbf{0}_d, v_c=\mathbf{0}_d, (b_t=\beta\mathbf{1}_t)_{t\in[T-1]}}$$

$$= \mathbb{E}\bigg[\left(\sum_{t=1}^{T}\ell_t^\intercal(V\ell_{1:t-1} + v_c\mathbf{1}_{t-1}^\intercal)\texttt{Softmax}(\ell_{1:t-1}^\intercal a + b_{t-1}) + R_\Pi\|\sum_{t=1}^{T}\ell_t\|_2\right)\bigg|_{a=\mathbf{0}_d, v_c=\mathbf{0}_d, (b_t=\beta\mathbf{1}_t)_{t\in[T-1]}}$$

$$\frac{\partial}{\partial V}\left(\sum_{t=1}^{T}\ell_t^\intercal(V\ell_{1:t-1} + v_c\mathbf{1}_{t-1}^\intercal)\texttt{Softmax}(\ell_{1:t-1}^\intercal a + b_{t-1}) + R_\Pi\|\sum_{t=1}^{T}\ell_t\|_2\right)\bigg|_{a=\mathbf{0}_d, v_c=\mathbf{0}_d, (b_t=\beta\mathbf{1}_t)_{t\in[T-1]}}\bigg]$$

$$= \mathbb{E}\bigg[\left(\sum_{t=1}^{T}\ell_t^\intercal V\sum_{i=1}^{t-1}\frac{1}{t-1}\ell_i + R_\Pi\|\sum_{t=1}^{T}\ell_t\|_2\right)\sum_{t=1}^{T}\sum_{i=1}^{t-1}\frac{1}{t-1}\ell_t\ell_i^\intercal\bigg]$$

$$= \mathbb{E}\bigg[\left(\sum_{t=1}^{T}\sum_{i=1}^{t-1}\left(\frac{1}{t-1}\ell_t^\intercal V\ell_i\right)\left(\frac{1}{t-1}\ell_t\ell_i^\intercal\right) + R_\Pi T\|\sum_{t'=1}^{T}\ell_{t'}\|_2\ell_t\ell_i^\intercal\right)\bigg]$$

$$= \mathbb{E}\bigg[\left(\sum_{t=1}^{T}\sum_{i=1}^{t-1}\sum_{x=1}^{d}\sum_{y=1}^{d}v_{xy}\ell_{tx}\ell_{iy}\left(\frac{1}{t-1}\right)^2[\ell_{tz}\ell_{iw}]_{(z,w)} + R_\Pi T\|\sum_{t'=1}^{T}\ell_{t'}\|_2\ell_t\ell_i^\intercal\right)\bigg]$$

$$= \sum_{t=1}^{T}\sum_{i=1}^{t-1}\sum_{x=1}^{d}\sum_{y=1}^{d}\frac{1}{(t-1)^2}[\sigma_{xz}v_{xy}\sigma_{yw}]_{(z,w)} + \mathbb{E}\bigg[R_\Pi T\|\sum_{t'=1}^{T}\ell_{t'}\|_2\ell_t\ell_i^\intercal\bigg]$$

$$= \left(\sum_{t=1}^{T-1}\frac{1}{t}\right)\Sigma V\Sigma + \mathbb{E}\bigg[R_\Pi T\|\sum_{t'=1}^{T}\ell_{t'}\|_2\ell_t\ell_i^\intercal\bigg].$$

Therefore, if $V^\star = R_\Pi\frac{T}{\sum_{t=1}^{T-1}1/t}\Sigma^{-1}\mathbb{E}\bigg[\|\sum_{t=1}^{T}\ell_t\|_2\ell_t\ell_i^\intercal\bigg]\Sigma^{-1}$, then $\frac{\partial f}{\partial V}\bigg|_{a=\mathbf{0}_d, v_c=\mathbf{0}_d, (b_t=\beta\mathbf{1}_t)_{t\in[T-1]}, V=V^\star} = \mathbf{O}_{d\times d}$. Lastly,

we have

$$\frac{\partial f}{\partial K}\big|_{K^\intercal(Qc+q_c)=v_c=\mathbf{0}_d, V=V^\star} = \left(\frac{\partial f}{\partial a}\frac{\partial a}{\partial K}\right)\bigg|_{a=\mathbf{0}_d, v_c=\mathbf{0}_d, (b_t=\beta\mathbf{1}_t)_{t\in[T-1]}, V=V^\star} = \mathbf{O}_{d\times d}$$

$$\frac{\partial f}{\partial Q}\big|_{K^\intercal(Qc+q_c)=v_c=\mathbf{0}_d, V=V^\star} = \left(\frac{\partial f}{\partial a}\frac{\partial a}{\partial Q}\right)\bigg|_{a=\mathbf{0}_d, v_c=\mathbf{0}_d, (b_t=\beta\mathbf{1}_t)_{t\in[T-1]}, V=V^\star} = \mathbf{O}_{d\times d}$$

$$\frac{\partial f}{\partial q_c}\big|_{K^\intercal(Qc+q_c)=v_c=\mathbf{0}_d, V=V^\star} = \left(\frac{\partial f}{\partial a}\frac{\partial a}{\partial q_c}\right)\bigg|_{a=\mathbf{0}_d, v_c=\mathbf{0}_d, (b_t=\beta\mathbf{1}_t)_{t\in[T-1]}, V=V^\star} = \mathbf{0}_d$$

which means that such configurations are first-order stationary points of Equation (5.2) with $N = 1$, $h(x) = x^2$, and $\Pi = B(0, R_\Pi, \|\cdot\|)$. $\qquad\square$

### G.8. Deferred Proof of Theorem G.2

**Theorem G.2.** *Consider the policy space $\Pi = B(0, R_\Pi, \|\cdot\|)$ for some $R_\Pi > 0$. The configuration of a single-layer linear self-attention model in Equation (G.6) $(V, K, Q, v_c, k_c, q_c)$ such that $K^\intercal(Qc + q_c) = v_c = \mathbf{0}_d$ and $V = -2R_\Pi\Sigma^{-1}\mathbb{E}\left(\|\sum_{t=1}^{T}\ell_t\|\ell_1\ell_2^\intercal\right)\Sigma^{-1}$ is **a global optimal solution** of Equation (5.2) with $N = 1$, $h(x) = x^2$. Moreover, every global optimal configuration of Equation (5.2) within the parameterization class of Equation (G.6) has the same output function g. Additionally, if $\Sigma$ is a diagonal matrix, then plugging any global optimal configuration into Equation (G.6), and projecting the output with $\texttt{Proj}_{\Pi,\|\cdot\|}$ is equivalent to FTRL with an $L_2$-regularizer.*

*Proof.* The output of the single-layer linear self-attention structure is as follows:

$$g(Z_t; V, K, Q, v_c, k_c, q_c)$$

$$= \sum_{i=1}^{t} \left( V\ell_i\ell_i^\mathsf{T}(K^\mathsf{T}(Qc + q_c)) + (Vk_c^\mathsf{T}(Qc + q_c) + v_c(Qc + q_c)^\mathsf{T}K)\ell_i + v_ck_c^\mathsf{T}(Qc + q_c) \right), \tag{G.9}$$

which can be expressed with a larger class

$$g(Z_t, \mathbb{A}, \beta, \mathbb{C}, \delta) := \sum_{i=1}^{t} (\mathbb{A}\ell_i\ell_i^\mathsf{T}\beta + \mathbb{C}\ell_i + \delta), \tag{G.10}$$

where $\mathbb{A} \in \mathbb{R}^{d \times d}$, $\beta, \mathbb{C}, \delta \in \mathbb{R}^d$. Then, if a minimizer of

$$f(\mathbb{A}, \beta, \mathbb{C}, \delta) := \mathbb{E}\left( \sum_{t=1}^{T} \langle \ell_t, \sum_{i=1}^{t-1} (\mathbb{A}\ell_i\ell_i^\mathsf{T}\beta + \mathbb{C}\ell_i + \delta) \rangle - \inf_{\pi \in \Pi} \left\langle \sum_{t=1}^{T} \ell_t, \pi \right\rangle \right)^2$$

can be expressed as $\mathbb{A} = V, \beta = K^\mathsf{T}(Qc + q_c), \mathbb{C} = Vk_c^\mathsf{T}(Qc + q_c) + v_c(Qc + q_c)^\mathsf{T}K, \beta = v_ck_c^\mathsf{T}(Qc + q_c)$, then we can conclude that the corresponding $V, Q, K, v_c, q_c, k_c$ are also a minimizer of

$$\mathbb{E}\left( \sum_{t=1}^{T} \langle \ell_t, g(Z_{t-1}) \rangle - \inf_{\pi \in \Pi} \left\langle \sum_{t=1}^{T} \ell_i, \pi \right\rangle \right)^2,$$

since the corresponding $V, Q, K, v_c, q_c, k_c$ constitute a minimizer among a larger class. Now, since $\Pi = B(\mathbf{0}_d, R_\Pi, \|\cdot\|)$, we can rewrite $f$ as

$$f(\mathbb{A}, \beta, \mathbb{C}, \delta) = \mathbb{E}\left( \sum_{t=1}^{T} \langle \ell_t, \sum_{i=1}^{t-1} (\mathbb{A}\ell_i\ell_i^\mathsf{T}\beta + \mathbb{C}\ell_i + \delta) \rangle + R_\Pi \left\| \sum_{t=1}^{T} \ell_i \right\|_2 \right)^2. \tag{G.11}$$

**Step 1.** **Finding condition for $\frac{\partial f}{\partial \delta} = 0$.**

Due to the Leibniz rule, if we calculate the partial derivative of Equation (G.11) w.r.t. $\delta$, we have

$$\frac{\partial f(\mathbb{A}, \beta, \mathbb{C}, \delta)}{\partial \delta} = \frac{\partial}{\partial \delta} \mathbb{E}\left( \sum_{t=1}^{T} \langle \ell_t, \sum_{i=1}^{t-1} (\mathbb{A}\ell_i\ell_i^\mathsf{T}\beta + \mathbb{C}\ell_i + \delta) \rangle + R_\Pi \| \sum_{t=1}^{T} \ell_t \|_2 \right)^2$$

$$= \mathbb{E}\frac{\partial}{\partial \delta} \left( \sum_{t=1}^{T} \langle \ell_t, \sum_{i=1}^{t-1} (\mathbb{A}\ell_i\ell_i^\mathsf{T}\beta + \mathbb{C}\ell_i + \delta) \rangle + R_\Pi \| \sum_{t=1}^{T} \ell_t \|_2 \right)^2$$

$$= \mathbb{E}\sum_{t=1}^{T} \ell_t \left( \sum_{t=1}^{T}\sum_{i=1}^{t-1}(t-1)\ell_t^\mathsf{T} (\mathbb{A}\ell_i\ell_i^\mathsf{T}\beta + \mathbb{C}\ell_i + \delta) + R_\Pi \| \sum_{t=1}^{T} \ell_t \| \right). \tag{G.12}$$

Since the expectation of either odd-order polynomial or even-order polynomial times $\|\cdot\|_2$ is 0, due to that $\ell_t$ follows a symmetric distribution, we have

$$\mathbb{E}\sum_{t=1}^{T}(t-1)\ell_t R_\Pi \left\| \sum_{t=1}^{T} \ell_t \right\|_2 = 0, \qquad \mathbb{E}\sum_{t=1}^{T}(t-1)\ell_t \sum_{t=1}^{T}\sum_{i=1}^{t-1} \ell_t^\mathsf{T}\mathbb{C}\ell_i = 0.$$

Now, we calculate

$$\mathbb{E}\sum_{t=1}^{T}(t-1)\ell_t \sum_{t=1}^{T}\sum_{i=1}^{t-1} \ell_t^\mathsf{T}\mathbb{A}\ell_i\ell_i^\mathsf{T}\beta = \mathbb{E}\sum_{t_1=1}^{T}\sum_{t=1}^{T}\sum_{i=1}^{t-1}(t_1-1)\ell_{t_1}\ell_t^\mathsf{T}\mathbb{A}\ell_i\ell_i^\mathsf{T}\beta$$

$$\overset{(i)}{=} \mathbb{E}\sum_{t=1}^{T}\sum_{i=1}^{t-1}(t-1)\ell_t\ell_t^\mathsf{T}\mathbb{A}\ell_i\ell_i^\mathsf{T}\beta = \mathbb{E}\sum_{t=1}^{T}(t-1)^2\ell_t\ell_t^\mathsf{T}\mathbb{A}\Sigma\beta = \frac{1}{6}T(2T^2 - 3T + 1)\Sigma\mathbb{A}\Sigma\beta,$$

where $(i)$ holds since if $t_1 \neq t$, due to the independence of $\ell_t, \ell_{t_1}$, we can use $\mathbb{E}\ell_t = 0$. Lastly,

$$\mathbb{E}\sum_{t=1}^{T}(t-1)\ell_t\sum_{t=1}^{T}\sum_{i=1}^{t-1}\ell_t^\mathsf{T}\delta = \mathbb{E}\sum_{t_1=1}^{T}\sum_{t=1}^{T}(t_1-1)(t-1)\ell_{t_1}\ell_t^\mathsf{T}\delta = \frac{1}{6}T(2T^2-3T+1)\Sigma\delta.$$

Plugging the above equations into Equation (G.12), we have

$$\frac{\partial f(\mathbb{A},\beta,\mathbb{C},\delta)}{\partial\delta} = \frac{1}{6}T(2T^2-3T+1)(\Sigma\mathbb{A}\Sigma\beta+\Sigma\delta).$$

Due to the optimality condition, we have

$$\mathbb{A}\Sigma\beta+\delta = 0. \tag{G.13}$$

**Step 2. Plugging the optimality condition for $\frac{\partial f}{\partial\delta}$ into Equation (G.11).**

Plugging Equation (G.13) to Equation (G.11), $f$ can be written as

$$f(\mathbb{A},\beta,\mathbb{C},-\mathbb{A}\Sigma\beta) = \mathbb{E}\left(\sum_{t=1}^{T}\sum_{i=1}^{t-1}\ell_t^\mathsf{T}\left(\mathbb{A}(\ell_i\ell_i^\mathsf{T}-\Sigma)\beta+\mathbb{C}\ell_i\right)+R_\Pi\left\|\sum_{t=1}^{T}\ell_t\right\|_2\right)^2$$

$$= \underbrace{\mathbb{E}\left(\sum_{t=1}^{T}\sum_{i=1}^{t-1}\ell_t^\mathsf{T}\mathbb{A}(\ell_i\ell_i^\mathsf{T}-\Sigma)\beta\right)^2}_{(i)} + \mathbb{E}\left(\sum_{t=1}^{T}\sum_{i=1}^{t-1}\ell_t^\mathsf{T}\mathbb{C}\ell_i\right)^2 + \mathbb{E}\left(R_\Pi\left\|\sum_{t=1}^{T}\ell_t\right\|_2\right)^2$$

$$+ \underbrace{2\mathbb{E}\left(\sum_{t=1}^{T}\sum_{i=1}^{t-1}\ell_t^\mathsf{T}\mathbb{A}(\ell_i\ell_i^\mathsf{T}-\Sigma)\beta\right)\left(\sum_{t=1}^{T}\sum_{i=1}^{t-1}\ell_t^\mathsf{T}\mathbb{C}\ell_i\right)}_{(ii)}$$

$$+ \underbrace{2\mathbb{E}\left(\sum_{t=1}^{T}\sum_{i=1}^{t-1}\ell_t^\mathsf{T}\mathbb{A}(\ell_i\ell_i^\mathsf{T}-\Sigma)\beta\right)\left(R_\Pi\left\|\sum_{t=1}^{T}\ell_t\right\|_2\right)}_{(iii)}$$

$$+ 2\mathbb{E}\left(\sum_{t=1}^{T}\sum_{i=1}^{t-1}\ell_t^\mathsf{T}\mathbb{C}\ell_i\right)\left(R_\Pi\left\|\sum_{t=1}^{T}\ell_t\right\|_2\right).$$

For the part $(i)$, we have

$$\mathbb{E}\left(\sum_{t=1}^{T}\sum_{i=1}^{t-1}\ell_t^\mathsf{T}\mathbb{A}(\ell_i\ell_i^\mathsf{T}-\Sigma)\beta\right)^2 = \mathbb{E}\left[\sum_{t_1=1}^{T}\sum_{i_1=1}^{t_1-1}\sum_{t=1}^{T}\sum_{i=1}^{t-1}\beta^\mathsf{T}(\ell_{i_1}\ell_{i_1}^\mathsf{T}-\Sigma)\mathbb{A}^\mathsf{T}\ell_{t_1}\ell_t^\mathsf{T}\mathbb{A}(\ell_i\ell_i^\mathsf{T}-\Sigma)\beta\right]$$

$$\underset{(1)}{=} \mathbb{E}\left[\sum_{t=1}^{T}\sum_{i_1=1}^{t-1}\sum_{i=1}^{t-1}\beta^\mathsf{T}(\ell_{i_1}\ell_{i_1}^\mathsf{T}-\Sigma)\mathbb{A}^\mathsf{T}\ell_t\ell_t^\mathsf{T}\mathbb{A}(\ell_i\ell_i^\mathsf{T}-\Sigma)\beta\right]$$

$$\underset{(2)}{=} \mathbb{E}\left[\sum_{t=1}^{T}\sum_{i=1}^{t-1}\beta^\mathsf{T}(\ell_i\ell_i^\mathsf{T}-\Sigma)\mathbb{A}^\mathsf{T}\ell_t\ell_t^\mathsf{T}\mathbb{A}(\ell_i\ell_i^\mathsf{T}-\Sigma)\beta\right]$$

$$= \frac{(T-1)T}{2}\beta^\mathsf{T}\mathbb{E}\left[(\ell_i\ell_i^\mathsf{T}-\Sigma)\mathbb{A}^\mathsf{T}\Sigma\mathbb{A}(\ell_i\ell_i^\mathsf{T}-\Sigma)\right]\beta \tag{G.14}$$

$$= \frac{(T-1)T}{2}\beta^\mathsf{T}\mathbb{E}\left[(\sqrt{\Sigma}A(\ell_i\ell_i^\mathsf{T}-\Sigma))^\mathsf{T}(\sqrt{\Sigma}A(\ell_i\ell_i^\mathsf{T}-\Sigma))\right]\beta.$$

Here, (1) holds because if $t_1 \neq t$, we know that $\mathbb{E}\ell_{t_1} = \mathbb{E}\ell_t = 0$, and they are independent, and (2) holds because if $i_1 \neq i$, we can calculate $\mathbb{E}(\ell_{i_1}\ell_{i_1}^\mathsf{T}-\Sigma) = \boldsymbol{O}_{d\times d}$. In addition, we can easily check that $(ii)$ and $(iii)$ are 0 as they are polynomials of odd degrees and we have $Z \overset{d}{=} -Z$. Note that Equation (G.14) is minimized when $\mathbb{P}(\sqrt{\Sigma}\mathbb{A}(\ell_i\ell_i^\mathsf{T}-\Sigma)\beta = \boldsymbol{0}_d) = 1$.

If $\mathbb{A} \neq \boldsymbol{O}_{d \times d}$, suppose that the singular value decomposition of $A = U\Lambda V$ yields that $\Lambda$ is a diagonal matrix whose first diagonal element is non-zero, and $U, V$ are orthogonal matrices. Then, we want to find $\beta$ that $\sqrt{\Sigma} U \Lambda V (\ell_i \ell_i^{\mathsf{T}} - \Sigma) \beta = \boldsymbol{0}_d$ for any $\ell_i$ such that $p(\ell_i) \neq 0$, where $p$ indicates the probability density function of loss vectors. Since $\Sigma$ and $U$ are invertible, we only need to consider $\Lambda V (\ell_i \ell_i^{\mathsf{T}} - \Sigma) \beta = \boldsymbol{0}_d$. Since $\Lambda$'s first diagonal component is non-zero, we will consider equation $e_1^{\mathsf{T}} \Lambda V (\ell_i \ell_i^{\mathsf{T}} - \Sigma) \beta = 0$. This is equivalent to $V_1 (\ell_i \ell_i^{\mathsf{T}} - \Sigma) \beta = 0$, where $V_1$ is the first row of $V$, and is a non-zero vector.

Now, we will generally consider $a_{x,y}(v) := vv^{\mathsf{T}} x - y$ where $x, y, v \in \mathbb{R}^d$ and $a_{x,y} : B(\boldsymbol{0}_d, 2\epsilon_1, \|\cdot\|) \to \mathbb{R}^d$ function. Then, we can check that the Jacobian of $a_{x,y}(v)$ is $vx^{\mathsf{T}} + (v \cdot x)I$, and we can find that the determinant of the Jacobian is nonzero when $v = \epsilon_1 x$ if $x \neq \boldsymbol{0}_d$. Therefore, the volume of $(V_1 (\ell_i \ell_i^{\mathsf{T}} - \Sigma))$ for $\ell_i \in B(\boldsymbol{0}_d, c_z, \|\cdot\|)$ is greater than the volume of $(V_1 (vv^{\mathsf{T}} - \Sigma))$ for $v \in B(\epsilon_1 V_1^{\mathsf{T}}, \epsilon_2, \|\cdot\|)$, where $c_z$ is a constant such that $B(\boldsymbol{0}_d, c_z, \|\cdot\|) \subseteq \operatorname{supp}(Z)$, and $\epsilon_1, \epsilon_2 > 0$ satisfy that $\epsilon_1 |V_1| + \epsilon_2 < c_z$. Here, we define $\epsilon_2 > 0$ sufficiently small so that the determinant of Jacobian$(vv^{\mathsf{T}} V_1^{\mathsf{T}} - \Sigma V_1^{\mathsf{T}}) > 0$ for $v \in B(\epsilon_1 V_1^{\mathsf{T}}, \epsilon_2, \|\cdot\|)$, and $v \to vv^{\mathsf{T}} V_1^{\mathsf{T}} - \Sigma V_1^{\mathsf{T}}$ is a one-to-one correspondence, by inverse function theorem. Therefore, the volume of $(V_1 (vv^{\mathsf{T}} - \Sigma))$ for $v \in B(\epsilon_1 V_1^{\mathsf{T}}, \epsilon_2, \|\cdot\|)$ can be calculated as

$$[\text{Volume } (V_1 (vv^{\mathsf{T}} - \Sigma)) \text{ for } v \in B(\epsilon_1 V_1^{\mathsf{T}}, \epsilon_2, \|\cdot\|)] = \int_{v \in B(\epsilon_1 V_1^{\mathsf{T}}, \epsilon_2, \|\cdot\|)} \left|\det(\text{Jacobian}(V_1(vv^{\mathsf{T}} - \Sigma)))\right| dv > 0.$$

Therefore, Volume$(V_1(vv^{\mathsf{T}} - \Sigma))$ where $v \in B(\epsilon_1 V_1^{\mathsf{T}}, \epsilon_2, \|\cdot\|)$ is non-zero, so that we can find $d$ loss vectors $\{\ell_i\}_{i \in [d]}$ such that the vectors $\{V_1(\ell_i \ell_i^{\mathsf{T}} - \Sigma)\}_{i \in [d]}$ are linearly independent. Hence, if we want to minimize Equation (G.14), either $A = \boldsymbol{O}_{d \times d}$ or $\beta = \boldsymbol{0}_d$ should hold. In both cases, Equation (G.10) can be re-written as

$$g(Z_t; \mathbb{A}, \beta, \mathbb{C}, \delta) := \sum_{i=1}^{t} \mathbb{C}\ell_i,$$

and this is covered by the original parametrization (Equation (G.9)) with $K^{\mathsf{T}}(Qc + q_c) = v_c = \boldsymbol{0}_d$.

**Step 3. Calculating $\frac{\partial f}{\partial \mathbb{C}}$.**

Now, we optimize over $\mathbb{C}$, by minimizing the following objective:

$$f(\mathbb{C}) := \mathbb{E}\left(\sum_{t=1}^{T}\sum_{i=1}^{t-1} \ell_t^{\mathsf{T}} \mathbb{C}\ell_i + R_\Pi \|\sum_{t=1}^{T} \ell_t\|\right)^2$$

$$= \underbrace{\mathbb{E}\left(\sum_{t=1}^{T}\sum_{i=1}^{t-1} \ell_t^{\mathsf{T}} \mathbb{C}\ell_i\right)^2}_{(i)} + 2\mathbb{E}\left(\left(\sum_{t=1}^{T}\sum_{i=1}^{t-1} \ell_t^{\mathsf{T}} \mathbb{C}\ell_i\right) R_\Pi \|\sum_{t=1}^{T}\ell_t\|\right) + \mathbb{E}\left(R_\Pi \|\sum_{t=1}^{T}\ell_t\|\right)^2$$

$$= \frac{T(T-1)}{2} \operatorname{Tr}\left(\mathbb{C}^{\mathsf{T}} \Sigma \mathbb{C}\Sigma\right) + 2\mathbb{E}\left(B \sum_{t=1}^{T}\sum_{i=1}^{t-1} \ell_t^{\mathsf{T}} \mathbb{C}\ell_i \|\sum_{j=1}^{T}\ell_j\|\right) + \mathbb{E}\left(R_\Pi \|\sum_{t=1}^{T}\ell_t\|\right)^2.$$

Here, $(i)$ can be calculated as follows:

$$\mathbb{E}\left(\sum_{t=1}^{T}\sum_{i=1}^{t-1} \ell_t^{\mathsf{T}} \mathbb{C}\ell_i\right)^2 = \mathbb{E}\left(\sum_{t_1=1}^{T}\sum_{i_1=1}^{t_1-1}\sum_{t=1}^{T}\sum_{i=1}^{t-1} \ell_{i_1}^{\mathsf{T}} \mathbb{C}^{\mathsf{T}} \ell_{t_1} \ell_i^{\mathsf{T}} \mathbb{C}\ell_i\right)$$

$$\underset{(1)}{=} \mathbb{E}\left(\sum_{t=1}^{T}\sum_{i_1=1}^{i-1}\sum_{i=1}^{t-1} \ell_{i_1}^{\mathsf{T}} \mathbb{C}^{\mathsf{T}} \ell_i \ell_i^{\mathsf{T}} \mathbb{C}\ell_i\right) = \mathbb{E}\left(\sum_{t=1}^{T}\sum_{i_1=1}^{i-1}\sum_{i=1}^{t-1} \ell_{i_1}^{\mathsf{T}} \mathbb{C}^{\mathsf{T}} \Sigma \mathbb{C}\ell_i\right)$$

$$\underset{(2)}{=} \mathbb{E}\left(\sum_{t=1}^{T}\sum_{i=1}^{t-1} \ell_k^{\mathsf{T}} \mathbb{C}^{\mathsf{T}} \Sigma \mathbb{C}\ell_i\right) \underset{(3)}{=} \mathbb{E}\operatorname{Tr}\left(\sum_{t=1}^{T}\sum_{i=1}^{t-1} \mathbb{C}^{\mathsf{T}} \Sigma \mathbb{C}\ell_i \ell_k^{\mathsf{T}}\right) = \frac{T(T-1)}{2}\operatorname{Tr}\left(\mathbb{C}^{\mathsf{T}} \Sigma \mathbb{C}\Sigma\right),$$

since (1) holds because if $t_1 \neq t$, we already know that $\mathbb{E}\ell_t = \mathbb{E}\ell_{t_1} = 0$, (2) holds due to a similar reason, and (3) comes from $\operatorname{Tr}(AB) = \operatorname{Tr}(BA)$.

We calculate $\frac{\partial f(\mathbb{C})}{\partial \mathbb{C}}$:

$$\frac{\partial f(\mathbb{C})}{\partial \mathbb{C}} = T(T-1)\Sigma\mathbb{C}\Sigma + 2R_\Pi\mathbb{E}\left(\|\sum_{j=1}^T \ell_j\|\sum_{t=1}^T\sum_{i=1}^{t-1}\ell_t\ell_i^\intercal\right).$$

Hence, the optimal $\mathbb{C} = -\frac{2R_\Pi}{T(T-1)}\Sigma^{-1}\mathbb{E}\left(\|\sum_{j=1}^T \ell_j\|\sum_{t=1}^T\sum_{i=1}^{t-1}\ell_t\ell_i^\intercal\right)\Sigma^{-1}$.

Now, we see that for the special case of $\Sigma = I$, we have $\mathbb{C} = -R_\Pi\mathbb{E}\left(\|\sum_{j=1}^T \ell_j\|\ell_t\ell_i^\intercal\right)$. If we calculate the $(a,b)$-coordinate of $\mathbb{C}$, we need to calculate

$$\mathbb{E}_\ell\left[\sqrt{\sum_{o=1}^d(\sum_{s=1}^T \ell_{so})^2}\ell_{ia}\ell_{kb}\right].$$

If $a \neq b$, then since $Z$ is symmetric, the term above becomes zero. Therefore, we only need to consider the case when $a = b$, which is $\mathbb{E}_\ell\left[\sqrt{\sum_{o=1}^d(\sum_{s=1}^T \ell_{so})^2}\ell_{ia}\ell_{ka}\right]$, and it will be the same value for all $a \in [d]$ since $\ell_i$'s coordinates are independent.

Now, we calculate the scale of $\mathbb{E}_\ell\left[\sqrt{\sum_{o=1}^d(\sum_{s=1}^T \ell_{so})^2}\ell_{i1}\ell_{k1}\right]$. We have $Z := \frac{\sum_{o=1}^{d-1}(\sum_{s=1}^T \ell_{so})^2}{T(d-1)} \overset{a.s.}{\to} 1$ as $d \to \infty$ (by the law of large numbers) and we define $W := \sum_{s\neq i,k}\ell_{s1}/\sqrt{T}$ which is independent of $\ell_{i1}$ and $\ell_{k1}$.

$$\mathbb{E}_\ell\left[\sqrt{\sum_{o=1}^d(\sum_{s=1}^T \ell_{so})^2}\ell_{i1}\ell_{k1}\right] = \mathbb{E}_{Z,W,\ell_{i1},\ell_{k1}}\left[\sqrt{T(d-1)Z + (\sqrt{T}W + \ell_{i1} + \ell_{k1})^2}\ell_{i1}\ell_{k1}\right]$$

$$= \mathbb{E}_{Z,W,\ell_{i1},\ell_{k1}\geq 0}\left[\sqrt{T(d-1)Z + (\sqrt{T}W + \ell_{i1} + \ell_{k1})^2}\ell_{i1}\ell_{k1} - \sqrt{T(d-1)Z + (\sqrt{T}W + \ell_{i1} - \ell_{k1})^2}\ell_{i1}\ell_{k1}\right]$$

$$= \mathbb{E}_{Z,W,\ell_{i1},\ell_{k1}\geq 0}\left[\frac{4(\sqrt{T}W + \ell_{i1})\ell_{k1}}{\sqrt{T(d-1)Z + (\sqrt{T}W + \ell_{i1} + \ell_{k1})^2} + \sqrt{T(d-1)Z + (\sqrt{T}W + \ell_{i1} - \ell_{k1})^2}}\ell_{i1}\ell_{k1}\right].$$

Taking $d \to \infty$, we have

$$\frac{\sqrt{T(d-1)Z + (\sqrt{T}W + \ell_{i1} + \ell_{k1})^2} + \sqrt{T(d-1)Z + (\sqrt{T}W + \ell_{i1} - \ell_{k1})^2}}{2\sqrt{Td}} \overset{d}{\to} 1,$$

which further implies

$$\sqrt{Td}\frac{4(\sqrt{T}W + \ell_{i1})\ell_{k1}}{\sqrt{T(d-1)Z + (\sqrt{T}W + \ell_{i1} + \ell_{k1})^2} + \sqrt{T(d-1)Z + (\sqrt{T}W + \ell_{i1} - \ell_{k1})^2}}\ell_{i1}\ell_{k1}$$

$$\overset{d}{\to} \sqrt{Td}\frac{4(\sqrt{T}W + \ell_{i1})\ell_{k1}}{2\sqrt{Td}}\ell_{i1}\ell_{k1} = 2(\sqrt{T}W + \ell_{i1})\ell_{i1}\ell_{k1}$$

as $d \to \infty$. Therefore,

$$\lim_{d\to\infty}\mathbb{E}_{Z,W,\ell_{i1},\ell_{k1}\geq 0}\left[\sqrt{Td}\frac{4(\sqrt{T}W + \ell_{i1})\ell_{k1}}{\sqrt{T(d-1)Z + (\sqrt{T}W + \ell_{i1} + \ell_{k1})^2} + \sqrt{T(d-1)Z + (\sqrt{T}W + \ell_{i1} - \ell_{k1})^2}}\ell_{i1}\ell_{k1}\right]$$

$$= \mathbb{E}_{Z,W,\ell_{i1},\ell_{k1}\geq 0}\left[2(\sqrt{T}W + \ell_{i1})\ell_{i1}\ell_{k1}\right] = \mathbb{E}_{\ell_{i1},\ell_{k1}\geq 0}\left[\ell_{i1}^2\ell_{k1}\right]$$

which is a constant. The last equality came from the fact that $W, \ell_{i1}, \ell_{k1}$ are independent random variables, and expectation of $\ell_{i1}$ is zero. Therefore, the output of the single-layer linear self-attention provides us with online gradient descent with step-size $\Theta(R_\Pi/\sqrt{Td})$. In the online learning literature, we usually set the gradient step size as $\Theta(R_\Pi/\sqrt{Td})$ (Hazan, 2016, Theorem 3.1), which is consistent with the result above. $\qquad\square$

## G.9. Empirical Validation of Theorem G.3 and Theorem G.2

We now provide empirical validations for Theorem G.3 and Theorem G.2. We provide the training details and the results as follows.

### G.9.1. EMPIRICAL VALIDATION OF THEOREM G.3

Our model architecture is defined as follows: the number of layers $T$ is set to 30 and the dimensionality $d$ to 32, with the loss vector $\ell_i$'s distribution $Z$ following a standard normal distribution $\mathcal{N}(0, 1)$. During training, we conducted 40,000 epochs with a batch size of 512. We employed the Adam optimizer, setting the learning rate to 0.001. We initialized the value, query, and key vectors ($v_c, q_c, k_c$) as zero vectors.

Our empirical analysis aims to demonstrate that the optimized model inherently emulates online gradient descent. To illustrate this, we will focus on two key convergence properties: $K^{\mathsf{T}}Q$ approaching the zero matrix $\boldsymbol{O}_{d \times d}$ and $V$ converging to $a\mathbf{1}_d\mathbf{1}_d^{\mathsf{T}} + bI_{d \times d}$, where $a$ and $b$ are constants in $\mathbb{R}$. The conditions $K^{\mathsf{T}}Q = \boldsymbol{O}_{d \times d}$ and $V = a\mathbf{1}_d\mathbf{1}_d^{\mathsf{T}} + bI_{d \times d}$ imply that the function $g(Z_t; V, Q, K) = \sum_{i=1}^{t}(b - a)\ell_i$, effectively emulating the process of an online gradient descent method. We repeated 10 times of the experiments. For verifying $K^{\mathsf{T}}Q = \boldsymbol{O}_{d \times d}$, we will measure Frobenius norm ($\|\cdot\|_F$) of $K^{\mathsf{T}}Q$. Also for measuring the closeness of $V$ and $a\mathbf{1}_d\mathbf{1}_d^{\mathsf{T}} + bI_{d \times d}$, we will measure $\min_{a,b\in\mathbb{R}} \|V - (a\mathbf{1}_d\mathbf{1}_d^{\mathsf{T}} + bI_{d \times d})\|_F/b$. The results are demonstrated in the first plot of Figure G.1.

### G.9.2. EMPIRICAL VALIDATION OF THEOREM G.2

We now focus on two key convergence properties: $K^{\mathsf{T}}(Q\mathbf{1}_d + q_c)$ approaching the zero vector $\mathbf{0}_d$ and $V$ converging to $a\mathbf{1}_d\mathbf{1}_d^{\mathsf{T}} + bI_{d \times d}$, where $a$ and $b$ are constants in $\mathbb{R}$. The conditions $K^{\mathsf{T}}(Q\mathbf{1}_d + q_c) = \mathbf{0}_d$ and $V = a\mathbf{1}_d\mathbf{1}_d^{\mathsf{T}} + bI_{d \times d}$ imply that the function $g(Z_t; V, Q, K) = \sum_{i=1}^{t}(b - a)\ell_i$, effectively emulating the process of an online gradient descent method. We repeated 10 times. For verifying $K^{\mathsf{T}}(Q\mathbf{1}_d + q_c) = \mathbf{0}_d$, we will measure 2-norm of $K^{\mathsf{T}}(Q\mathbf{1}_d + q_c)$. Also for measuring the closeness of $V$ and $a\mathbf{1}_d\mathbf{1}_d^{\mathsf{T}} + bI_{d \times d}$, we will measure $\min_{a,b\in\mathbb{R}} \|V - (a\mathbf{1}_d\mathbf{1}_d^{\mathsf{T}} + bI_{d \times d})\|_F/b$. The results are demonstrated in the second plot of Figure G.1.

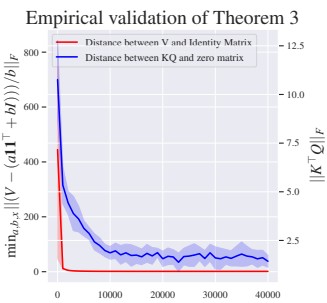 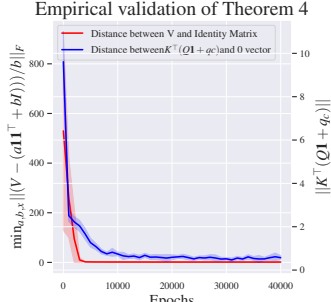 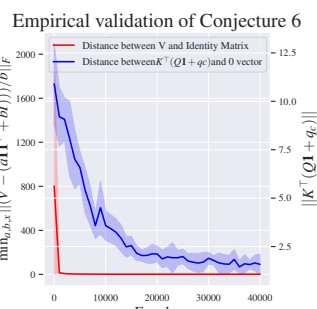

*Figure G.1.* Empirical validation of Theorem G.3 (top), Theorem G.2 (middle), and Conjecture 4 (bottom). The observed convergence in Theorem G.3 and Conjecture 4's result suggests that configuration in Theorem G.3 and Conjecture 4 are not only the local optimal point, but it has the potential as being the global optimizer.

## G.10. Discussions on the Production of FTRL with Entropy Regularization

Now, we will consider projecting a single-layer linear self-attention model into a constrained domain such as a simplex, which is more amenable to the Experts Problem setting. To this end, we consider the following parameterization by adding an additional *non-linear* structure for the single-layer linear self-attention:

$$g(Z_t; V, K, Q, v_c, k_c, q_c) = \texttt{Operator}\left(\sum_{i=1}^{t}(V\ell_i + v_c)((K\ell_i + k_c))^{\mathsf{T}} \cdot (Qc + q_c)\right), \tag{G.15}$$

where the `Operator` denotes projection to the convex set.

**Conjecture 4.** *Assume $\Sigma = I$. Then, the configuration that $K^{\mathsf{T}}(Qc + q_c) = v_c = \mathbf{0}_d$ and $V = \widetilde{\Omega}\left(-\frac{1}{\sqrt{nd}}\right)I_{d \times d}$ is a*

*first-order stationary point of Equation (5.2) with $N = 1$ and $h(x) = x^2$ when $\text{LLM}_\theta$ is parameterized with Equation (G.15), Operator = Softmax, and $\Pi = \Delta(\mathcal{A})$. This configuration performs FTRL with an entropy regularizer which is a no-regret algorithm.*

We provide an idea for proving the conjecture, together with its numerical validation. Also, we have observed in Figure G.1 that Theorem G.3 and Conjecture 4 might also be a global optimizer, as training results have provided the configuration that Theorem G.3 and Conjecture 4 have suggested.

To be specific, we will consider

$$f(V, a, \beta, v_c) = \mathbb{E}\left(\sum_{t=1}^{T}\sum_{s=1}^{d}\ell_{ts}\frac{\exp\left(e_s^\intercal \sum_{j=1}^{t-1}(V\ell_j\ell_j^\intercal a + (\beta V + v_c a^\intercal)\ell_j + v_c\beta)\right)}{\sum_{y=1}^{d}\exp\left(e_y^\intercal \sum_{j=1}^{t-1}(V\ell_j\ell_j^\intercal a + (\beta V + v_c a^\intercal)\ell_j + v_c\beta)\right)} - \min_s \sum_{t=1}^{T}\ell_{ts}\right)^2$$

and will try to prove that $a = \mathbf{0}_d, v_c = v\mathbf{1}_d, V = kI$ is a first-order stationary point.

**Step 1.** Calculating $\frac{\partial f}{\partial v_c}$.

We use the following formula: for $x \in [d]$ and $t \geq 2$, we have

$$\frac{\partial}{\partial v_{cx}}\exp\left(e_y^\intercal \sum_{i=1}^{t}(V\ell_i\ell_i^\intercal a + (\beta V + v_c a^\intercal)\ell_i + v_c\beta)\right)\Bigg|_{a=\mathbf{0}_d, v_c=v\mathbf{1}_d, V=kI}$$

$$= \exp\left(e_y^\intercal \sum_{i=1}^{t}(V\ell_i\ell_i^\intercal a + (\beta V + v_c a^\intercal)\ell_i + v_c\beta)\right)\frac{\partial}{\partial v_{cx}}\left(e_y^\intercal \sum_{i=1}^{t}(V\ell_i\ell_i^\intercal a + (\beta V + v_c a^\intercal)\ell_i + v_c\beta)\right)\Bigg|_{a=\mathbf{0}_d, v_c=v\mathbf{1}_d, V=kI}$$

$$= \exp\left(e_y^\intercal \sum_{i=1}^{t}(V\ell_i\ell_i^\intercal a + (\beta V + v_c a^\intercal)\ell_i + v_c\beta)\right)\sum_{i=1}^{t}(a^\intercal \ell_i\ell_i^\intercal e_x + \beta)\Bigg|_{a=\mathbf{0}_d, v_c=v\mathbf{1}_d, V=kI}$$

$$= t\beta\exp(v\beta)\exp(\beta k\sum_{i=1}^{t}\ell_{iy}),$$

and for $t = 1$, $\frac{\partial}{\partial v_{cx}}\exp\left(e_y^\intercal \sum_{i=1}^{t}(V\ell_i\ell_i^\intercal a + (\beta V + v_c a^\intercal)\ell_i + v_c\beta)\right)\Bigg|_{a=\mathbf{0}_d, v_c=v\mathbf{1}_d, V=kI} = 0$, so we can use the same formula with $t \geq 2$. Thus, we have

$$\frac{\partial}{\partial v_{cx}}\left(\sum_{t=1}^{T}\sum_{s=1}^{d}\ell_{ts}\frac{\exp\left(e_s^\intercal \sum_{j=1}^{t-1}(V\ell_j\ell_j^\intercal a + (\beta V + v_c a^\intercal)\ell_j + v_c\beta)\right)}{\sum_{y=1}^{d}\exp\left(e_y^\intercal \sum_{j=1}^{t-1}(V\ell_j\ell_j^\intercal a + (\beta V + v_c a^\intercal)\ell_j + v_c\beta)\right)} - \min_s \sum_{t=1}^{T}\ell_{ts}\right)\Bigg|_{a=\mathbf{0}_d, v_c=v\mathbf{1}_d, V=kI}$$

$$= \beta\exp(v\beta)$$

$$\sum_{t=1}^{T}t\sum_{s=1}^{d}\ell_{ts}\frac{\sum_{y=1}^{d}\exp\left(\sum_{j=1}^{t-1}\beta k\ell_{jy}\right)\exp\left(\sum_{j=1}^{t-1}\beta k\ell_{js}\right) - \sum_{y=1}^{d}\exp\left(\sum_{j=1}^{t-1}\beta k\ell_{js}\right)\exp\left(\sum_{j=1}^{t-1}\beta k\ell_{jy}\right)}{\left(\sum_{y=1}^{d}\exp\left(e_y^\intercal \sum_{j=1}^{t-1}\beta V\ell_j\right)\right)^2}$$

$$= 0.$$

Therefore,

$$\frac{\partial f(V, a, \beta, v_c)}{\partial v_{cx}}\Bigg|_{a=\mathbf{0}_d, v_c=v\mathbf{1}_d, V=kI}$$

$$= \mathbb{E}\Bigg[\left(\sum_{t=1}^{T}\sum_{s=1}^{d}\ell_{ts}\frac{\exp\left(e_s^\intercal \sum_{j=1}^{t-1}(V\ell_j\ell_j^\intercal a + (\beta V + v_c a^\intercal)\ell_j + v_c\beta)\right)}{\sum_{y=1}^{d}\exp\left(e_y^\intercal \sum_{j=1}^{t-1}(V\ell_j\ell_j^\intercal a + (\beta V + v_c a^\intercal)\ell_j + v_c\beta)\right)} - \min_s \sum_{t=1}^{T}\ell_{ts}\right)$$

$$\frac{\partial}{\partial v_{cx}}\left(\sum_{t=1}^{T}\sum_{s=1}^{d}\ell_{ts}\frac{\exp\left(e_s^\intercal \sum_{j=1}^{t-1}(V\ell_j\ell_j^\intercal a + (\beta V + v_c a^\intercal)\ell_j + v_c\beta)\right)}{\sum_{y=1}^{d}\exp\left(e_y^\intercal \sum_{j=1}^{t-1}(V\ell_j\ell_j^\intercal a + (\beta V + v_c a^\intercal)\ell_j + v_c\beta)\right)} - \min_s \sum_{t=1}^{T}\ell_{ts}\right)\Bigg]\Bigg|_{a=\mathbf{0}_d, v_c=v\mathbf{1}_d, V=kI}$$

$$= 0.$$

**Step 2.** Calculating $\frac{\partial f}{\partial V}$.

The following formula will be used for calculating $\frac{\partial f}{\partial V}\Big|_{a=\mathbf{0}_d, v_c=v\mathbf{1}_d, V=kI}$ : for $r, c \in [d]$, we have

$$
\frac{\partial}{\partial V_{rc}} \exp\left( e_y^\intercal \sum_{i=1}^{t} (V\ell_i\ell_i^\intercal a + (\beta V + v_c a^\intercal)\ell_i + v_c\beta) \right)\Bigg|_{a=\mathbf{0}_d, v_c=v\mathbf{1}_d, V=kI}
$$

$$
= \exp\left( e_y^\intercal \sum_{i=1}^{t} (V\ell_i\ell_i^\intercal a + (\beta V + v_c a^\intercal)\ell_i + v_c\beta) \right) \frac{\partial}{\partial V_{rc}} \left( e_y^\intercal \sum_{i=1}^{t} (V\ell_i\ell_i^\intercal a + (\beta V + v_c a^\intercal)\ell_i + v_c\beta) \right)\Bigg|_{a=\mathbf{0}_d, v_c=v\mathbf{1}_d, V=kI}
$$

$$
= \exp\left( \sum_{i=1}^{t} k\beta\ell_{iy} + v\beta \right) \sum_{i=1}^{t} \beta\mathbf{1}(y=r)\ell_{ic}.
$$

Therefore,

$$
\frac{\partial f(V, a, \beta, v_c)}{\partial V_{rc}}\Bigg|_{a=\mathbf{0}_d, v_c=v\mathbf{1}_d, V=kI}
$$

$$
= \mathbb{E}\Bigg[ \left( \sum_{t=1}^{T}\sum_{s=1}^{d} \ell_{ts} \frac{\exp\left( e_s^\intercal \sum_{j=1}^{t-1}(V\ell_j\ell_j^\intercal a + (\beta V + v_c a^\intercal)\ell_j + v_c\beta) \right)}{\sum_{y=1}^{d} \exp\left( e_y^\intercal \sum_{j=1}^{t-1}(V\ell_j\ell_j^\intercal a + (\beta V + v_c a^\intercal)\ell_j + v_c\beta) \right)} - \min_s \sum_{t=1}^{T} \ell_{ts} \right)
$$

$$
\frac{\partial}{\partial V_{rc}}\left( \sum_{t=1}^{T}\sum_{s=1}^{d} \ell_{ts} \frac{\exp\left( e_s^\intercal \sum_{j=1}^{t-1}(V\ell_j\ell_j^\intercal a + (\beta V + v_c a^\intercal)\ell_j + v_c\beta) \right)}{\sum_{y=1}^{d} \exp\left( e_y^\intercal \sum_{j=1}^{t-1}(V\ell_j\ell_j^\intercal a + (\beta V + v_c a^\intercal)\ell_j + v_c\beta) \right)} - \min_s \sum_{t=1}^{T} \ell_{ts} \right)\Bigg]\Bigg|_{a=\mathbf{0}_d, v_c=v\mathbf{1}_d, V=kI}
$$

$$
= \mathbb{E}\Bigg[ \left( \sum_{t=1}^{T}\sum_{s=1}^{d} \ell_{ts} \frac{\exp\left( \sum_{j=1}^{t-1}\beta k\ell_{js} + v\beta \right)}{\sum_{y=1}^{d}\exp\left( \sum_{j=1}^{t-1}\beta V\ell_{jy} + v\beta \right)} - \min_s \sum_{t=1}^{T} \ell_{ts} \right)
$$

$$
\left( \sum_{t=1}^{T}\sum_{s=1}^{d} \ell_{ts} \frac{\sum_{j=1}^{t-1}\beta\mathbf{1}(s=r)\ell_{jc}\exp\left( \sum_{j=1}^{t-1}\beta k\ell_{js} + v\beta \right)\sum_{y=1}^{d}\exp\left( \sum_{j=1}^{t-1}\beta k\ell_{jy} + v\beta \right)}{\left( \sum_{y=1}^{d}\exp\left( \sum_{j=1}^{t-1}\beta k\ell_{jy} + v\beta \right) \right)^2} \right.
$$

$$
\left. - \sum_{t=1}^{T}\sum_{s=1}^{d} \ell_{ts} \frac{\exp\left( \sum_{j=1}^{t-1}\beta k\ell_{js} + v\beta \right)\sum_{y=1}^{d}\left( \sum_{j=1}^{t-1}\beta\mathbf{1}(y=r)\ell_{jc}\exp\left( \sum_{j=1}^{t-1}\beta k\ell_{jy} + v\beta \right) \right)}{\left( \sum_{y=1}^{d}\exp\left( \sum_{j=1}^{t-1}\beta k\ell_{jy} + v\beta \right) \right)^2} \right)\Bigg]
$$

$$
= \beta\mathbb{E}\Bigg[ \left( \sum_{t=1}^{T}\sum_{s=1}^{d} \ell_{ts} \frac{\exp\left( \sum_{j=1}^{t-1}\beta k\ell_{js} \right)}{\sum_{y=1}^{d}\exp\left( \sum_{j=1}^{t-1}\beta V\ell_{jy} \right)} - \min_s \sum_{t=1}^{T} \ell_{ts} \right)
$$

$$
\left( \underbrace{\frac{\sum_{t=1}^{T}\sum_{j=1}^{t-1}\sum_{y=1}^{d}\ell_{tr}\ell_{jc}\exp\left( \beta k\sum_{j=1}^{t-1}\ell_{jr} \right)\exp\left( \beta k\sum_{j=1}^{t-1}\ell_{jy} \right)}{\left( \sum_{y=1}^{d}\exp\left( \beta k\sum_{j=1}^{t-1}\ell_{jy} \right) \right)^2}}_{(i)} \right.
$$

$$
\left. \underbrace{- \frac{\sum_{t=1}^{T}\sum_{j=1}^{t-1}\sum_{y=1}^{d}\ell_{ty}\ell_{jc}\exp\left( \beta k\sum_{j=1}^{t-1}\ell_{jr} \right)\exp\left( \beta k\sum_{j=1}^{t-1}\ell_{jy} \right)}{\left( \sum_{y=1}^{d}\exp\left( \beta k\sum_{j=1}^{t-1}\ell_{jy} \right) \right)^2}}_{(ii)} \right)\Bigg].
$$

We can observe the followings: 1) if $r_1 \neq c_1$ and $r_2 \neq c_2$, $\frac{\partial f}{\partial V_{r_1 c_1}}\Big|_{a=\mathbf{0}_d, v_c=v\mathbf{1}_d, V=kI} = \frac{\partial f}{\partial V_{r_2 c_2}}\Big|_{a=\mathbf{0}_d, v_c=v\mathbf{1}_d, V=kI}$ holds,

and 2) $\frac{\partial f}{\partial V_{r_1 r_1}}\Big|_{a=\mathbf{0}_d, v_c=v\mathbf{1}_d, V=kI} = \frac{\partial f}{\partial V_{r_2 r_2}}\Big|_{a=\mathbf{0}_d, v_c=v\mathbf{1}_d, V=kI}$.

**Step 3.** Calculating $\frac{\partial f}{\partial \beta}$.

The following formula will be used for calculating $\frac{\partial f}{\partial \beta}\Big|_{a=\mathbf{0}_d, v_c=v\mathbf{1}_d, V=kI}$ :

$$
\frac{\partial}{\partial \beta} \exp\left( e_y^{\mathsf{T}} \sum_{i=1}^{t} (V\ell_i\ell_i^{\mathsf{T}} a + (\beta V + v_c a^{\mathsf{T}})\ell_i + v_c\beta) \right)\Bigg|_{a=\mathbf{0}_d, v_c=v\mathbf{1}_d, V=kI}
$$

$$
= \exp\left( e_y^{\mathsf{T}} \sum_{i=1}^{t} (V\ell_i\ell_i^{\mathsf{T}} a + (\beta V + v_c a^{\mathsf{T}})\ell_i + v_c\beta) \right) \frac{\partial}{\partial \beta}\left( e_y^{\mathsf{T}} \sum_{i=1}^{t} (V\ell_i\ell_i^{\mathsf{T}} a + (\beta V + v_c a^{\mathsf{T}})\ell_i + v_c\beta) \right)\Bigg|_{a=\mathbf{0}_d, v_c=v\mathbf{1}_d, V=kI}
$$

$$
= tv\beta \exp\left( \sum_{i=1}^{t} k\beta\ell_{iy} + v\beta \right).
$$

Further, we have

$$
\frac{\partial}{\partial \beta}\left( \sum_{t=1}^{T}\sum_{s=1}^{d} \ell_{ts} \frac{\exp\left( e_s^{\mathsf{T}} \sum_{j=1}^{t-1}(V\ell_j\ell_j^{\mathsf{T}} a + (\beta V + v_c a^{\mathsf{T}})\ell_j + v_c\beta) \right)}{\sum_{y=1}^{d}\exp\left( e_y^{\mathsf{T}} \sum_{j=1}^{t-1}(V\ell_j\ell_j^{\mathsf{T}} a + (\beta V + v_c a^{\mathsf{T}})\ell_j + v_c\beta) \right)} - \min_s \sum_{t=1}^{T} \ell_{ts} \right)\Bigg|_{a=\mathbf{0}_d, v_c=v\mathbf{1}_d, V=kI}
$$

$$
= v\beta \exp(v\beta)
$$

$$
\sum_{t=1}^{T} t \sum_{s=1}^{d} \ell_{ts} \frac{\sum_{y=1}^{d}\exp\left( \sum_{j=1}^{t-1}\beta k\ell_{jy} \right)\exp\left( \sum_{j=1}^{t-1}\beta k\ell_{js} \right) - \sum_{y=1}^{d}\exp\left( \sum_{j=1}^{t-1}\beta k\ell_{js} \right)\exp\left( \sum_{j=1}^{t-1}\beta k\ell_{jy} \right)}{\left( \sum_{y=1}^{d}\exp\left( e_y^{\mathsf{T}} \sum_{j=1}^{t-1}\beta V\ell_j \right) \right)^2}
$$

$$
= 0.
$$

**Step 4.** Calculating $\frac{\partial f}{\partial a}$.

Note that

$$
\frac{\partial}{\partial a_x} \exp\left( e_y^{\mathsf{T}} \sum_{i=1}^{t} (V\ell_i\ell_i^{\mathsf{T}} a + (\beta V + v_c a^{\mathsf{T}})\ell_i + v_c\beta) \right)\Bigg|_{a=\mathbf{0}_d, v_c=v\mathbf{1}_d, V=kI}
$$

$$
= \exp\left( e_y^{\mathsf{T}} \sum_{i=1}^{t} (V\ell_i\ell_i^{\mathsf{T}} a + (\beta V + v_c a^{\mathsf{T}})\ell_i + v_c\beta) \right) \frac{\partial}{\partial a_x}\left( e_y^{\mathsf{T}} \sum_{i=1}^{t} (V\ell_i\ell_i^{\mathsf{T}} a + (\beta V + v_c a^{\mathsf{T}})\ell_i + v_c\beta) \right)\Bigg|_{a=\mathbf{0}_d, v_c=v\mathbf{1}_d, V=kI}
$$

$$
= \exp\left( e_y^{\mathsf{T}} \sum_{i=1}^{t} (V\ell_i\ell_i^{\mathsf{T}} a + (\beta V + v_c a^{\mathsf{T}})\ell_i + v_c\beta) \right) \sum_{i=1}^{t} \left( e_y^{\mathsf{T}} V\ell_i\ell_i^{\mathsf{T}} e_x + e_y^{\mathsf{T}} v_c\ell_i^{\mathsf{T}} e_x \right)\Bigg|_{a=\mathbf{0}_d, v_c=v\mathbf{1}_d, V=kI}
$$

$$
= \exp\left( \sum_{i=1}^{t} \beta k\ell_{iy} + v\beta \right) \sum_{i=1}^{t} (k\ell_{iy}\ell_{ix} + v\ell_{ix}).
$$

Therefore,

$$\frac{\partial f(V, a, \beta, v_c)}{\partial a_x}\bigg|_{a=\mathbf{0}_d, v_c=v\mathbf{1}_d, V=kI}$$

$$= \mathbb{E}\left[\left(\sum_{t=1}^{T}\sum_{s=1}^{d}\ell_{ts}\frac{\exp\left(e_s^{\mathsf{T}}\sum_{j=1}^{t-1}(V\ell_j\ell_j^{\mathsf{T}}a + (\beta V + v_c a^{\mathsf{T}})\ell_j + v_c\beta)\right)}{\sum_{y=1}^{d}\exp\left(e_y^{\mathsf{T}}\sum_{j=1}^{t-1}(V\ell_j\ell_j^{\mathsf{T}}a + (\beta V + v_c a^{\mathsf{T}})\ell_j + v_c\beta)\right)} - \min_s\sum_{t=1}^{T}\ell_{ts}\right)\right.$$

$$\left.\frac{\partial}{\partial a_x}\left(\sum_{t=1}^{T}\sum_{s=1}^{d}\ell_{ts}\frac{\exp\left(e_s^{\mathsf{T}}\sum_{j=1}^{t-1}(V\ell_j\ell_j^{\mathsf{T}}a + (\beta V + v_c a^{\mathsf{T}})\ell_j + v_c\beta)\right)}{\sum_{y=1}^{d}\exp\left(e_y^{\mathsf{T}}\sum_{j=1}^{t-1}(V\ell_j\ell_j^{\mathsf{T}}a + (\beta V + v_c a^{\mathsf{T}})\ell_j + v_c\beta)\right)} - \min_s\sum_{t=1}^{T}\ell_{ts}\right)\right]\bigg|_{a=\mathbf{0}_d, v_c=v\mathbf{1}_d, V=kI}$$

$$= \mathbb{E}\left[\left(\sum_{t=1}^{T}\sum_{s=1}^{d}\ell_{ts}\frac{\exp\left(\sum_{j=1}^{t-1}\beta k\ell_{js}\right)}{\sum_{y=1}^{d}\exp\left(\sum_{j=1}^{t-1}\beta k\ell_{jy}\right)} - \min_s\sum_{t=1}^{T}\ell_{ts}\right)\right.$$

$$\left(\sum_{t=1}^{T}\sum_{s=1}^{d}\ell_{ts}\frac{\sum_{j=1}^{t-1}(k\ell_{js}\ell_{jx} + v\ell_{jx})\exp\left(\sum_{j=1}^{t-1}\beta k\ell_{js}\right)\sum_{y=1}^{d}\exp\left(\sum_{j=1}^{t-1}\beta k\ell_{jy}\right)}{\left(\sum_{y=1}^{d}\exp\left(\sum_{j=1}^{t-1}\beta k\ell_{jy}\right)\right)^2}\right.$$

$$\left.\left. -\sum_{t=1}^{T}\sum_{s=1}^{d}\ell_{ts}\frac{\exp\left(\sum_{j=1}^{t-1}\beta k\ell_{js}\right)\sum_{y=1}^{d}\left(\sum_{j=1}^{t-1}(k\ell_{jy}\ell_{jx} + v\ell_{jx})\exp\left(\sum_{j=1}^{t-1}\beta k\ell_{jy}\right)\right)}{\left(\sum_{y=1}^{d}\exp\left(\sum_{j=1}^{t-1}\beta k\ell_{jy}\right)\right)^2}\right)\right]$$

$$= \mathbb{E}\left[k\left(\sum_{t=1}^{T}\sum_{s=1}^{d}\ell_{ts}\frac{\exp\left(\sum_{j=1}^{t-1}\beta k\ell_{js}\right)}{\sum_{y=1}^{d}\exp\left(\sum_{j=1}^{t-1}\beta k\ell_{jy}\right)} - \min_s\sum_{t=1}^{T}\ell_{ts}\right)\right.$$

$$\left(\sum_{t=1}^{T}\sum_{s=1}^{d}\ell_{ts}\frac{\sum_{j=1}^{t-1}\ell_{js}\ell_{jx}\exp\left(\sum_{j=1}^{t-1}\beta k\ell_{js}\right)\sum_{y=1}^{d}\exp\left(\sum_{j=1}^{t-1}\beta k\ell_{jy}\right)}{\left(\sum_{y=1}^{d}\exp\left(\sum_{j=1}^{t-1}\beta k\ell_{jy}\right)\right)^2}\right.$$

$$\left.\left. -\sum_{t=1}^{T}\sum_{s=1}^{d}\ell_{ts}\frac{\exp\left(\sum_{j=1}^{t-1}\beta k\ell_{js}\right)\sum_{y=1}^{d}\left(\sum_{j=1}^{t-1}\ell_{jy}\ell_{jx}\exp\left(\sum_{j=1}^{t-1}\beta k\ell_{jy}\right)\right)}{\left(\sum_{y=1}^{d}\exp\left(\sum_{j=1}^{t-1}\beta k\ell_{jy}\right)\right)^2}\right)\right]$$

Note that the value does not depend on $x$, which means that $\frac{\partial f}{\partial a}\bigg|_{a=\mathbf{0}_d, v_c=v\mathbf{1}_d, V=kI} = \widetilde{c}\mathbf{1}_d$ for some constant $\widetilde{c}$.

### G.10.1. NUMERICAL ANALYSIS OF STEP 2 AND STEP 4

In Steps 2 and 4 above, we were not able to show that a $k$ whose value becomes zero exists. We hence provide some empirical evidence here. First, we attach the estimated $\frac{\partial f}{\partial V_{rc}}\bigg|_{a=\mathbf{0}_d, v_c=v\mathbf{1}_d, V=kI}$ $(r \neq c)$, $\frac{\partial f}{\partial V_{rr}}\bigg|_{a=\mathbf{0}_d, v_c=v\mathbf{1}_d, V=kI}$, $\frac{\partial f}{\partial a_x}\bigg|_{a=\mathbf{0}_d, v_c=v\mathbf{1}_d, V=kI}$ and $\frac{\partial f}{\partial a_x}\bigg|_{a=\mathbf{0}_d, v_c=v\mathbf{1}_d, V=kI}$ graph with respect to $k$ value when $\ell_{ts} \sim \text{Unif}([0,1])$ for all $t \in [T], s \in [d]$. While the graph of $\frac{\partial f}{\partial V}\bigg|_{a=\mathbf{0}_d, v_c=v\mathbf{1}_d, V=kI}$ is not stable, we can see that $k$ for $\frac{\partial f}{\partial V_{rc}}\bigg|_{a=\mathbf{0}_d, v_c=v\mathbf{1}_d, V=kI} = 0$, $\frac{\partial f}{\partial V_{rr}}\bigg|_{a=\mathbf{0}_d, v_c=v\mathbf{1}_d, V=kI} = 0$ and $\frac{\partial f}{\partial a_x}\bigg|_{a=\mathbf{0}_d, v_c=v\mathbf{1}_d, V=kI} = 0$ is very similar in Figure G.2. We used the Monte Carlo estimation of $1,000,000$ times.

### G.10.2. EMPIRICAL VALIDATION

Our model architecture is defined as follows: the number of layers $T$ is set to 30 and the dimensionality $d$ to 32, with the loss vector $l_i$'s distribution $Z$ following a standard normal distribution $\mathcal{N}(0, 1)$. During training, we conducted 40,000 epochs with a batch size of 512. We employed the Adam optimizer, setting the learning rate to 0.001. We focus on

two key convergence properties: $K^\intercal(Q\mathbf{1} + q_c)$ approaching the zero vector $\mathbf{0}_d$ and $V$ converging to $a\mathbf{1}_d\mathbf{1}_d^\intercal + bI_{d\times d}$, where $a$ and $b$ are constants in $\mathbb{R}$. The conditions $K^\intercal(Q\mathbf{1} + q_c) = \mathbf{0}_d$ and $V = a\mathbf{1}_d\mathbf{1}_d^\intercal + bI_{d\times d}$ imply that the function $g(Z_t; V, Q, K) = \sum_{i=1}^{t}(b-a)l_i$, effectively emulating the process of an online gradient descent method. We repeated 10 times. For verifying $K^\intercal(Q\mathbf{1} + q_c) = \mathbf{0}_d$, we will measure 2-norm of $K^\intercal(Q\mathbf{1} + q_c)$. Also for measuring the closeness of $V$ and $a\mathbf{1}_d\mathbf{1}_d^\intercal + bI_{d\times d}$, we will measure $\min_{a,b\in\mathbb{R}} \|V - (a\mathbf{1}_d\mathbf{1}_d^\intercal + bI_{d\times d})\|_{2,2}/b$. The results are demonstrated in the third plot of Figure G.1.

### G.11. Comparison with (Ahn et al., 2023; Zhang et al., 2023a; Mahankali et al., 2023)

The very recent studies by (Ahn et al., 2023; Zhang et al., 2023a; Mahankali et al., 2023) have demonstrated that if $Z_t = ((x_1, y_1), \ldots, (x_t, y_t), (x_{t+1}, 0))$ and the "instruction tuning" loss (i.e., $\mathbb{E}[\|\widehat{y}_{t+1} - y_{t+1}\|^2]$) is being minimized with a single-layer linear self-attention model, then a global optimizer among single-layer linear self-attention models yields the output $\widehat{y}_{n+1} = \eta \sum_{i=1}^{n} y_i x_i^\intercal x_{n+1}$. This output can be interpreted as a *gradient descent* algorithm, indicating that a single-layer linear self-attention model **implicitly** performs gradient descent. However, in the online learning setting where there are no $y$-labels, such an implicit gradient descent update-rule is hard to define. Compared to the previous studies, our global optimizer among single-layer linear self-attention models is an *explicit* and *online* gradient descent update for online learning. With a different loss (regret-loss v.s. instruction-tuning-loss), the techniques to obtain the seemingly similar results are also fundamentally different.

### G.12. Details of Experiments for Regret-loss Minimization

**Randomly generated loss sequences.** We use the same loss vectors as those in Section 3.2 for randomly generated loss functions, and compare the results with that using GPT-4. The results show that with regret-loss, both the trained single-layer self-attention model and the trained Transformers with multi-layer self-attention structures can achieve comparable regrets as FTRL and GPT-4. The results can be found in Figure G.3.

**Loss sequences with certain trends.** We investigate the case where the loss sequences have predictable trends such as linear-trend or sine-trend. One might expect that the performance of the trained Transformer would surpass the performance of traditional no-regret learning algorithms such as FTRL, since they may not be an optimal algorithm for the loss sequence with a predictable trend. We modify the training distribution by changing the distribution of random variable $Z$ (which generates the loss vectors $\ell_t$) to follow two kinds of trends: linear and sine functions. The results, as illustrated in Figure G.4, show that the trained single-layer self-attention model and the trained Transformer with multi-layer self-attention structures with regret-loss outperformed GPT-4 and FTRL in terms of regret, when the loss sequence is a linear trend. Similarly, Figure G.4 shows that the trained Transformer with multi-layer self-attention structures with regret-loss is comparable to GPT-4 and outperformed FTRL in terms of regret, when the loss sequence is a sine-trend. Note that the training dataset does not contain the sequence of losses. Nonetheless, by focusing on the overall trend during training, we can attain performance that is either superior to or on par with that of FTRL and GPT-4.

**Repeated games.** We then investigate the case of multi-player repeated games. We study 2x2, 3x3x3, 3x3x3x3 games, where each entry of the payoff matrix is sampled randomly from $\text{Unif}([0, 10])$. The results, as illustrated in Figure G.5, show that the trained single-layer self-attention model and the trained Transformer with multi-layer self-attention structures with regret-loss have a similar performance as that of FTRL. However, GPT-4 still outperforms the trained single-layer self-attention model and the trained Transformer with multi-layer self-attention structures in terms of regret. Since for repeated games (in which the environment faced by the agent can be less adversarial than that in the online setting), there might be a better algorithm than FTRL (see e.g., (Daskalakis et al., 2021)), while our self-attention models have a similar structure as FTRL (Theorem G.3 or Theorem G.2). Also, in practical training (with the empirical loss in Equation (G.3)), we possibly did not find the exact global minimum or stationary point of the *expected* loss in Equation (5.2). Hence, it is possible that GPT-4 may have lower regret than our trained models with the regret-loss.

**Two scenarios that caused regrettable behaviors of GPT-4.** Finally, we investigate the cases that have caused GPT-4 to have regrettable performance in Section 3.2. The results, which can be found in Figure E.7, show that both the trained single-layer self-attention model and the trained Transformer with regret-loss can achieve comparable no-regret performance as FTRL, and outperforms that of GPT-4. This validates that our new unsupervised training loss can address the regrettable cases, as our theory in Section 5.2 has predicted.

 G.12.1. TRAINING DETAILS OF EXPEIMENTS

For the multi-layer Transformer training, we used 4 layers, 1 head Transformer. For both single-layer and multi-layer, we employed the Adam optimizer, setting the learning rate to 0.001. During training, we conducted 2,000 epochs with a batch size 512. Moreover, when we trained for the loss sequences with the predictable trend, we used 4 layers, 1 head Transformer. For both single-layer and multi-layer, we employed the Adam optimizer, setting the learning rate to 0.001. During training, we conducted 9,000 epochs with a batch size of 512.

### G.13. Ablation Study on Training Equation (5.2)

In this section, we provide an ablation study that changes $N$ and $k$ in Equation (5.2). To be specific, we will set $N = 1, 2, 4$, $f(x, k) = \max(x, 0)^k$, $h(x) = \max(x, 0)^2$, and $k = 1, 2$. For the multi-layer Transformer training, we used 4 layers and 1 head Transformer. For both single-layer and multi-layer, we employed the Adam optimizer, setting the learning rate to 0.001. During training, we conducted 2,000 epochs with a batch size of 512. We experimented on the randomly generated loss sequences. Especially, we used the uniform loss sequence ($\ell_t \sim \text{Unif}([0, 10]^2)$), with the results in Figure G.6 and Figure G.7; and the Gaussian loss sequence ($\ell_t \sim \mathcal{N}(5 \cdot \mathbf{1}_2, I)$), with the results in Figure G.8 and Figure G.9.

# H. Limitations and Concluding Remarks

In this paper, we studied the online decision-making and strategic behaviors of LLMs quantitatively, through the metric of regret. We first examined and validated the no-regret behavior of several representative pre-trained LLMs in benchmark settings of online learning and games. As a consequence, (coarse correlated) equilibrium can oftentimes emerge as the long-term outcome of multiple LLMs playing repeated games. We then provide some theoretical insights into the no-regret behavior, by connecting pre-trained LLMs to the follow-the-perturbed-leader algorithm in online learning, under certain assumptions. We also identified (simple) cases where pre-trained LLMs fail to be no-regret, and thus proposed a new unsupervised training loss, *regret-loss*, to provably promote the no-regret behavior of Transformers without the labels of (optimal) actions. We established both experimental and theoretical evidence for the effectiveness of our regret-loss.

As a first attempt toward rigorously understanding the online and strategic decision-making behaviors of LLMs through the metric of regret, We provide the following limitations and list some potential directions for future research:

- There are more than one definitions of (dynamic-)regret in the online learning literature, and we mainly focused on the so-called *external-regret* in the literature. There are some other regret metrics we have studied, e.g., swap-regret (Blum & Mansour, 2007), which may lead to stronger equilibrium notions in playing repeated games.

- Our new regret-loss has exhibited promises in our experiments for training modest-scale Transformers. One limitation is that we haven't trained other larger-scale models, such as Foundation Models, for decision-making.

- No-regret behavior can sometimes lead to better outcomes in terms of social efficiency (Blum et al., 2008; Roughgarden, 2015; Nekipelov et al., 2015). It would thus be interesting to further validate the efficiency of no-regret LLM agents in these scenarios, as well as identifying new prompts and training losses for LLMs to promote the efficiency of the outcomes.

- To evaluate the performance quantitatively, we focused on online learning and games with *numeric valued* payoffs. It would be interesting to connect our no-regret-based and game-theoretic framework with existing multi-LLM frameworks, e.g., debate, collaborative problem-solving, and human/social behavior simulation, with potentially new notions of regret (defined in different spaces) as performance metrics.

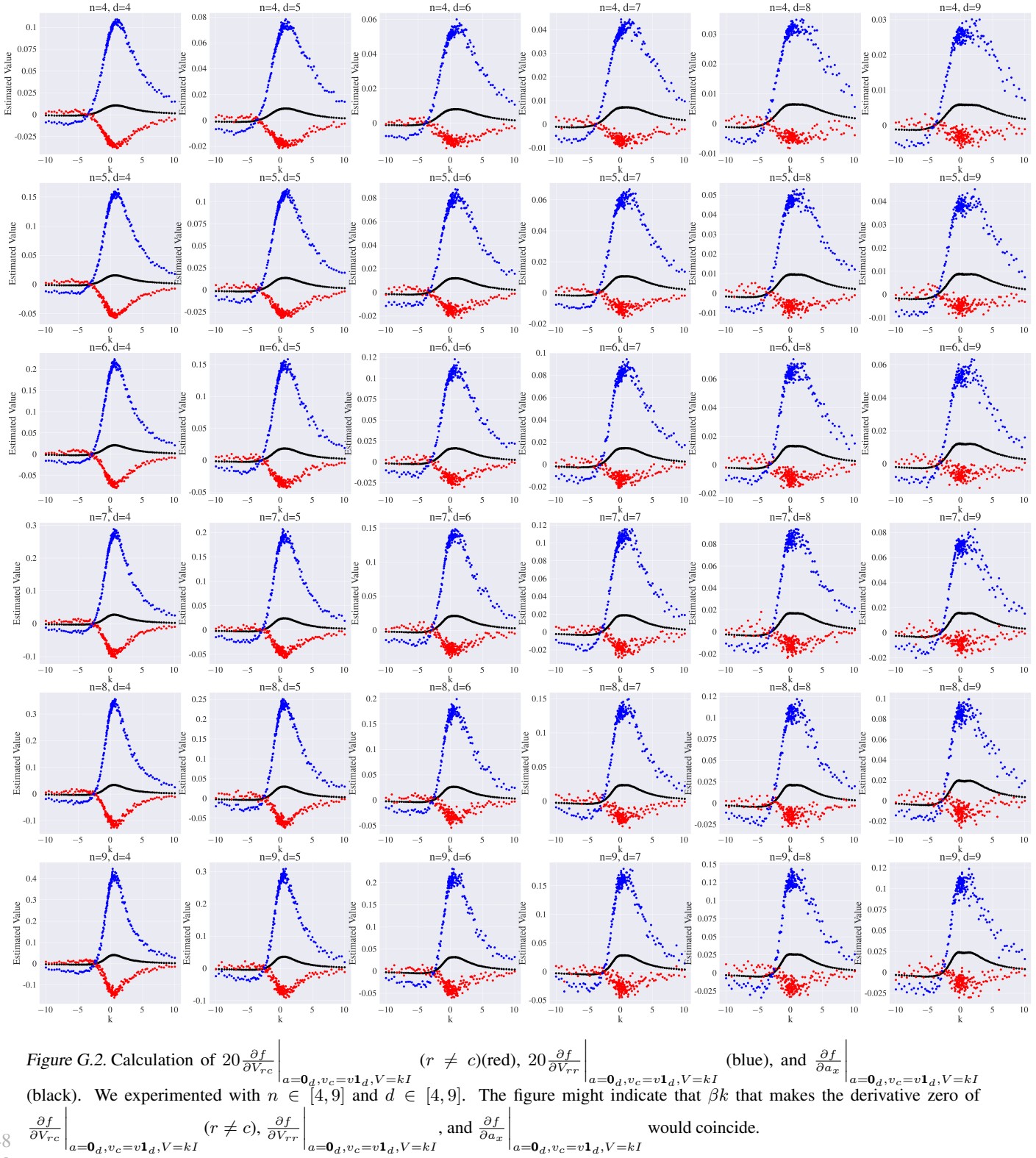

*Figure G.2.* Calculation of $20\frac{\partial f}{\partial V_{rc}}\Big|_{a=\mathbf{0}_d,v_c=v\mathbf{1}_d,V=kI}$ ($r \neq c$)(red), $20\frac{\partial f}{\partial V_{rr}}\Big|_{a=\mathbf{0}_d,v_c=v\mathbf{1}_d,V=kI}$ (blue), and $\frac{\partial f}{\partial a_x}\Big|_{a=\mathbf{0}_d,v_c=v\mathbf{1}_d,V=kI}$ (black). We experimented with $n \in [4,9]$ and $d \in [4,9]$. The figure might indicate that $\beta k$ that makes the derivative zero of $\frac{\partial f}{\partial V_{rc}}\Big|_{a=\mathbf{0}_d,v_c=v\mathbf{1}_d,V=kI}$ ($r \neq c$), $\frac{\partial f}{\partial V_{rr}}\Big|_{a=\mathbf{0}_d,v_c=v\mathbf{1}_d,V=kI}$, and $\frac{\partial f}{\partial a_x}\Big|_{a=\mathbf{0}_d,v_c=v\mathbf{1}_d,V=kI}$ would coincide.

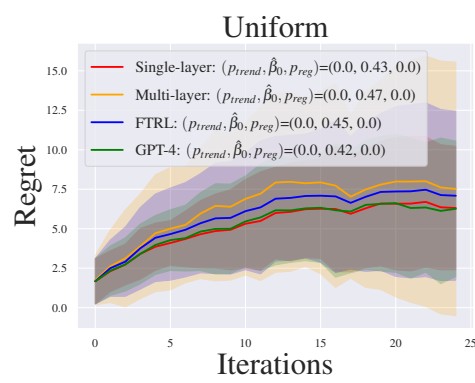
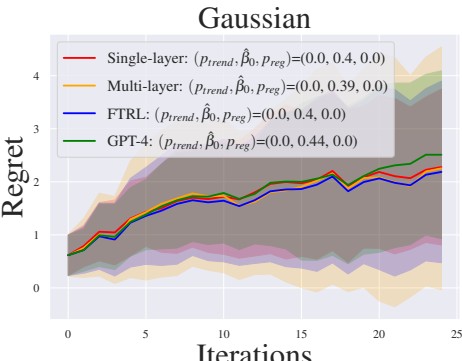

*Figure G.3.* Regret performance for the randomly generated loss sequences that are generated by Gaussian with truncation and uniform distribution. No-regret behaviors of single-layer and multi-layer self-attention models are validated by both of our frameworks (low $p$-values and $\widehat{\beta}_0 < 1$).

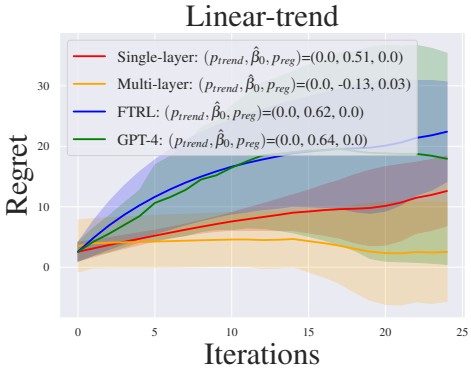
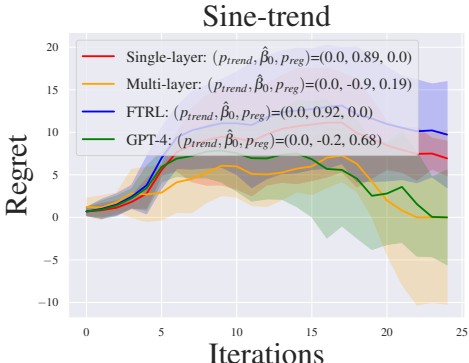

*Figure G.4.* Regret performance for the randomly generated loss sequences that are generated by linear-trend and sine-trend. No-regret behaviors of single-layer and multi-layer self-attention models are validated by both of our frameworks (low $p$-values and $\widehat{\beta}_0 < 1$).

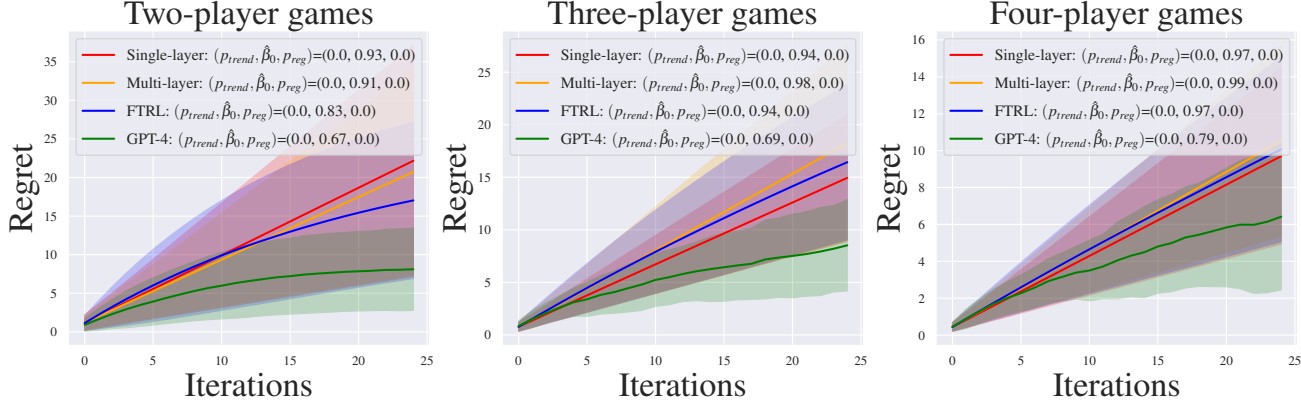

*Figure G.5.* Regret performance for the game with two players, three players, and four players general-sum games. No-regret behaviors of single-layer and multi-layer self-attention models are validated by both of our frameworks (low $p$-values and $\widehat{\beta}_0 < 1$).

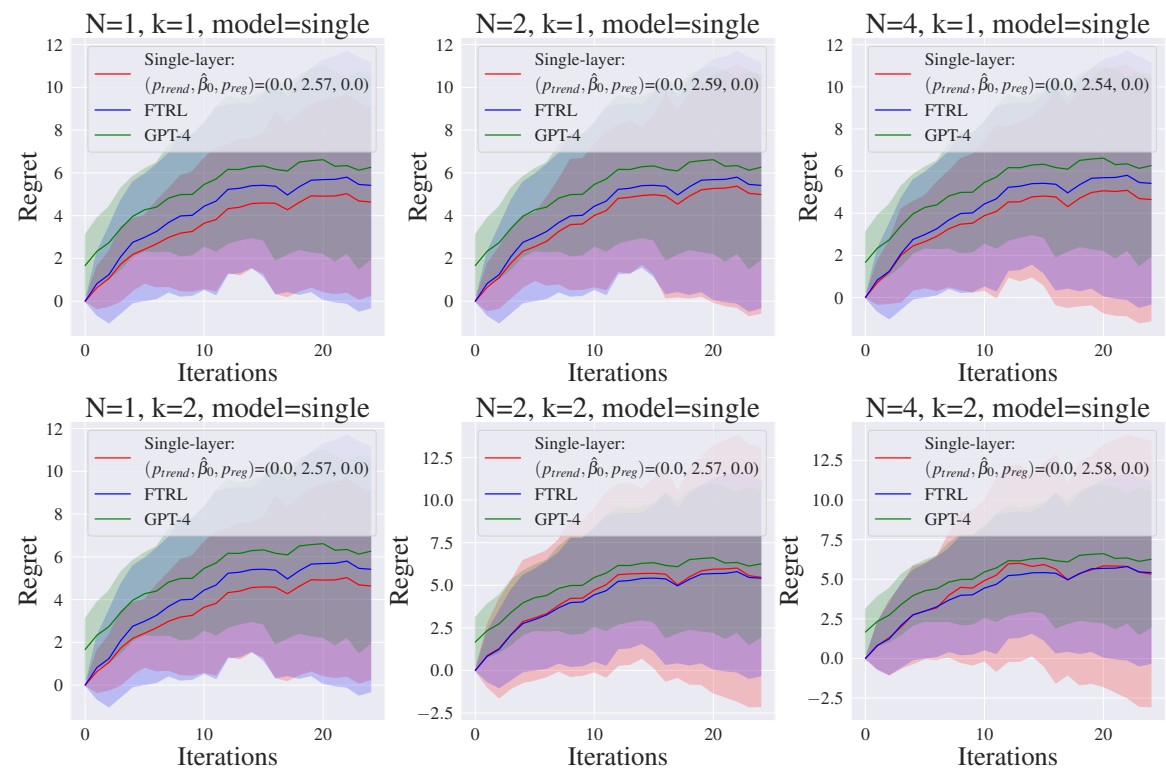

*Figure G.6.* Ablation study for the uniform loss sequence trained with single-layer self-attention layer and `Softmax` projection.

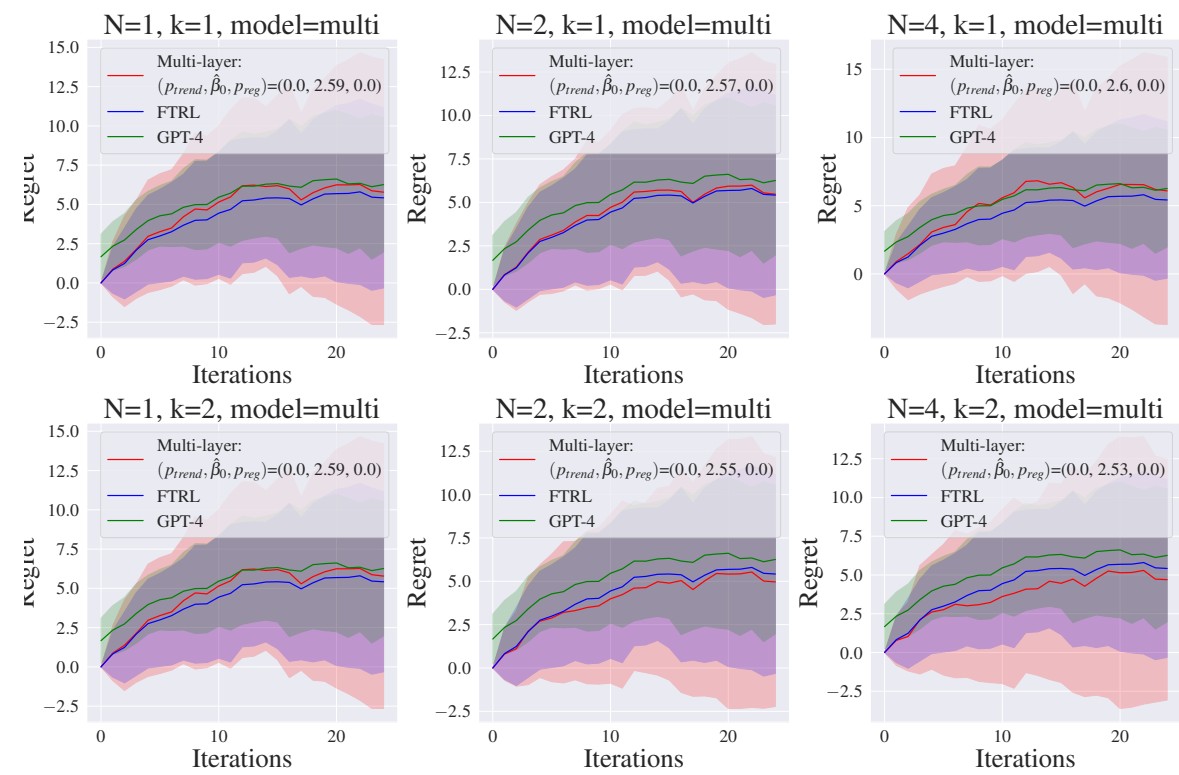

*Figure G.7.* Ablation study for the uniform loss sequence trained with multi-layer self-attention layer and `Softmax` projection.

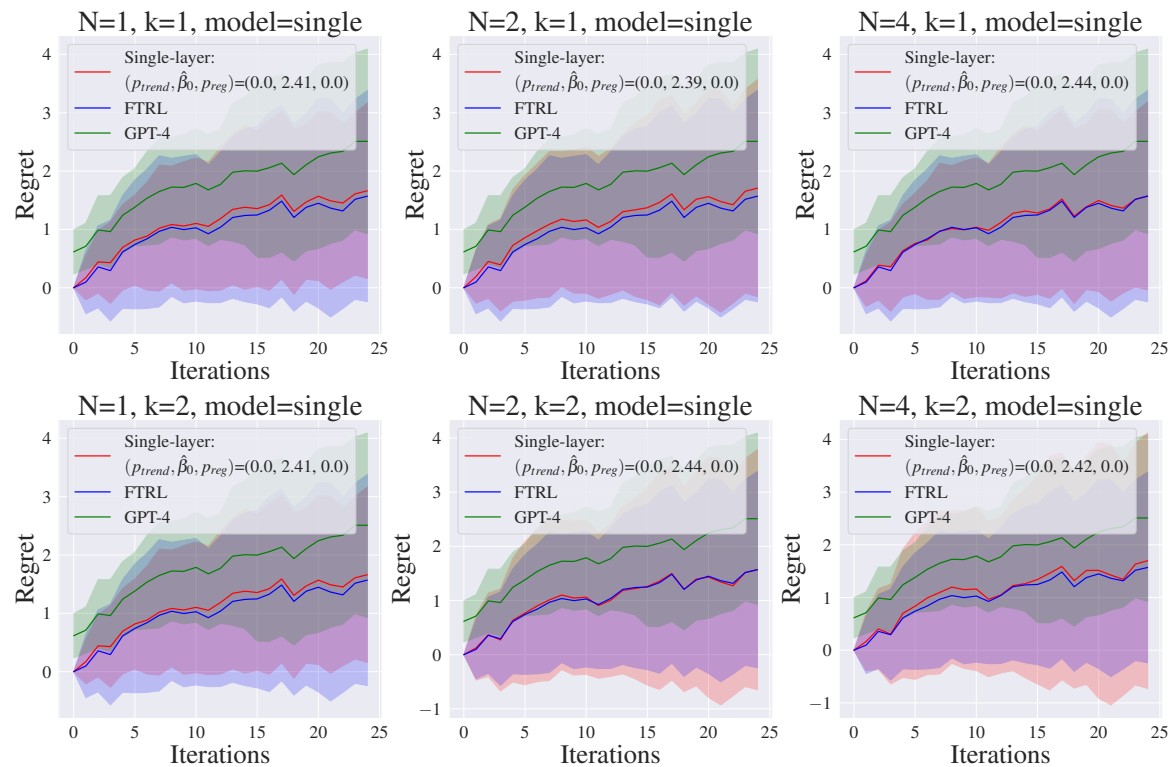

*Figure G.8.* Ablation study for the Gaussian loss sequence trained with single-layer self-attention layer and `Softmax` projection.

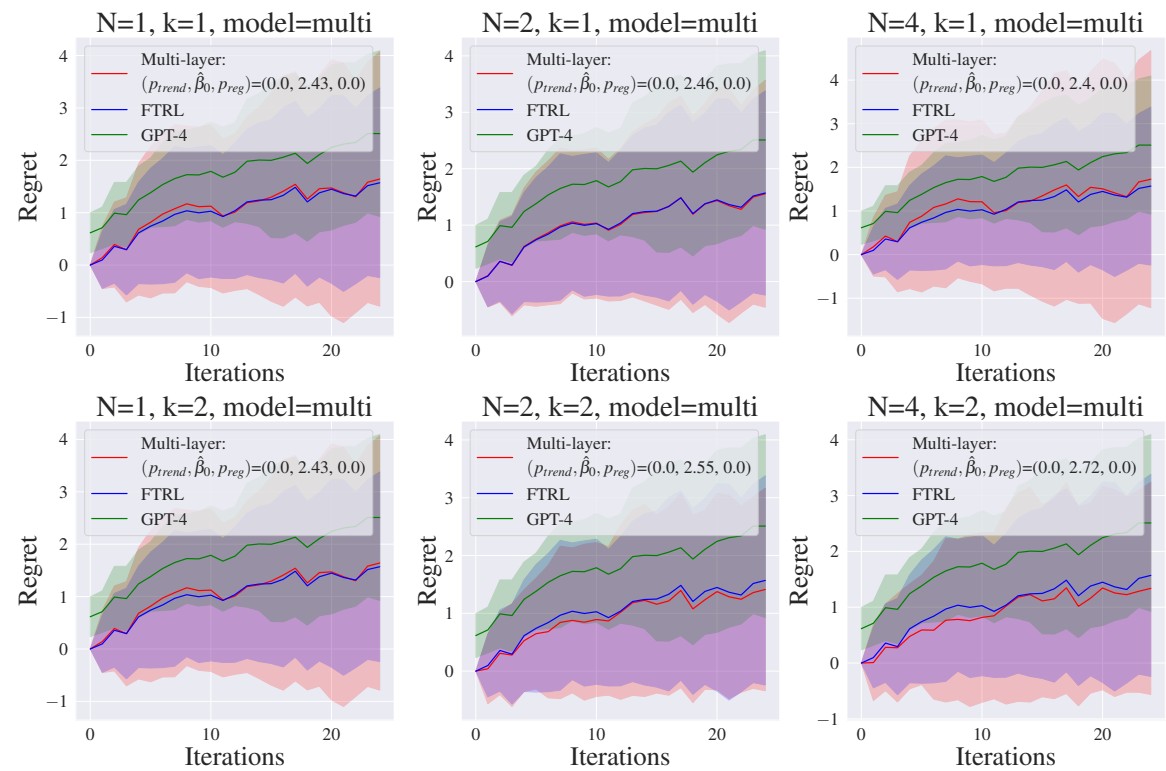

*Figure G.9.* Ablation study for the Gaussian loss sequence trained with single-layer self-attention layer and `Softmax` projection.

