# OpenReview forum: "Do LLM Agents Have Regret? A Case Study in Online Learning and Games"
_ICML.cc/2024/Workshop/Agentic_Markets — Agentic Markets @ ICML'24 Poster_

### Official Review · Reviewer_Z83z · 2024-06-10
**Review of Submission 6**

**Rating:** 7
**Confidence:** 3

**Review:**

**Quality**

In the age of LLMs and foundation models, this paper asks the important question of whether or not regret minimization is a suitable lens to study LLMs with, and if so how standard models perform using these lens. Overall, the paper does a good job of balancing both empirical observations and theoretical results. Due to the complexity of LLMs, the paper focuses on benchmark settings such as online learning with linear loss functions and repeated games, providing a rather comprehensive suite of results which establishes that LLMs exhibit strong no-regret guarantees in certain scenarios, but can also have failure modes.

**Clarity**

The paper is quite notation heavy for a workshop paper, which gives less space for ample explanation of the trend-checking/regression-based framework and LLM experimental setup studied. All in all, it makes for quite an unclear reading - it is not clear how the results are being derived within the main text itself. Otherwise, the appendices are very substantial and the explanations of the methods used are reasonably clear. I would suggest some reshuffling or simplification of the technical notation to improve the readability of the paper.

**Originality**

The marriage between the regret minimization framework most commonly seen in online learning and foundation models/LLMs is a fascinating direction. Given the plethora of modern research into LLMs, using regret as a theoretically sound metric for evaluating and studying them is a very interesting perspective. The approach of having results based around intuition from online learning literature makes for a convincing analysis which manages to lead to an original unsupervised training loss called regret-loss.

**Significance**

Overall the paper analyzes the connection between LLM agents in canonical online learning problems and game theoretic equilibria. While much of the analysis uses technical ideas from the literature, the modifications of standard frameworks which are needed to make these techniques amenable to the LLM setting makes this paper an important first step towards obtaining a better intuition for the behavior and performance of LLMs for particular important tasks. That being said, the paper as it stands is very dense with notation and results, so a writing pass to improve clarity and flow would be recommended for the purposes of the workshop.

---

### Official Review · Reviewer_Rzw2 · 2024-06-12
**An interesting work that is not supported adequately in the main body of the paper.**

**Rating:** 7
**Confidence:** 3

**Review:**

The authors explore the no-regret capabilities of Large Language Models (LLMs) through a series of experimental single- and multi-player setups. They propose an alternative training loss for LLMs that under assumptions guarantees the no-regret property for a trained LLM.

Strengths:

1. The theoretical results are an early step toward understanding the performance of LLMs.
2. The insights about the justification the LLMs provide for their no-regret behavior are quite interesting and could motivate further work on the topic.

Weaknesses:

1. The main body of the paper is not self-contained. Specifically, the authors promise four different contributions, none of which is sufficiently supported in the main text. I attempted to base my evaluation on the information found in the Appendix, but I  believe the following issues should be addressed before publication.
2. The claim that LLMs exhibit no-regret behavior, expanded in Section 3, requires an understanding of the evaluation framework, and some statistical, or otherwise, data that support it.
3. The claim that GPT-4 fails to achieve no-regret under certain conditions should be supported by an example in Section 3.8.
4. Regarding the theoretical claims, the formal (or nearly formal) statements should be provided in the main body of the paper along with the most important assumptions, e.g., in Theorem 4.1, what is Assumption 1?